# The GTPase activating protein Gyp7 regulates Rab7/Ypt7 activity on late endosomes

Nadia Füllbrunn[1,2]⊕, Raffaele Nicastro[3]⊕, Muriel Mari[4]⊕, Janice Griffith[5], Eric Herrmann[6]⊕, René Rasche[6]⊕, Ann-Christin Borchers[1]⊕, Kathrin Auffarth[1]⊕, Daniel Kümmel[6]⊕, Fulvio Reggiori[4,5]⊕, Claudio De Virgilio[3]⊕, Lars Langemeyer[1,2]⊕, and Christian Ungermann[1,2]⊕

**Organelles of the endomembrane system contain Rab GTPases as identity markers. Their localization is determined by guanine nucleotide exchange factors (GEFs) and GTPase activating proteins (GAPs). It remains largely unclear how these regulators are specifically targeted to organelles and how their activity is regulated. Here, we focus on the GAP Gyp7, which acts on the Rab7-like Ypt7 protein in yeast, and surprisingly observe the protein exclusively in puncta proximal to the vacuole. Mistargeting of Gyp7 to the vacuole strongly affects vacuole morphology, suggesting that endosomal localization is needed for function. In agreement, efficient endolysosomal transport requires Gyp7. In vitro assays reveal that Gyp7 requires a distinct lipid environment for membrane binding and activity. Overexpression of Gyp7 concentrates Ypt7 in late endosomes and results in resistance to rapamycin, an inhibitor of the target of rapamycin complex 1 (TORC1), suggesting that these late endosomes are signaling endosomes. We postulate that Gyp7 is part of regulatory machinery involved in late endosome function.**

## Introduction

Maintaining membrane integrity and organelle homeostasis requires intracellular transport between organelles, which occurs via vesicular transport or membrane contact sites. During vesicular transport, proteins are concentrated in forming vesicles. These pinch off from a donor membrane and fuse with an acceptor membrane. Fusion of vesicles relies on a whole set of proteins, termed the fusion machinery, including SNAREs, tethering factors, and Rab GTPases.

Rab GTPases (Rabs) are key identity markers of endomembranes (Müller and Goody, 2018; Borchers et al., 2021; Barr, 2013; Hutagalung and Novick, 2011). They function as molecular switches and exist in an active GTP-bound and an inactive GDP-bound form. Rabs require specific guanine nucleotide exchange factors (GEFs) for their GTP loading and GTPase activating proteins (GAPs) for their inactivation. Rabs exist in the cytosol in complex with the chaperone-like guanine nucleotide dissociation inhibitor (GDI) and randomly associate with membranes via their C-terminal prenyl anchor. If they encounter their GEF, it promotes nucleotide exchange of GDP for the more abundant GTP by destabilizing the nucleotide-binding pocket, which triggers loading with the more abundant GTP and stable membrane association. In this active, membrane-bound form, Rabs interact with effectors, such as tethering factors, to mediate

fusion. As Rabs are inefficient enzymes (Müller and Goody, 2018), GAPs are required to trigger GTP hydrolysis. The Rab-GDP is subsequently extracted by GDI from membranes, thus completing the Rab cycle.

Along the endolysosomal pathway, Rab5 and Rab7 define organelle identity of early and late endosomes and lysosomes by coordinating membrane fission and fusion processes (Borchers et al., 2021). Endocytic vesicles deliver their cargo to Rab5-positive endosomes. These endosomes change in morphology by sorting cargo into intraluminal vesicles with support of the ESCRT complexes, which results in the formation of multivesicular bodies (MVBs) or late endosomes, while other proteins are rerouted into retrograde tubules (McNally and Cullen, 2018; Vietri et al., 2020). In yeast, endosomes accumulate in a prevacuolar compartment, proximal to the vacuole (Day et al., 2018). In addition, a subpopulation of endosomes, signaling endosomes, has been described, which carry a fraction of the otherwise vacuolar target of rapamycin complex 1 (TORC1) (Hatakeyama et al., 2019).

During endosome maturation, Rab5 (Vps21 in yeast) is replaced by Rab7 (Ypt7 in yeast) (Borchers et al., 2021; Rink et al., 2005; Poteryaev et al., 2010). This process seems to occur in a sharp transition, which is likely driven by Rab5 levels. These

[1]Department of Biology/Chemistry, Biochemistry Section, Osnabrück University, Osnabrück, Germany; [2]Center of Cellular Nanoanalytics, Osnabrück University, Osnabrück, Germany; [3]Department of Biology, University of Fribourg, Fribourg, Switzerland; [4]Department of Biomedicine, Aarhus University, Aarhus, Denmark; [5]Department of Cell Biology, University Medical Center Utrecht, Utrecht, Netherlands; [6]Institute of Biochemistry, University of Münster, Münster, Germany.

Correspondence to Christian Ungermann: cu@uos.de; Lars Langemeyer: lars.langemeyer@uos.de.

may activate the Rab7-GEF and recruit Rab7 to membranes. In turn, Rab7 may trigger Rab5 release by recruiting the corresponding Rab5 GAP. Mathematical modeling suggests that the crosstalk of GEF and GAP with the involved Rabs determine this transition (Del Conte-Zerial et al., 2008; Barr, 2013). This transition may be further tuned by corresponding Rab effectors. In agreement, reconstitution assays of the Rab5 GEF cascade together with Rab5 effectors showed strongly confined Rab5-positive zones on membranes (Bezeljak et al., 2020; Cezanne et al., 2020).

The conserved Mon1–Ccz1 complex was identified as the Ypt7 GEF complex in yeast (Nordmann et al., 2010) and subsequently in human cells (Gerondopoulos et al., 2012). Mon1–Ccz1 is a Vps21/Rab5 effector (Li et al., 2015; Cui et al., 2014; Langemeyer et al., 2020; Singh et al., 2014; Kinchen and Ravichandran, 2010). We showed before that Vps21/Rab5 both recruits and activates yeast and metazoan Mon1–Ccz1 on membranes (Langemeyer et al., 2020). This process is further enhanced by the membrane environment, which the complex samples (Herrmann et al., 2023), and allows Mon1–Ccz1 to target both endosomes and autophagosomes (Gao et al., 2018; Hegedűs et al., 2016; Herrmann et al., 2023). In *Drosophila* and human cells, the GEF complex contains a third subunit, whose loss results in strong autophagy and endosomal defects, and lysosomal cholesterol accumulation (Vaites et al., 2017; Dehnen et al., 2020; van den Boomen et al., 2020).

Yeast Mon1–Ccz1 is an endosomal complex (Gao et al., 2018, 2022), yet Ypt7 is required both on endosomes and the vacuole to promote recycling and fusion. Ypt7 has several effector proteins. Ypt7 binds the retromer complex, which is involved in membrane protein recycling (Liu et al., 2012; Balderhaar et al., 2010; Purushothaman et al., 2017). It also interacts with the inverted BAR protein Ivy1, a protein involved in signaling at endosomes and activity control of the Fab1 lipid kinase complex, which generates phosphatidylinositol-3,5-bisphosphate ($PI(3,5)P_2$) (Numrich et al., 2015; Varlakhanova et al., 2018; Malia et al., 2018). Finally, Ypt7 interacts with the homotypic fusion and vacuole protein sorting (HOPS) tethering complex, which is required for SNARE-mediated membrane fusion of endosomes, autophagosomes, and Golgi-derived AP-3 vesicles with the vacuole (Shvarev et al., 2022; Wickner and Rizo, 2017).

Less is known about the GAP-mediated inactivation of Ypt7. Almost all GAPs have a central Tre/Bub2/Cdc16 (TBC) domain with a catalytic arginine-glutamine finger (Albert et al., 1999). These fingers complete the nucleotide-binding site of a Rab and thus allow for GTP hydrolysis (Pan et al., 2006). Although Gyp7 has been one of the first identified GAPs, its substrate specificity remained unclear as the in vitro activity revealed low substrate specificity (Vollmer et al., 1999; Albert et al., 1999; Lachmann et al., 2012). However, Gyp7 seems to act on Ypt7 as its overexpression results in Ypt7 inactivation and vacuole fragmentation in vivo (Brett et al., 2008). Furthermore, Gyp7 can inhibit vacuole–vacuole fusion at the docking stage in vitro (Eitzen et al., 2000).

Yeast encodes for eight GAPs, but 11 Rabs, though the specificity of these GAPs to their Rab remains unclear. To inactivate Rabs, GAPs may decode the membrane by binding to specific proteins and/or recognize specific phosphoinositides. These interactions can occur as part of a Rab cascade, where the downstream Rab recruits the GAP of the upstream Rab (Barr, 2013). For mammalian Rab7, the four GAPs Armus/TBC1D2A, TBC1D2B, TBC1D5, and TBC1D15 have been identified. All indeed recognize membranes via lipid-binding motifs, coiled-coil motifs, or LC3-interacting regions (Stroupe, 2018; Popovic and Dikic, 2014; Kanno et al., 2010; Frasa et al., 2010; Jia et al., 2016; Zhang et al., 2005; Peralta et al., 2010). Most Rab7 GAPs function in autophagy, while TBC1D5, together with the retromer complex, specifically restricts Rab7 to endosomal microcompartments and affects signaling processes and endosomal maturation (Jimenez-Orgaz et al., 2018; Kvainickas et al., 2019).

Although Gyp7 has been identified as the only Ypt7-specific GAP, it remains unclear how and when Gyp7 inactivates Ypt7. We therefore set out to analyze Gyp7 function in detail. Here, we show that Gyp7 localizes in dot-like structures next to the vacuole, suggesting that they are of endosomal origin. Using in vitro assays, we demonstrate that Gyp7 has a high affinity for membranes, which enhances its GAP activity for membrane-bound Ypt7. We further show that Gyp7 overproduction can retain Ypt7 on late endosomes, which enhances endosomal TORC1 signaling. These Ypt7-positive endosomes lack ESCRTs, yet require ESCRTs for their formation. We thus speculate that these late endosomes correspond to signaling endosomes.

## Results

### Gyp7 localization depends on an intact endosomal system

In yeast, Ypt7 functions in multiple fusion and fission reactions at the vacuole as well as in formation of the membrane contact site between the vacuole and mitochondria (vacuolar and mitochondrial patch; vCLAMP) (Fig. 1 A). To clarify the Ypt7 pool targeted by Gyp7, we tagged Gyp7 C-terminally with mNeon-Green and determined its localization by fluorescence microscopy. We observed Gyp7 in single puncta proximal and peripheral to the vacuole (Fig. 1 B). Gyp7 was strongly concentrated in the so-called Class E compartments, which were also stained by the lipophilic dye FM4-64, upon inactivation of the ESCRT-IV subunit Vps4 (Babst et al., 1998) (Fig. 1 B). Here, Gyp7 colocalized with other endosomal proteins such as the Rab5-like Vps21 and the retromer subunit Vps35 (Fig. 1, C and D). In contrast, Msb3, the previously identified GAP of Vps21 that shows some GAP activity for Ypt7 as well (Lachmann et al., 2012), was not enriched in this compartment (Fig. 1 B).

To determine whether specific endosomal proteins are required for Gyp7 localization, we analyzed several mutants (Fig. 1, E and F; and Fig. S1 A), including deletions of the major Rab5 proteins Vps21 and Ypt52, their corresponding GEFs Vps9 and Muk1, respectively, the class C core vacuole/endosome tethering (CORVET) complex subunit Vps3, the endosomal Sec1/Munc18-like Vps45, the endosome-specific subunit of the phosphatidylinositol 3-kinase Vps34 (*vps38Δ*), and several proteins involved in endosomal retrograde transport (*snx4Δ*, *vps5Δ*, *vps35Δ*, *mvp1Δ*). None of these mutants abolished the distribution in puncta of Gyp7 completely. However, all impairing mutants of fusion proteins in the endosomal system, such as *vps21Δ ypt52Δ*, *vps3Δ*, or *vps45Δ*, had more than five times more Gyp7 puncta, which predominantly were localized more distal from the vacuole

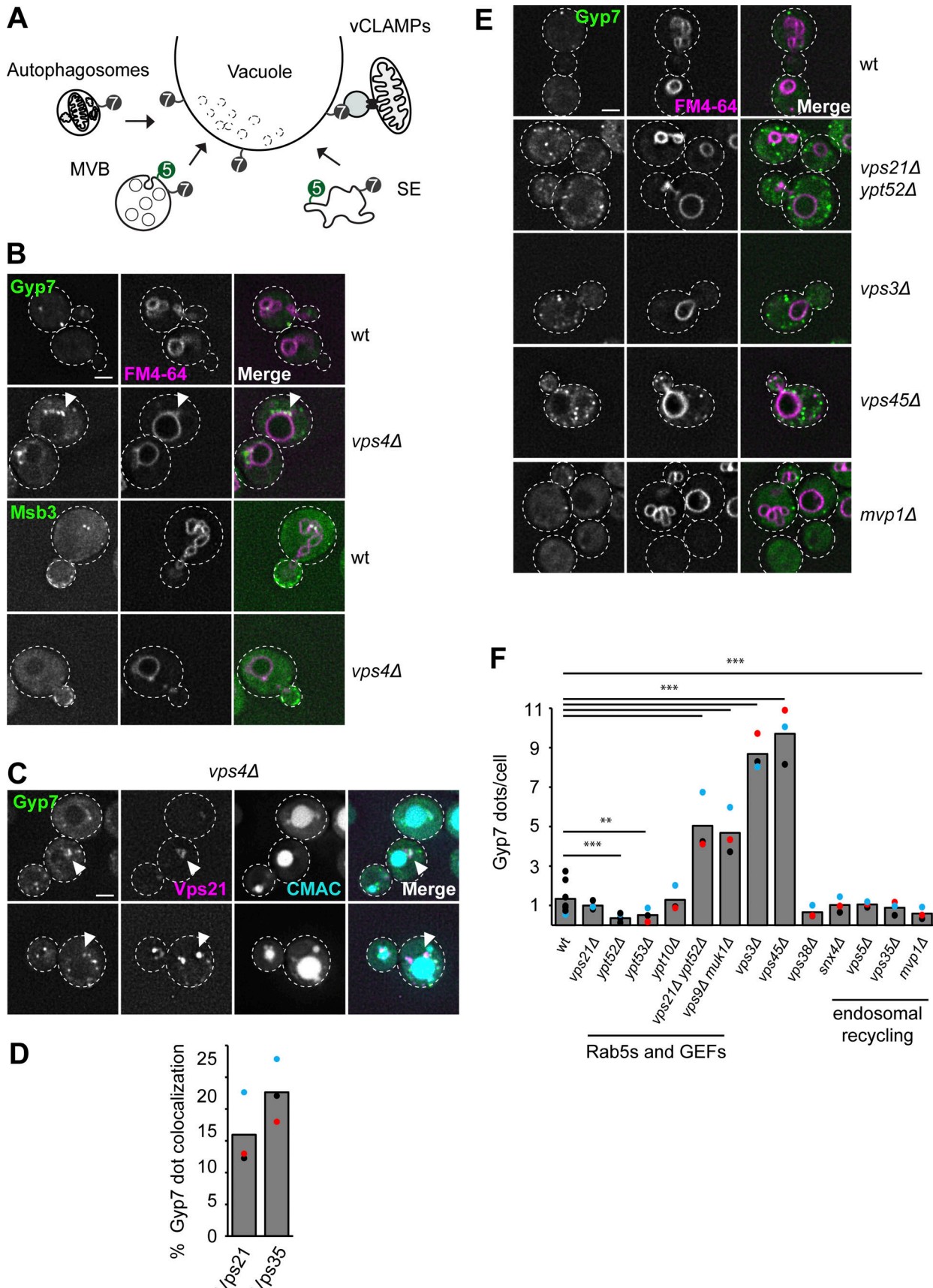

**Figure 1. Gyp7 localization depends on a functional endosomal system. (A)** Overview of Ypt7 function in fusion and fission reactions at the vacuole. For details, see text. **(B)** Localization of endogenously expressed Gyp7 and Msb3. Gyp7 and Msb3 were C-terminally tagged with mNeonGreen in wild-type (wt) and

*vps4Δ* cells. Vacuolar membranes were stained with FM4-64 (see Materials and methods). Cells were imaged by fluorescence microscopy. Individual slices are shown. Arrowheads depict Gyp7 accumulations. Dashed lines indicate yeast cell boundaries. Scale bar, 2 μm. **(C)** Localization of endosomal markers relative to Gyp7. Marker proteins mCherry-Vps21 and Vps35-2xmKate were coexpressed in *vps4Δ* cells encoding endogenous Gyp7-mNeonGreen. Vacuoles were stained with CMAC (see Materials and methods). Cells were imaged by fluorescence microscopy. Individual slices are shown. Arrowheads depict representative colocalization. Dashed lines indicate yeast cell boundaries. Scale bar, 2 μm. **(D)** Quantification of Gyp7 puncta colocalizing with endosomal markers in C. Cells (*n* ≥ 100) from three independent experiments were quantified in Fiji. Bar graphs represent the averages from three experiments and puncta represent the mean of each experiment. **(E)** Localization of Gyp7 in selected deletion mutants. Gyp7 was tagged with mNeonGreen in wild-type, *vps21Δ ypt52Δ*, *vps3Δ*, *vps45Δ*, and *mvp1Δ* cells. Vacuolar membranes were stained with FM4-64. Cells were imaged by fluorescence microscopy and individual slices are shown. Dashed lines indicate yeast cell boundaries. Scale bar, 2 μm. **(F)** Quantification of Gyp7 puncta per cell in E and Fig. S1 A. Cells (*n* ≥ 100) from three independent experiments were quantified in Fiji. Bar graphs represent the averages from three experiments and puncta represent the mean of each experiment. P value **<0.01, ***<0.001, using ANOVA one-way test.

(Fig. 1 E). This could be either explained by disruption of Gyp7 recruitment or an overall alteration of endosomal morphology per se. Furthermore, among all proteins involved in membrane recycling, only *MVP1* deletion caused a reduction in Gyp7 puncta. Similar observations were made for *ypt52Δ* and *ypt53Δ* cells. Our data suggest that Gyp7 recruitment does not depend on the presence of single endosomal proteins but on an intact endosomal system.

We also analyzed the influence of Gyp7 on Ypt7 function in autophagy and vCLAMP formation. Neither *GYP7* deletion nor its overexpression altered transport of the autophagy-specific Atg8 protein to the vacuole lumen upon starvation (Fig. S1, B–D). We noticed, however, that overexpression of Gyp7 resulted in slightly more Atg8-positive puncta in growth conditions (Fig. S1 C). To follow vCLAMPs, we overexpressed mCherry-tagged Vps39, which accumulates in wild-type cells between vacuoles and DAPI-stained mitochondria (Fig. S1, E and F). Again, manipulation of Gyp7 expression levels had no effect. In addition, Gyp7 did not localize to vCLAMPs.

We conclude that any deletion of key endosomal proteins results in multiple Gyp7-positive puncta, yet no release of Gyp7 from membranes. This suggests that Gyp7 recruitment to the endolysosomal system occurs independent of the analyzed endosomal proteins.

### Relocalization of Gyp7 to vacuoles impairs vacuole morphology

A major pool of Ypt7 is found on the vacuolar rim, while Gyp7 localizes in dot-like structures of the endolysosomal system. Nevertheless, overexpression of Gyp7 from the *GAL1* promoter can trigger vacuole fragmentation (Fig. 2, A and B) (Brett et al., 2008). This suggests that Gyp7-mediated inactivation of Ypt7 strongly impairs vacuole morphology.

To determine whether Gyp7 dynamically localizes to both vacuoles and endosomes to control Ypt7 activity, or functions exclusively at endosomes, we tagged the endosomal CORVET subunit Vps8 or the vacuolar zinc transporter Zrc1 with a nanobody against GFP (chromobody, CB) in strains expressing endogenous Gyp7-GFP, an approach we previously established to confine proteins at specific subcellular locations (Malia et al., 2018). We first analyzed vacuole morphology of strains exclusively expressing Vps8-CB or Zrc1-CB and observed no effect on vacuole morphology, indicating that tagging Vps8 or Zrc1 does not impair their functionality (Fig. 2, C and E). We then turned to strains that additionally expressed Gyp7-GFP or the catalytic dead version of Gyp7-GFP, Gyp7$^{R458K}$. Sequestering Gyp7 or

Gyp7$^{R458K}$ to endosomes via Vps8-CB confined these variants to single puncta, and vacuoles looked like wild-type (Fig. 2, D–G). In contrast, relocalizing Gyp7 but not Gyp7$^{R458K}$ to the vacuole via Zrc1-CB strongly fragmented vacuoles. Importantly, the estimated copy number of Zrc1 is significantly higher than that of Gyp7 as an important precondition for a knock-sideways experiment (Ho et al., 2018). This indicates that Gyp7, which was present in multiple puncta at the vacuole, inactivated Ypt7 here.

To exclude that the artificial confinement of Gyp7 to the vacuole via Zrc1-CB caused a non-specific effect on vacuole fusion or fission, we expressed the Ypt7$^{K127E}$ mutant in this background. Ypt7$^{K127E}$ has a fast nucleotide exchange and can bypass the Ypt7 GEF requirement and possibly also the requirement for the GAP (Kucharczyk et al., 2001; Cabrera and Ungermann, 2013). Indeed, Ypt7$^{K127E}$ expression completely rescued the vacuole morphology, indicating that the previously observed vacuole fragmentation was caused by Ypt7 inactivation at the vacuolar membrane. Our observations thus agree with a major functional role of Gyp7 at endosomes and not at the vacuole.

### Gyp7 is required for homeostasis of the endosomal system

To analyze the role of Gyp7 in endosomal functions, we analyzed cells lacking *GYP7* in growth and endocytosis assays. For growth assays, we spotted cells in serial dilutions on plates containing 4 mM Zn$^{2+}$, a stressor of the endosomal pathway (Fig. 3 A). Here, we observed a slight growth defect of *gyp7Δ*, which was comparable with the one of *vps21Δ* cells. Deletion of the Vps21 GAP Msb3 was even more deficient, suggesting that Gyp7 is as important for a functional endosomal pathway as normal Vps21 activity. We also analyzed whether Gyp7 is required for normal function of the TORC1, which localizes to signaling endosomes and lysosomes (Hatakeyama and De Virgilio, 2019; Hatakeyama et al., 2019) (Fig. 3 B). TORC1 is sensitive to the inhibitor rapamycin, and sensitivity of cells to this drug indicates defective targeting and/or function of this complex. Like *msb3Δ* and *tor1Δ* cells, yeast cells lacking Gyp7 were sensitive to rapamycin. Similarly, cells with deletions of proteins involved in endosomal recycling (*vps35Δ*, *vps5Δ*) or Golgi-to-vacuole trafficking (*apl5Δ*) showed comparable sensitivity to rapamycin, whereas cells expressing a non-phosphorylatable Fab1 mutant are resistant to rapamycin (Chen et al., 2021) (Fig. S2 A). Importantly, tagging of Gyp7 with either mNeonGreen or GFP was without effect on growth, indicating that this modification does not interfere with its function (Fig. 3, A and B). Thus, Gyp7 function affects TORC1 function within the endolysosomal system.

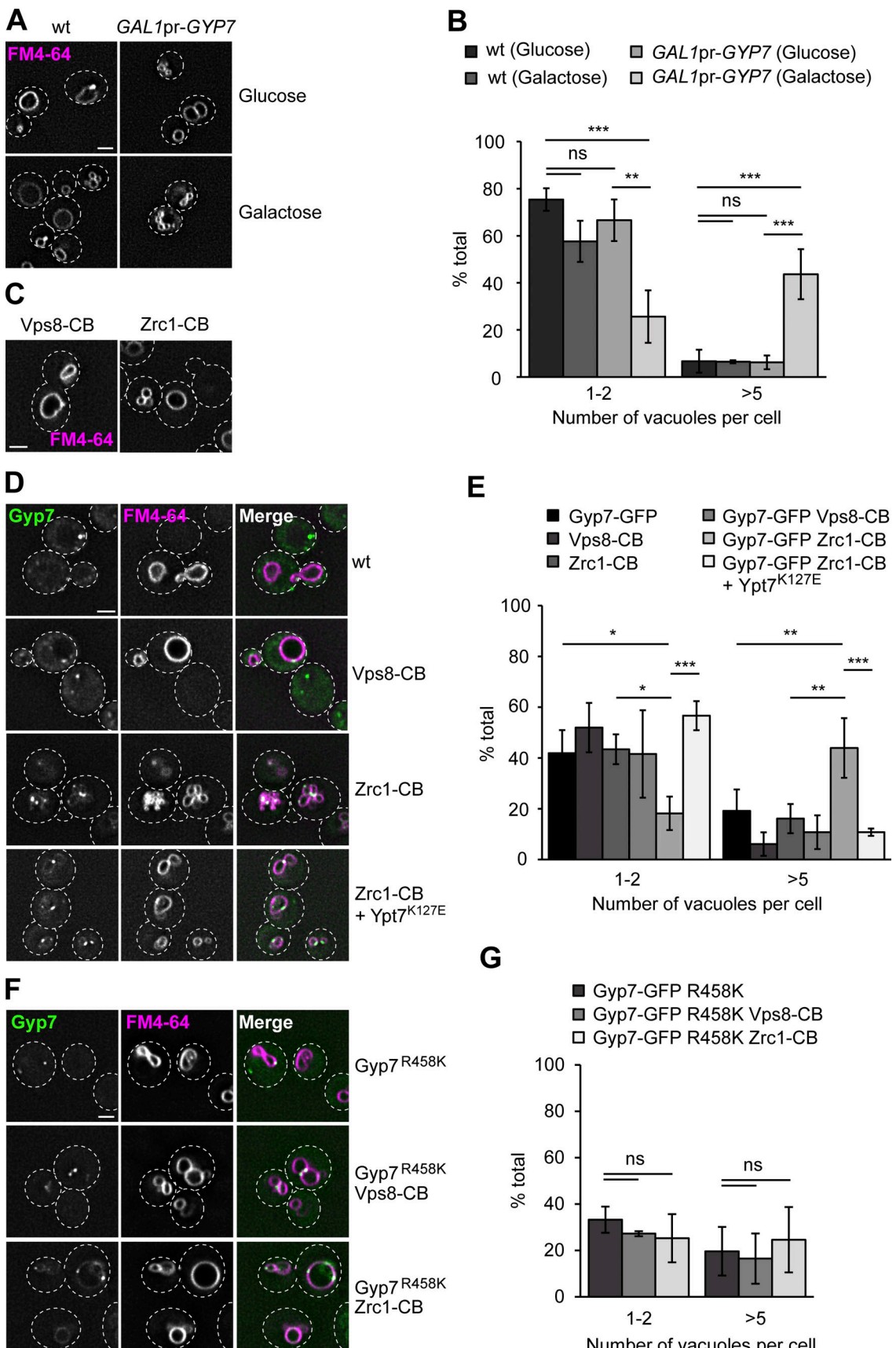

Figure 2. **Vacuolar localization of Gyp7 impairs vacuolar function. (A)** Vacuole morphology upon galactose-induced overexpression of Gyp7. Gyp7 was expressed from the *GAL1* promoter. Wild-type (wt) cells and cells encoding *GAL1*pr-*GYP7* were grown in glucose- or galactose-containing media (see Materials

and methods). Vacuolar membranes were stained with FM4-64. Cells were imaged by fluorescence microscopy and individual slices are shown. Dashed lines indicate yeast cell boundaries. Scale bar, 2 µm. **(B)** Quantification of the number of vacuoles per cell in A. Cells were grouped into three different classes: 1–2 vacuoles, 3–4 vacuoles (not shown), and >5 vacuoles. Cells (n ≥ 100) from three independent experiments were quantified in Fiji. Bar graphs represent the averages and error bars the SD from three experiments. P value ns, **<0.01, ***<0.001 using ANOVA one-way test. **(C)** Vacuole morphology of cells expressing Vps8- or Zrc1-Chromobody. Vps8 and Zrc1 were C-terminally tagged with a nanobody against GFP (CB). Vacuolar membranes were stained with FM4-64. Cells were imaged by fluorescence microscopy and individual slices are shown. Dashed lines indicate yeast cell boundaries. Scale bar, 2 µm. **(D)** Vacuole morphology of cells with Gyp7 targeted to endosomes or the vacuole. Vps8 and Zrc1 were C-terminally tagged with CB in cells expressing Gyp7-GFP. Where indicated, a Ypt7 fast-cycling mutant (Ypt7$^{K127E}$) was expressed from an integrative plasmid. Vacuolar membranes were stained with FM4-64. Cells were imaged by fluorescence microscopy and individual slices are shown. Dashed lines indicate yeast cell boundaries. Scale bar, 2 µm. **(E)** Quantification of the number of vacuoles per cell in C and D. Cells were classified as in B. Cells (n ≥ 150) from three independent experiments were quantified in Fiji. Bar graphs represent the averages and error bars the SD from three experiments. P value *<0.05, **<0.01 and ***<0.001, using ANOVA one-way test. **(F)** Vacuole morphology of cells expressing Gyp7$^{R458K}$, the catalytic dead mutant of Gyp7. The mutation was introduced into cells expressing Gyp7-GFP. Where indicated, Vps8 and Zrc1 were C-terminally tagged with a CB. Vacuolar membranes were stained with FM4-64. Cells were imaged by fluorescence microscopy and individual slices are shown. Dashed lines indicate yeast cell boundaries. Scale bar, 2 µm. **(G)** Quantification of the number of vacuoles per cell in F. Cells were classified as in B. Cells (n ≥ 130) from three independent experiments were quantified in Fiji. Bar graphs represent the averages and error bars the SD from three experiments. P value ns, using ANOVA one-way test.

To analyze the role of Gyp7 in endocytosis, we followed the transport of the methionine transporter Mup1-GFP in wild-type and *gyp7Δ* cells. In the absence of methionine, Mup1 accumulates at the plasma membrane (Fig. 3 C). Once methionine is added, Mup1 is endocytosed and transported via endosomes to the vacuole lumen. The initial uptake of Mup1 and delivery to endosomes at early time points upon methionine addition was comparable in both tested strains (Fig. S2, B and C). In contrast, *gyp7Δ* cells showed a clear delay in Mup1 delivery to the vacuole at later time points, i.e., 20–30 min after methionine addition, which was reflected by a decreased vacuole/plasma membrane Mup1 intensity ratio and more endosomal Mup1 (Fig. 3, D and E). Overall, we conclude that Gyp7 is required for efficient endocytosis and thus endosomal functions.

**Gyp7 activity depends on the membrane environment**

To understand Gyp7 function and GAP activity in more detail, we adapted a simple in vitro assay to our necessities (Thomas et al., 2021). Liposomes with a vacuole mimicking lipid (VML) composition (Zick and Wickner, 2014) were incubated with prenylated Ypt7 in complex with GDI in the presence of EDTA, GTP, and MgCl$_2$ (see Materials and methods). Under these conditions, prenylated Ypt7 is chemically activated and loaded with GTP, and thus becomes resistant to free GDI (molar ratio of GDI to Ypt7 is 1:1) unless its bound GTP is hydrolyzed to GDP with the help of a GAP. To determine the membrane-bound fraction of Ypt7, liposomes are floated in a sucrose gradient before analyzing the input and floated material by western blotting (Fig. 4 A). In the absence of a GAP, Ypt7 was anchored to liposomes and not extracted by GDI. In the presence of increasing amounts of full-length Gyp7, corresponding to a molar ratio of 1:20,000 to 1:32 (Gyp7 to Ypt7), Ypt7 was efficiently inactivated and extracted by GDI as shown by the decreasing amount of Ypt7 in the floated fraction (Fig. 4, B and C). We initially incubated samples for 1 h. To analyze the kinetics of Gyp7, as determined by GDI extraction, we incubated reactions containing 0.75 nM Gyp7 for different time points and then observed the membrane association of Ypt7 (Fig. 4, D and E). Our data revealed that 20 min were sufficient for almost 90% of Gyp7-mediated GTP-hydrolysis on Ypt7. Unless indicated otherwise, we incubated Ypt7-liposomes with 3.75 nM Gyp7 for

10 min in the following experiments to allow for efficient inactivation and membrane removal of Ypt7.

To determine whether Gyp7 associated with membranes, we added Gyp7 to liposomes and analyzed binding to membranes in a simple liposome sedimentation assay (Fig. 4, F and G). Gyp7 strongly pelleted in liposome-containing samples indicating that it binds membranes, while pelleting of Gyp7 in the absence of liposomes resulted in negligible background. The VML mixture of our liposomes contains a complex lipid mixture of 47 mol% phosphatidylcholine (PC), 18 mol% phosphatidylethanolamine (PE), 18 mol% phosphatidylinositol, 1 mol% phosphatidylinositol-3-phosphate, 4.4 mol% phosphatidylserine, 2 mol% phosphatidic acid, 1% diacylglycerol, and 8% ergosterol. We asked if a simpler mixture of 82 mol% dilinoleoyl phosphatidylcholine (DLPC) and 18 mol% dilinoleoyl phosphatidylethanolamine (DLPE) would have the same effect. However, Gyp7 was completely inactive in our assay (Fig. 4, H and I), as it did not bind to these membranes efficiently (Fig. 4, J and K). Importantly, association of Ypt7 with membranes was unaffected by the liposome composition (Fig. 4 H). Inefficient GAP activity of Gyp7 could thus be simply explained by its poor membrane binding.

To confirm that GDI is not limiting in our assay and able to extract Ypt7-GDP from PC/PE liposomes, we added 10-fold more GDI to our reactions (Fig. S3, A and B). We observed similar levels of Ypt7 extraction on VMLs either in the absence or presence of excess GDI, suggesting that the GDI available in solution was sufficient to extract all Ypt7-GDP from membranes as soon as it became available during our assay (Fig. S3, A and B). Importantly, addition of excess GDI did not significantly decrease the amount of Ypt7 bound to PC/PE liposomes, indicating that GDI is not limiting in our assay. Furthermore, we took advantage of the catalytically active TBC domain of Gyp1 (Gyp1-46), which was previously described to nonspecifically target Ypt7 among several other Rabs in solution and does not rely on membranes for its activity (Brett and Merz, 2008; Eitzen et al., 2000). Upon titration of Gyp1-46 instead of Gyp7 into our assay, we observed GAP activity toward membrane-bound Ypt7, followed by GDI extraction, on VMLs as well as on PC/PE liposomes, suggesting that GDI is in principle able to extract Ypt7-GDP from both VMLs and PC/PE liposomes (Fig. 4, L and M). Interestingly, more than 1,000-fold more Gyp1-46 was required to achieve comparable

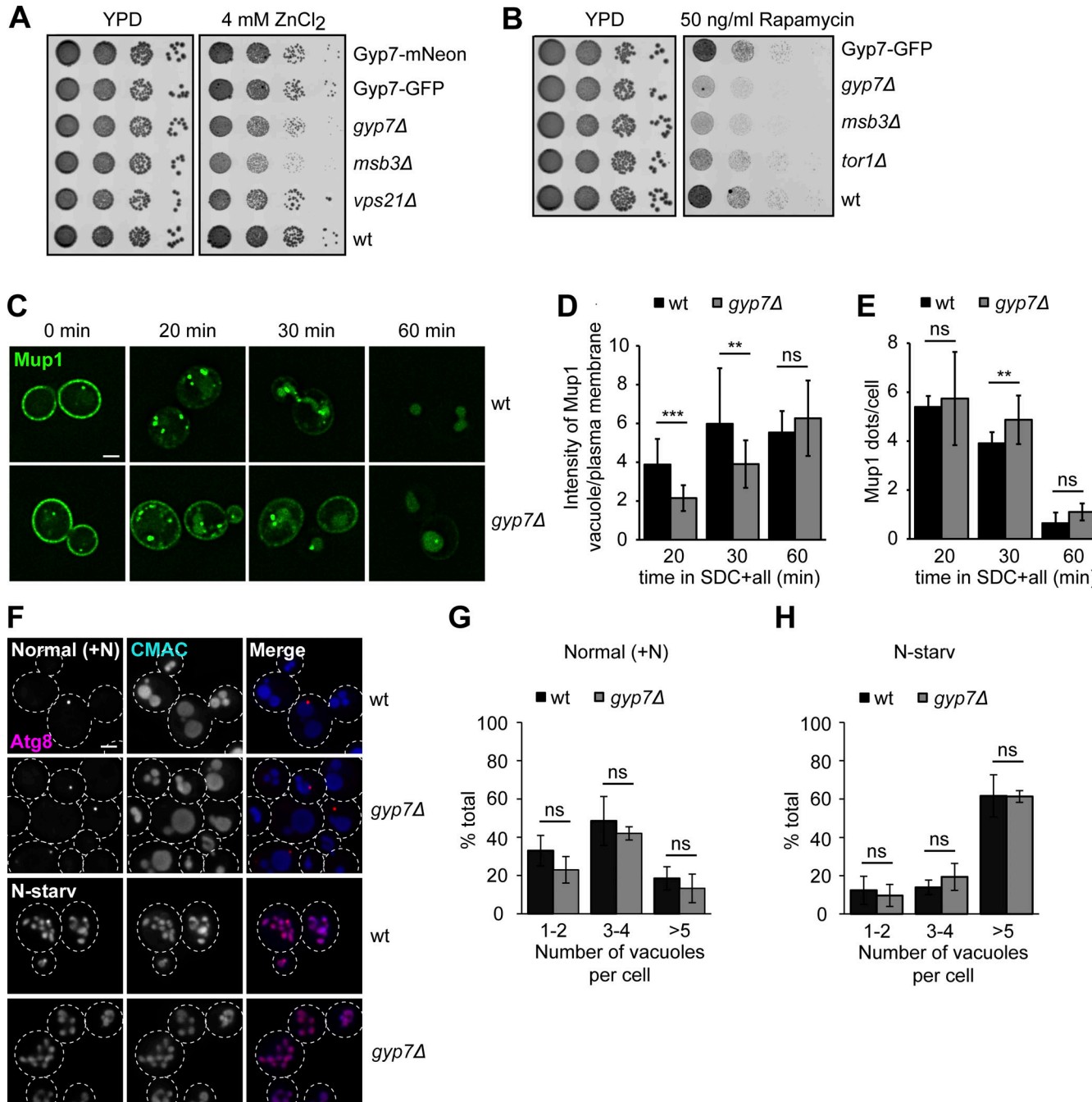

Figure 3. **Gyp7 is required for endosomal physiology and efficient endocytosis. (A)** Growth assay on ZnCl$_2$-containing plates. Indicated yeast strains were grown to the same OD$_{600}$ in YPD media and serial dilutions were spotted onto agar plates containing YPD or YPD supplemented with 4 mM ZnCl$_2$ (see Materials and methods). Plates were incubated at 30°C for several days before imaging. Images are representative for three independent experiments. **(B)** Growth assay on rapamycin-containing plates. Indicated yeast strains were spotted onto agar plates containing YPD or YPD supplemented with 50 ng/ml rapamycin as in A. Plates were incubated at 30°C for several days before imaging. Images are representative of three independent experiments. **(C)** Endocytosis of Mup1 in wild-type (wt) and *gyp7Δ* cells. Cells were grown to logarithmic phase in SDC-MET media, analyzed by fluorescence microscopy, and then shifted to SDC+all media. Cells were imaged at indicated time points by fluorescence microscopy. Individual slices are shown. Scale bar, 2 µm. **(D)** Quantification of the vacuole to plasma membrane fluorescence intensity ratio of Mup1 in C. The maximal fluorescence intensity of Mup1-GFP signal in the vacuolar lumen was divided by the maximal intensity of Mup1 at the plasma membrane. For each time point, cells (*n* ≥ 100) from three independent experiments were quantified in Fiji. Bar graphs represent the averages and error bars the SD from three experiments. P value ns, **<0.01, ***<0.001, using two-sided Student's *t* test. **(E)** Quantification of Mup1-GFP puncta per cell in C. For each time point, cells (*n* ≥ 100) from three independent experiments were quantified in Fiji. Bar graphs represent the averages and error bars the SD from three experiments. P value ns, **<0.01, using two-sided Student's *t* test. **(F)** Vacuole morphology of wild-type and *gyp7Δ* cells in growth and starvation conditions. Cells were grown in SDC+all and then shifted to SD-N for 2 h, where indicated (see Materials and methods). Cells were imaged by fluorescence microscopy and individual slices are shown. Dashed lines indicate yeast cell boundaries. Scale bar, 2 µm. **(G)** Quantification of the number of vacuoles per cell in F during growth. Cells were grouped into three different classes: 1–2 vacuoles, 3–4 vacuoles, and >5

vacuoles. Cells ($n \geq 150$) from three independent experiments were quantified in Fiji. Bar graphs represent the averages from three experiments and puncta represent the mean of each experiment. P value ns, using ANOVA one-way test. **(H)** Quantification of the number of vacuoles per cell in F during nitrogen starvation. Cells were grouped as described in G. Cells ($n \geq 150$) from three independent experiments were quantified in Fiji. Bar graphs represent the averages from three experiments and puncta represent the mean of each experiment. P value ns, using ANOVA one-way test.

Ypt7 inactivation and membrane extraction compared with Gyp7 on VMLs (Fig. 4, D and E), indicating that Gyp7 is highly specific for Ypt7. Together, we conclude that Gyp7 but not GDI depends on the right membrane composition for function.

To ask whether the membrane has additional functions beyond Gyp7 recruitment, we took advantage of the N-terminal His-tag of Gyp7 and generated liposomes containing the lipid dioleoyl [(N-(5-amino-1-carboxypentyl)iminodiacetic acid)succinyl] (DOGS-NTA), which can recruit His-tagged proteins to membranes (Cabrera et al., 2014). When present in liposomes containing just PC and PE, we now had sufficient Gyp7 on liposomes (Fig. 5, A and B), yet did not significantly recover activity of Gyp7 (Fig. 5, C and D). Importantly, DOGS-NTA had no negative impact on the Gyp7 GAP activity as Gyp7 shows comparable inactivation of Ypt7 on liposomes with the VML mixture lacking or containing DOGS-NTA (Fig. 4 B and Fig. 5 D). Together, our observations suggest that Gyp7 requires correct positioning and orientation on membranes, possibly by a distinct membrane environment, for full activity.

To identify the corresponding membrane-interacting region, we analyzed the Gyp7 model. According to the AlphaFold prediction (Fig. S3, C and D), Gyp7 has an N-terminal pleckstrin homology (PH) domain (Fidler et al., 2016), a connecting middle domain and the catalytic TBC domain toward the C-terminal (Fig. 5 E). The N-terminal PH domain with two positively charged patches and the middle domain with a potential amphipathic helix are possible Gyp7 regions involved in membrane binding. To search for a minimal membrane binding domain, we generated C-terminal truncations that contain just the predicted PH domain of Gyp7 (Fig. S3 E) and observed no binding to liposomes (Fig. S3, F and G). Likewise, the minimal GAP domain of just the TBC domain of Gyp7 (Fig. 5 E) had poor activity on membrane-bound Ypt7 compared with the full-length protein (Fig. 5, H and I), as it did not bind to membranes efficiently (Fig. 5, F and G), indicating that full-length Gyp7 is required for recognition and binding of membranes.

To ask whether the missing membrane recruitment causes the reduced GAP activity of the TBC domain toward membrane-bound Ypt7 or whether the membrane could have a direct activating effect on the GAP activity itself, we turned to a high performance liquid chromatography (HPLC)–based GAP assay. Here, the GTPase is constantly chemically reloaded with nucleotide due to the presence of EDTA and $MgCl_2$ (Araki et al., 2021; Eberth and Ahmadian, 2009) (Fig. 5 J and Fig. S3 H). This approach allowed us to directly compare the inactivation of soluble, not-prenylated Ypt7 by Gyp7 and the TBC domain in the absence or presence of liposomes (see Materials and methods), and thus determine the role of the Gyp7 membrane association for Ypt7 inactivation. By following the amount of GTP left in the reactions over time (0, 10, 60, 180, and 300 min), we determined the activity of our tested GAPs. In the absence of membranes,

Gyp7 showed GAP activity toward Ypt7 over time. In line with our previous findings, this activity was only slightly increased in the presence of PC/PE liposomes, but significantly enhanced in the presence of liposomes with the VML composition (Fig. 5 J). As expected, the presence of membranes did not affect the GAP activity of the TBC domain, as it did not bind membranes (Fig. S3 H). Importantly, only background GTP hydrolysis occurred in samples without Ypt7, without GAP, or neither Ypt7 nor GAP (Fig. S3 I). Together, our data indicate that direct membrane association increases Gyp7 activity for Ypt7. As the GAP domain should be available for Ypt7, our data suggest that full-length Gyp7 recognizes the membrane-bound Ypt7 possibly at additional sites prior to its binding of the GTPase domain.

## Gyp7 activity shifts Ypt7 localization from vacuoles to MVBs

Previous studies implied that high Gyp7 activity can remove Ypt7 from membranes if sufficient GDI is available (Cabrera and Ungermann, 2013). We also recently observed that the Ypt7 GEF Mon1–Ccz1 is hyperactive if the N-terminal part of Mon1 is truncated, i.e., Mon1$^{\Delta 100}$ (Borchers et al., 2023). Given that both Mon1–Ccz1 (Gao et al., 2018, 2022) and Gyp7 (as shown here) localize within the endolysosomal system, we wondered whether the levels or activity of the Ypt7 GEF and GAP could enhance endocytic trafficking as faster Ypt7 activation and turnover would be expected. We initially followed Ypt7 localization in strains lacking or overexpressing Gyp7 from the *TEF1* promoter (Fig. 6, A and B). In wild-type cells, Ypt7 localizes to the vacuolar rim and in puncta proximal to the vacuole (Fig. 6 A). As described, deletion of Gyp7 or Msb3 had no effect on Ypt7 localization, while the absence of both GAPs resulted in a slight, though significant decrease in the number of Ypt7 puncta (Fig. S4, A and B), indicating that other GAPs could take over the function of the main Ypt7 GAP Gyp7 upon its loss and under certain conditions. However, Gyp7 overexpression resulted in an increased number of Ypt7 puncta and a fraction of Ypt7 puncta not proximal to the vacuole anymore. We repeated this analysis in a strain expressing Mon1$^{\Delta 100}$. This strain also accumulates more Ypt7 puncta, suggesting enhanced early to late endosome transition (Borchers et al., 2023). Deletion of Gyp7 did not affect this phenotype. However, overexpression of Gyp7 in the Mon1$^{\Delta 100}$ strain resulted in the same accumulation of Ypt7 puncta that now shows increased fluorescence intensity and more Ypt7 puncta away from the vacuole (Fig. 6 B). This suggests that Gyp7 can relocate Ypt7 from vacuoles to endosomes. We thus wondered how the Rab, the GEF, and the GAP localize relative to each other (Fig. 6 C). In wild-type cells and in the Mon1$^{\Delta 100}$ strain, Gyp7 does not colocalize with Ccz1, while overproduction of Gyp7 results in strong colocalization (Fig. 6 D), suggesting that the Ypt7 GEF and GAP can indeed come together at the same endosomal compartment. However, these Gyp7-positive puncta did not colocalize with the Ypt7

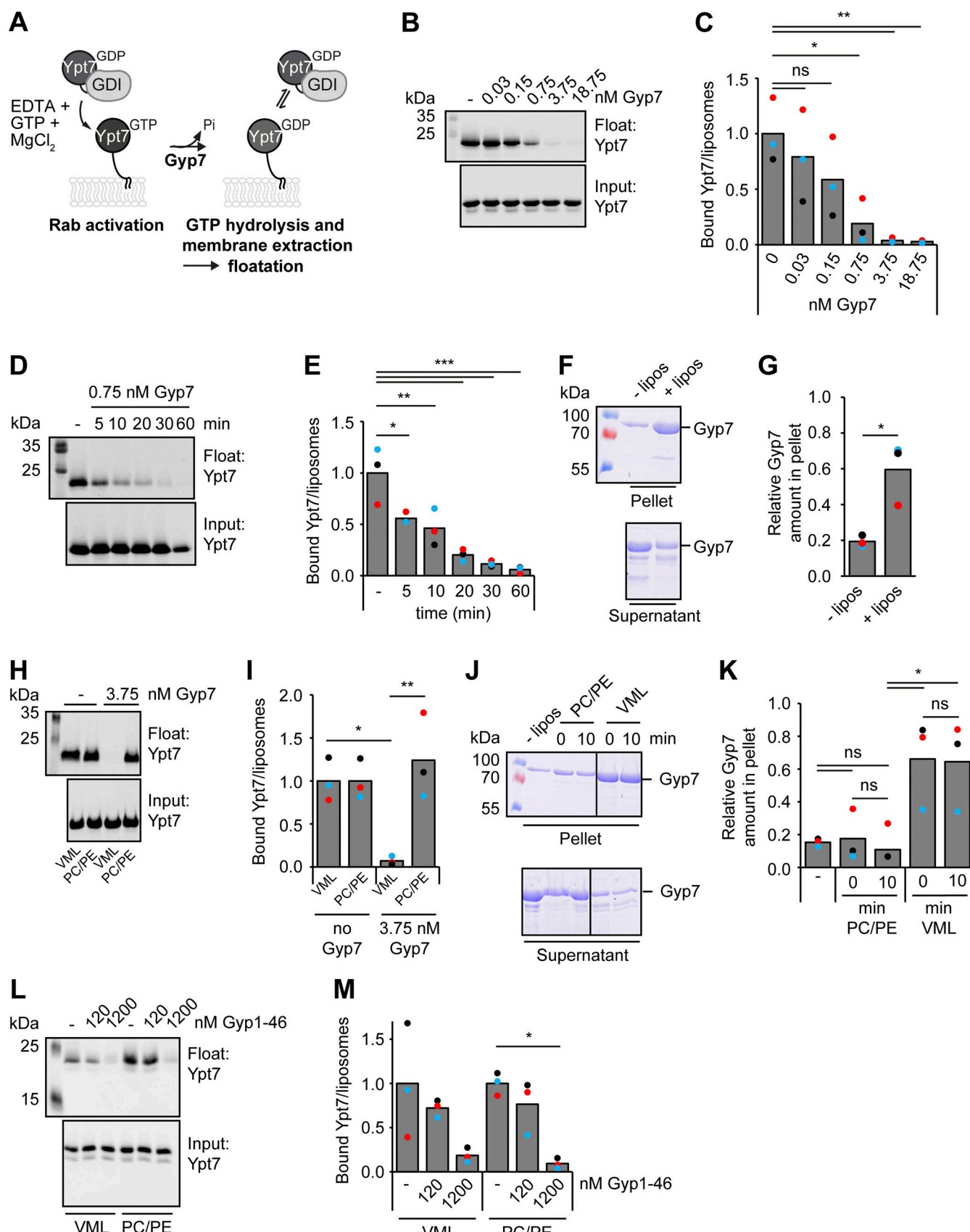

Figure 4. **Gyp7 requires a distinct membrane environment for efficient GAP activity. (A)** Overview of the GDI extraction assay. 250 μM liposomes with VML composition are preloaded with 0.6 μM Ypt7-GDI complex in the presence of 3.75 mM EDTA and 125 μM GTP. The nucleotide binding is stabilized by

addition of 7.5 mM MgCl$_2$. Incubation with the GAP Gyp7 triggers GTP hydrolysis. GDI extracts inactivated Ypt7 from liposomal membranes. Liposomes with bound Ypt7 are floated in a sucrose gradient and separated from unbound protein. Floated membrane fractions and inputs were analyzed by western blotting (see Materials and methods). **(B)** Ypt7 inactivation increases with the concentration of Gyp7. Assay was performed as in A. Reactions were incubated with different amounts of Gyp7 for 1 h. Control reaction contained no Gyp7. 40% of the float was analyzed together with 3% input by western blotting using an anti-Ypt7 antibody. **(C)** Quantification of bound Ypt7 to liposomes in B. Band intensity of Ypt7 signal in float was measured in Fiji and compared with input. Reactions containing Gyp7 were normalized to the average value of the control reaction. Bar graphs represent the averages from three independent experiments and puncta represent the mean of each experiment. P value ns, *<0.05, **<0.01, using ANOVA one-way test. **(D)** Kinetics of Gyp7 activity toward Ypt7-GTP. Assay was performed as in A. Reactions were incubated with 0.75 nM Gyp7 for different time points. Control reaction contained no Gyp7. 40% of the float was analyzed together with 3% input by western blotting using an anti-Ypt7 antibody. **(E)** Quantification of bound Ypt7 to liposomes in D. Quantification was performed as in C. P value *<0.05, **<0.01, ***<0.001 using ANOVA one-way test. **(F)** Membrane association of Gyp7. 715 µM liposomes with VML composition were incubated with 715 nM Gyp7 for 10 min. Membranes were separated from supernatant by centrifugation at 100,000 $g$ and both fractions were analyzed by SDS-PAGE and Coomassie staining. Control reaction contained no liposomes (see Materials and methods). **(G)** Quantification of the relative Gyp7 amount in the pellet in F. Band intensity of Gyp7 signal in the pellet was measured in Fiji and compared with Gyp7 signal in the supernatant. Bar graphs represent the averages from three independent experiments and puncta represent the mean of each experiment. P value *<0.05, using two-sided Student's $t$ test. **(H)** Comparison of Gyp7 activity on liposomes with VML composition and PC/PE liposomes. The assay was performed as in A. 3.75 nM Gyp7 was added to reactions containing liposomes with VML composition or PC/PE liposomes for 10 min. Control reactions contained respective liposomes and no Gyp7. 40% of the float was analyzed together with 3% input by western blotting using an anti-Ypt7 antibody. **(I)** Quantification of bound Ypt7 to liposomes in H. Quantification was performed as in C. Reactions containing Gyp7 were normalized to the average value of the respective control reaction. P value *<0.05, **<0.01, using ANOVA one-way test. **(J)** Association of Gyp7 with liposomes of VML composition and PC/PE liposomes. 715 nM Gyp7 was incubated with 715 µM liposomes for 0 and 10 min. Membrane association was analyzed as in F. **(K)** Quantification of the relative Gyp7 amount in the pellet in J. Quantification was performed as in G. P value ns, *<0.05, using ANOVA one-way test. **(L)** Comparison of Gyp1-46 activity on liposomes with VML composition and PC/PE liposomes. Assay was performed as in A, except for the addition of Gyp1-46 instead of Gyp7 to reactions. Reactions were incubated with different amounts of Gyp1-46 for 10 min. Control reactions contained respective liposomes and no GAP. 40% of the float was analyzed together with 3% input by western blotting using an anti-Ypt7 antibody. **(M)** Quantification of bound Ypt7 to liposomes in L. Quantification was performed as in C. Reactions containing Gyp1-46 were normalized to the average value of the respective control reaction. P value *<0.05, using ANOVA one-way test. Source data are available for this figure: SourceData F4.

---

puncta, even upon overproduction of the GAP (Fig. 6 E), suggesting that active Ypt7 resides in a different endosomal compartment population. Since Gyp7 is not present on vacuoles in any of our tested conditions, inactivation of Ypt7 might rather take place on endosomes, although we cannot exclude an additional role of Gyp7 at the vacuole or even elsewhere.

To determine the identity of the Ypt7 puncta under these conditions, we analyzed their colocalization with selected marker proteins. Ivy1 as a previously identified protein on signaling endosomes (Gao et al., 2022; Chen et al., 2021) strongly colocalized with Ypt7 puncta in Gyp7 overexpression strains, whereas colocalization with the retromer subunit Vps35 and the ESCRT protein Vps4 was mostly lost (Fig. 7, A and B; and Fig. S4 C). We did not detect colocalization with the Vps21 protein. We then analyzed Pep12 as a Q-SNARE of endosomes and observed that the number of Pep12 puncta was slightly reduced in the Mon1$^{\Delta100}$ strain and strongly reduced in the strain overexpressing Gyp7 (Fig. 7 C and Fig. S4 D). Moreover, several of these puncta were also more distant from the vacuole upon overproduction of Gyp7 and in combination with expression of Mon1$^{\Delta100}$ (Fig. 7 D and Fig. S4 D), similar to what we observed for Ypt7 puncta under the same conditions (Fig. 6, A and B). However, no change in the localization of Tco89 as a TORC1 subunit was detected (Fig. 7 E and Fig. S4 E). These data indicate that the Ypt7 puncta correspond to mature late endosomes, i.e., MVBs and/or signaling endosomes.

### Ypt7 confinement to late endosomes affects protein traffic in the endolysosomal system

To determine if the Ypt7 confinement due to Gyp7 overexpression affects transport toward the vacuole, we first analyzed the biosynthetic transport of carboxypeptidase 1 (Cps1) from the Golgi to the vacuole. In previous analyses, we observed

that this transport is strongly delayed when Vps21 and the CORVET subunit Vps8 are overproduced. This manipulation causes the arrest of endosomes with early endosomal markers, but not the vacuolar SNARE Vam3 or HOPS subunits, and results in the accumulation of Cps1 in puncta proximal to the vacuole (Fig. 8, A–C) (Markgraf et al., 2009). However, Ypt7 confinement by Gyp7 overproduction in cells expressing Mon1$^{\Delta100}$ resulted in similar localization of GFP-Cps1 as in wild-type cells (Fig. 8, A–C).

We next analyzed the endocytic pathway toward the vacuole by monitoring Mup1-GFP transport upon methionine addition (Lin et al., 2008). To analyze the effect of altered Gyp7 or Mon1–Ccz1 activity, we followed Mup1-GFP trafficking at early time points (5, 10 min) after methionine addition (Fig. 8, D and E). For each time point, we determined the ratio between the number of Mup1 puncta and the intensity of Mup1 signal in the plasma membrane. In strains overexpressing Gyp7, we observed a higher ratio at early time points of Mup1 uptake, while combining hyperactive Mon1–Ccz1 and overexpression of Gyp7 revealed the highest ratio (5, 10 min). In neither case, Mup1 was completely arrested on endosomes but arrived at the vacuole lumen after 60 min. Together, the data indicate a slight delay of endocytic transport due to overexpressing Gyp7, as expected for a regulator of Ypt7 activity such as a GAP. We thus conclude that the confinement of Ypt7 impairs but does not block transport pathways to the vacuole.

### Ypt7-positive structures correspond to MVBs

We previously showed that the formation of signaling endosomes as a subset of late endosomes requires both the ESCRT pathway and HOPS-mediated fusion of endosomes with vacuoles (Gao et al., 2022). One of the observations is that ESCRT and HOPS mutants are strongly impaired in TORC1 signaling

Figure 5. **Gyp7 is activated by a distinct membrane environment. (A)** Membrane association of Gyp7 with DOGS-NTA containing liposomes. 715 nM Gyp7 was incubated with 715 µM liposomes (VML + DOGS-NTA, PC/PE + DOGS-NTA, PC/PE) for 10 min. Membranes were separated from supernatant by centrifugation at 100,000 g and both fractions were analyzed by SDS-PAGE and Coomassie staining. Control reaction contained no liposomes. **(B)** Quantification of the relative Gyp7 amount in the pellet in A. Band intensity of Gyp7 signal in the pellet was measured in Fiji and compared with Gyp7 signal in the supernatant. Bar graphs represent the averages from three independent experiments and puncta represent the mean of each experiment. P value ns, **<0.01, ***<0.001, using ANOVA one-way test. **(C)** Comparison of Gyp7 activity on DOGS-NTA containing liposomes. 250 µM liposomes were preloaded with 0.6 µM Ypt7:GDI complex in the presence of 3.75 mM EDTA and 125 µM GTP. Nucleotide binding was stabilized by addition of 7.5 mM MgCl₂. Reactions were incubated with 3.75 µM Gyp7 for 10 min. Liposomes were floated in a sucrose gradient. Control reactions contained no Gyp7. 40% of the float was analyzed together with 3% input by western blotting using an anti-Ypt7 antibody. **(D)** Quantification of bound Ypt7 to liposomes in C. Band intensity of Ypt7 signal in float was measured in Fiji and compared to input. Reactions containing Gyp7 were normalized to the average value of the respective control reaction. Bar graphs represent the averages from three independent experiments and puncta represent the mean of each experiment. P value ns, ***<0.001, using ANOVA one-way test. **(E)** AlphaFold2 structure prediction of Gyp7. The N-terminal PH domain is colored blue and the C-terminal TBC domain is colored cyan with the catalytic Arg (R458) and Glu (Q531) residues shown red in stick representation. A middle domain, which is modeled with low predicted local distance difference test (pLDDT) confidence scores (Fig. S3, C and D), is colored green. **(F)** Membrane association of the TBC domain compared to full-length Gyp7. Gyp7 and the TBC domain were incubated with liposomes of VML composition as in A. Control reactions contained no liposomes. **(G)** Quantification of the relative amount of Gyp7 in the pellet in F. Quantification performed as in B. P value *<0.05, using ANOVA one-way test. **(H)** Comparison of Gyp7 and TBC domain activities on liposomes with VML

composition. Assay was performed as in C. Pre-loaded liposomes were incubated with different amounts of Gyp7 or the TBC domain for 10 min. **(I)** Quantification of bound Ypt7 to liposomes in H. Quantification was performed as in D. Reactions containing GAP were normalized to the average value of the control reaction. P value ns, *<0.05, using ANOVA one-way test. **(J)** Comparison of Gyp7 activity toward soluble Ypt7-GTP in solution and on membranes. 5 µM Ypt7 was incubated with 5 µM GAP and 50 µM GTP in the presence of 1 mM DTT, 20 mM EDTA, and 5 mM MgCl$_2$. Where indicated, reactions contained 1 mM liposomes with VML composition or PC/PE liposomes. Control reactions contained no Ypt7, no GAP, or neither Ypt7 nor GAP (see Fig. S3 I). Reactions were stopped after 0, 10, 60, 180, and 300 min by snap-freezing and boiling at 95°C. Samples were applied to a HPLC system and the absorbance of GDP and GTP was monitored at 254 nm. Peaks were analyzed with OpenChrom and for each time point the percentage of GDP and GTP in the samples was determined. The percentage of GTP left at each time point was normalized to the respective percentage of GTP at $t$ = 0 min. Normalized % GTP left plotted against the time in min. Bar graphs represent the averages and error bars the SD from three independent experiments. P value **<0.01, ***<0.001, using ANOVA one-way test. Source data are available for this figure: SourceData F5.

(Gao et al., 2022; Zurita-Martinez et al., 2007). We therefore analyzed TORC1 activity in Gyp7 overexpressing strains. When grown on rapamycin to inhibit TORC1, cells lacking Gyp7 were clearly sensitive to this drug (Fig. 3 B and Fig. S2 A). In contrast, Gyp7-overproducing cells became slightly resistant to rapamycin, suggesting a likely higher TORC1 activity (Fig. 9 A). This effect was also modestly enhanced in the presence of Mon1$^{\Delta 100}$. To resolve which pool of TORC1 activity is mostly affected, we employed an established reporter assay, where the TORC1 target Sch9 localizes either to endosomes (endosomal TORC1, ET) or to the vacuole (vacuolar TORC1, VT) (Fig. S5, A and B). ET and VT activity were then analyzed by monitoring the Sch9 phosphorylation on the ET or VT reporter using a phospho-specific antibody to the TORC1 target site on Sch9 (Hatakeyama et al., 2019). Importantly, we observed a clear decrease in VT activity in the *gyp7Δ* mutant, whereas ET activity was increased in the Gyp7 overproduction strain. The observations were less clear when overexpression of Gyp7 was combined with the Mon1$^{\Delta 100}$ mutant. This may be due to the Mon1$^{\Delta 100}$ allele causing a trafficking defect of the ET and VT probes as the endosomal system is perturbed. All in all, we conclude that Gyp7-mediated confinement of Ypt7 to puncta next to the vacuole results in higher endosomal TORC1 activity.

We next asked whether the Gyp7-induced dot-like Ypt7 would accumulate in strains impaired in the ESCRT pathway, where the Class E compartment is found proximal to the vacuole. When Vps4 was deleted, mNeon-Ypt7 strongly accumulated in puncta proximal to the vacuole (Fig. 9 B, top). This accumulation was likewise seen in the strain overproducing Gyp7 (Fig. 9 B, bottom). Importantly, the endocytosed lipophilic dye FM4-64 also accumulated in these Ypt7-positive structures. This was not observed if Vps4 was present (Fig. 6 A), indicating that the Ypt7-enriched endosomes allow efficient FM4-64 transport to the vacuole. The puncta localization of Ypt7 in *vps4Δ* cells is similar to previous findings, in which wild-type Ypt7 was overproduced in *vps4Δ* cells (Balderhaar et al., 2010). We thus concluded that Ypt7 puncta persist downstream of the formation of MVBs by ESCRTs.

All previous data suggest that Ypt7 is prominently present on MVBs, which accumulate upon overproduction of Gyp7 in our fluorescence microscopy data. We wondered whether we could also observe an accumulation of MVBs in the mNeon-Ypt7 expressing strains by electron microscopy (Fig. 9 C). In wild-type cells, single MVBs are occasionally found next to the vacuole. In the Mon1$^{\Delta 100}$ Gyp7 overproduction mutant, we detected MVBs with higher frequency throughout the cell sections and often

organized in a cluster of two to three late endosomes, in line with the accumulation of Ypt7 puncta in this mutant. We then wondered if these structures may indeed carry Ypt7. Since the signal of endogenous Ypt7 is not sufficient for immunoelectron microscopy (IEM), we overproduced GFP-tagged Ypt7 in a wild-type background, which may mirror the endosomal effect of Ypt7 confinement by Gyp7 (Balderhaar et al., 2010). We analyzed the localization of overproduced GFP-tagged Ypt7 with nano-scale resolution in these cells by IEM. Immunogold labeling of sections with an anti-GFP antibody revealed that Ypt7 was distributed on the vacuole membrane and even more prominently on multiple MVBs, which accumulated proximal to vacuoles (Fig. 9 D). We thus conclude that Ypt7 functions on MVBs, which in part correspond to signaling endosomes. As Gyp7 can strongly confine Ypt7 proximal to the vacuole, we speculate that Gyp7 is a regulator of Ypt7 function at signaling endosomes.

## Discussion

Within this study, we uncovered that the Ypt7-specific GAP Gyp7 localizes to puncta that correspond to compartments of the endosomal system, where it is needed for normal endolysosomal transport. In the absence of Gyp7, cells become sensitive to endolysosomal stresses and TORC1 inhibition. In vitro, Gyp7 membrane association and activity are strongly regulated by the membrane environment. Surprisingly, Gyp7 overproduction does not liberate Ypt7 from endosomes but rather confines it to a subpopulation proximal to the vacuole. This effect is even stronger in a strain also having hyperactive Ypt7 GEF due to the expression of the Mon1$^{\Delta 100}$–Ccz1 mutant complex. Under these conditions, cells become moderately resistant to TORC1 inhibition. This subpopulation of Ypt7-positive endosomes requires ESCRTs for their formation, yet lack Vps4, suggesting that they correspond to mature late endosomes/MVBs and are in part equivalent to signaling endosomes (Chen et al., 2021; Gao et al., 2022; Hatakeyama et al., 2019). Our data strongly suggests that Gyp7 regulates the function of these compartments.

Gyp7 is the Ypt7-specific GAP (Brett et al., 2008; Vollmer et al., 1999; Lachmann et al., 2012; Eitzen et al., 2000). However, deletion of Gyp7 has little effect on Ypt7 function, and vacuoles fragment only upon strong overexpression (Vollmer et al., 1999; Brett et al., 2008; Eitzen et al., 2000). We confirmed these findings and further showed that mistargeting of endogenous Gyp7 to the vacuole membrane resulted in the same vacuole fragmentation phenotype. We can now explain the relatively minor effects of Gyp7 deletion on vacuole morphology

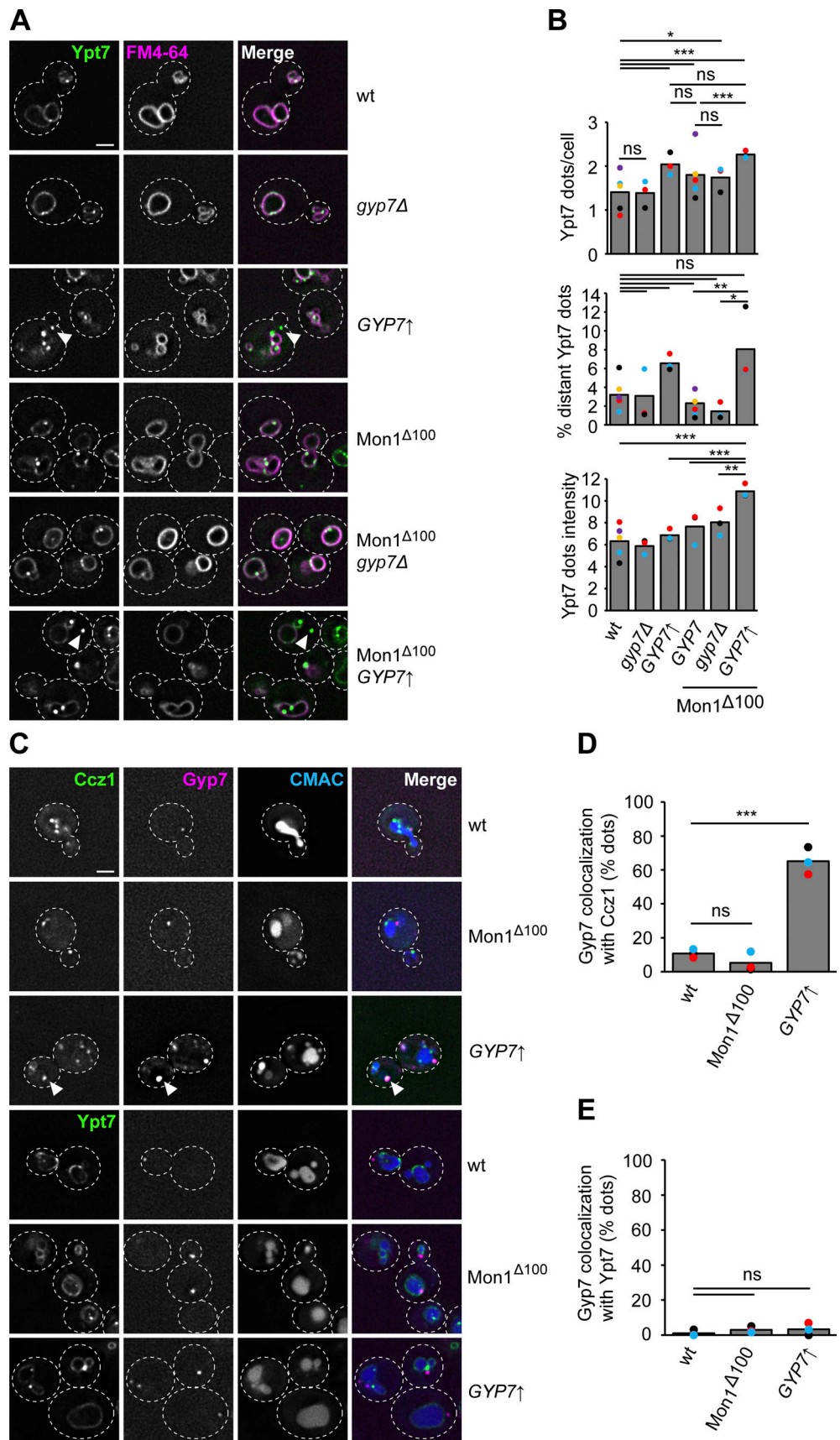

Figure 6. **Gyp7 and Mon1–Ccz1 shift Ypt7 from the vacuole to dot-like structures. (A)** The localization of Ypt7 depends on the expression level or activity of Gyp7 and Mon1–Ccz1. Endogenous mNeon-Ypt7 was expressed from an integrative plasmid in *ypt7Δ* cells. Where indicated, 100 amino acids at the

N-terminus of Mon1 were deleted (Mon1$^{\Delta100}$). Gyp7 was either deleted or expressed from the *TEF1* promoter in mNeon-Ypt7 expressing cells with wild-type (wt) Mon1 or Mon1$^{\Delta100}$. Vacuolar membranes were stained with FM4-64. Cells were imaged by fluorescence microscopy. Individual slices are shown. Arrowheads depict Ypt7 accumulations not proximal to the vacuole. Dashed lines indicate yeast cell boundaries. Scale bar, 2 µm. **(B)** Quantification of the total number of Ypt7 puncta per cell, the percentage of distant Ypt7 puncta, and the fluorescence intensity of Ypt7 puncta in A. The number of distant Ypt7 puncta (not at the vacuole) was divided by the total number of Ypt7 puncta per cell. The maximum fluorescence intensity of mNeon-Ypt7 puncta was normalized to the maximum fluorescence intensity of mNeon-Ypt7 at the vacuolar membrane. Cells ($n \geq 100$) from three independent experiments were quantified in Fiji. Bar graphs represent the averages from three experiments and puncta represent the mean of each experiment. P value ns, *<0.05, **<0.01, ***<0.001, using ANOVA one-way test. **(C)** Localization of Gyp7 relative to Ypt7 and Mon1–Ccz1. Gyp7 was C-terminally tagged with 2xmKate in the Mon1$^{100}$ strain, in *TEF1*pr-*GYP7* or wild-type cells encoding endogenous Ccz1-mNeon (top) or mNeon-Ypt7 (bottom). Vacuoles were stained with CMAC. Cells were imaged by fluorescence microscopy. Individual slices are shown. Arrowheads depict representative colocalization. Dashed lines indicate yeast cell boundaries. Scale bar, 2 µm. **(D)** Quantification of Gyp7 puncta colocalizing with Ccz1 puncta in C. Cells ($n \geq 100$) from three independent experiments were quantified in Fiji. Bar graphs represent the averages from three experiments and puncta represent the mean of each experiment. P value ns, ***<0.001, using ANOVA one-way test. **(E)** Quantification of Gyp7 puncta colocalizing with Ypt7 puncta in C. Cells ($n \geq 100$) from three independent experiments were quantified in Fiji. Bar graphs represent the averages from three experiments and puncta represent the mean of each experiment. P value ns, using ANOVA one-way test.

as Gyp7 localizes to puncta proximal to the vacuole, presumably endosomes, and accumulates in late endosomes upon ESCRT deletion. In this regard, Gyp7 seems to function like mammalian TBC1D5 as a retromer-associated Rab7 GAP (Kvainickas et al., 2019; Jimenez-Orgaz et al., 2018). However, deletions of proteins involved in retrograde transport from endosomes did not completely abolish Gyp7 localization in puncta proximal to the vacuole. Only upon deletion of both Rab5-specific GAPs, Vps9, and Muk1, or other endosomal fusion proteins, Gyp7 relocalized to multiple puncta (Fig. S1 A). How Gyp7 is targeted to these structures, apart from binding to Ypt7, remains an open question at this point. It is, however, possible that Gyp7 binds specifically to endosomal membranes as artificial targeting of Gyp7 to more rigid membranes was not sufficient for its full activation in vitro (Fig. 5 D).

Our analysis of Gyp7 uncovered a striking link between Ypt7 cycling and the formation of both mature late endosomes/MVBs and signaling endosomes. We previously showed that a subpopulation of endosomes harbors the TORC1 complex, which is otherwise found on vacuoles (Hatakeyama et al., 2019). These endosomes were thus named signaling endosomes. At this location, TORC1 phosphorylates the Fab1 complex and presumably modulates its activity (Chen et al., 2021). Additional factors involved in the biogenesis of the signaling endosomes are the HOPS and ESCRT complexes (Gao et al., 2022). Here, we discovered that enhanced Ypt7 cycling by Gyp7 overproduction and a hyperactive Mon1–Ccz1 complex confines Ypt7 to late endosomes. We postulate that these structures mature from Vps21-positive into Ypt7-positive late endosomes, a transition culminating with the loss of the ESCRT machinery (Fig. 9 E). Even though MVBs may look phenotypically similar if arrested early by overproducing Vps21 or Vps8 (Markgraf et al., 2009), or late by overproducing Ypt7 (Fig. 9 D), they differ in their surface composition based on our analysis presented here. We therefore believe that the late, Ypt7-positive endosomes correspond in part to signaling endosomes as they are (i) positive for the specific marker protein Ivy1 (Numrich et al., 2015; Varlakhanova et al., 2018; Malia et al., 2018), (ii) contain the late endosomal SNARE Pep12, (iii) lack the ESCRT protein Vps4, (iv) require the ESCRT machinery for their formation, and (v) regulate endosomal TORC1 activity. As they are also reduced in their Vps21 content, these structures are likely matured Ypt7-positive MVBs as also suggested by our ultrastructural analysis of cells overproducing Ypt7 (Fig. 9, C and D).

Why have these structures been overlooked? Ypt7 has been previously found in puncta proximal to the vacuole (Arlt et al., 2015; Balderhaar et al., 2010; Shimamura et al., 2019), which we interpreted as a minor pool or a vacuolar domain. However, this may have been a misconception. As both Mon1–Ccz1 (Gao et al., 2018) and Gyp7 (as shown here) are only found within the endosomal system and not on the vacuole, Ypt7 activation and cycling seem to be largely confined to late endosomes. By enhancing the Ypt7 cycle, we have been able to trap Ypt7 at the late endosomes, which thereby greatly facilitates its examination by fluorescence microscopy. This has allowed us now to separate Vps21- and ESCRT-positive endosomes, and thus still immature MVBs, from Ypt7-positive late endosomes, which may include signaling endosomes. Moreover, this interpretation of a maturing MVB would also explain the persistence of a prevacuolar compartment proximal to the vacuole (Casler and Glick, 2020; Raymond et al., 1992; Prescianotto-Baschong and Riezman, 2002; Bryant et al., 1998; Gerrard et al., 2000; Singer and Riezman, 1990; Vida et al., 1990; Day et al., 2018; Griffith and Reggiori, 2009). Here, maturation of Vps21 to Ypt7 positive endosomes is paralleled by signaling via the TORC1 complex, which may delay fusion of MVBs. Likewise, the recycling of proteins from MVBs via the retromer and other retrograde transport systems as well as a change in lipid composition such as PI(3)P or PI(3,5)P$_2$ may delay the fusion of late MVBs (Laidlaw et al., 2022; Suzuki et al., 2021; Chi et al., 2014; Liu et al., 2012). It is also likely that even this late Ypt7-positive MVB population is not homogenous as endocytic transport of selected cargos to the vacuole occurs rather efficiently (Day et al., 2018; Casler and Glick, 2020). However, we do not yet understand how this transition is controlled. We expect that both the Ypt7 GEF and GAP, i.e., Mon1–Ccz1 and Gyp7, are regulated in their activity as both Mon1–Ccz1 (Langemeyer et al., 2020) and Gyp7 (as shown here) also colocalize with Vps21-positive early endosomal compartments.

Our data further suggest that Gyp7 also regulates TORC1 function via Ypt7 as cells with more Ypt7-positive structures due to Gyp7 overexpression have higher endosomal TORC1 activity, whereas *gyp7Δ* cells have reduced vacuolar TORC1 activity. In this regard, our findings agree with observations in mammalian cells, in which the inactivation of TBC1D5 resulted in hyperactive Rab7, a mixing of Rab5 and Rab7 compartments, and a strong defect in mTORC1 signaling (Kvainickas et al., 2019). Furthermore, enhanced endosomal TORC1 signaling in Gyp7

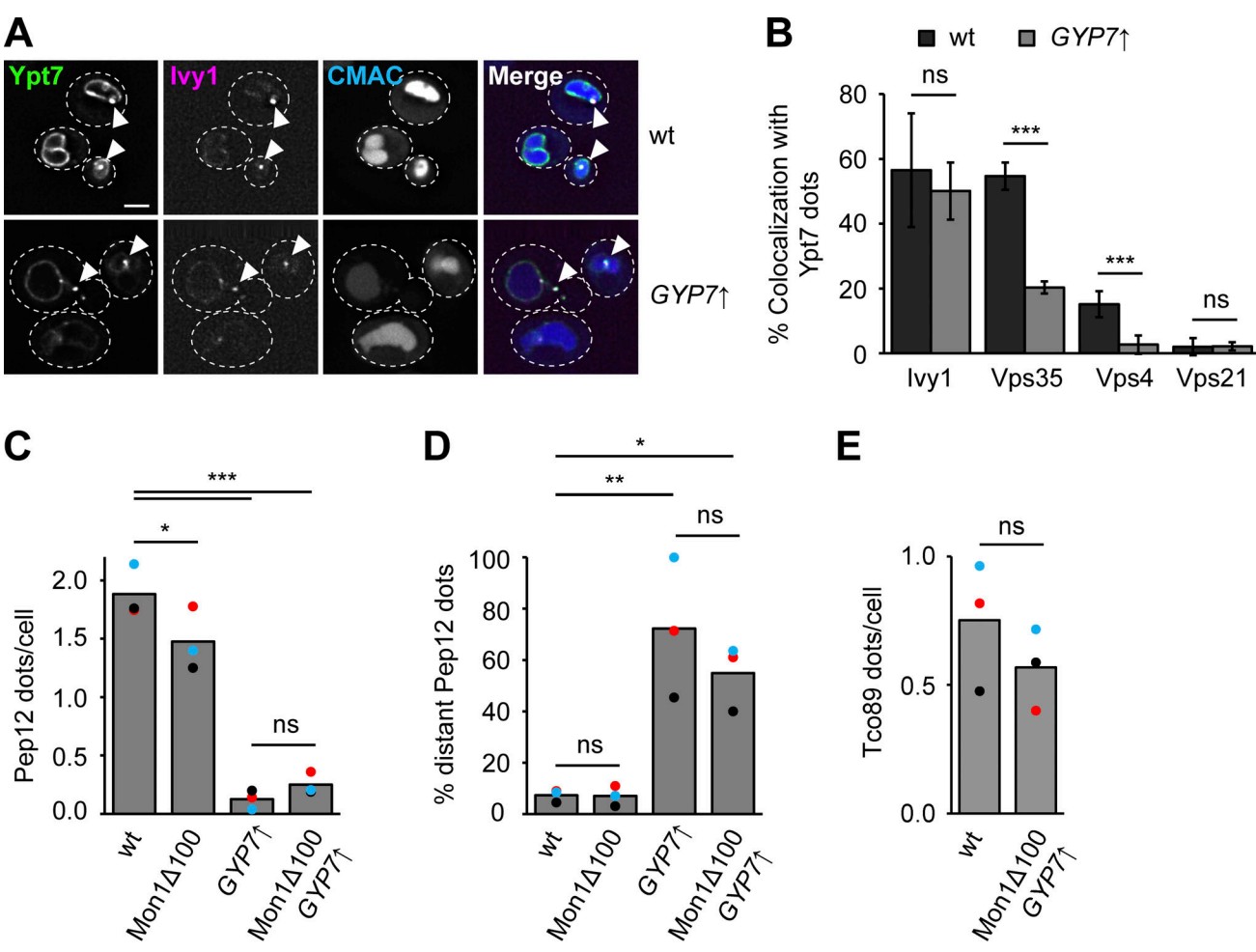

Figure 7. **Ypt7-positive puncta correspond to signaling endosomes. (A)** Localization of mNeon-Ypt7 puncta relative to the endosomal marker Ivy1. Ivy1-mKate was expressed in *TEF1pr-GYP7* or wild-type (wt) cells encoding endogenous mNeon-Ypt7. Vacuoles were stained with CMAC. Cells were imaged by fluorescence microscopy. Individual slices are shown. Arrowheads depict representative colocalization. Dashed lines indicate yeast cell boundaries. Scale bar, 2 µm. **(B)** Quantification of Ypt7 colocalizing with endosomal markers in A and Fig. S4 C. Cells ($n \geq 100$) from three independent experiments were quantified in Fiji. Bar graphs represent the averages and error bars the SD from the three experiments. P value ns, ***<0.001, using two-sided Student's *t* test. **(C)** Quantification of the number of Pep12 puncta per cell in Fig. S4 D. Cells ($n \geq 150$) from three independent experiments were quantified in Fiji. Bar graphs represent the averages from three experiments and puncta represent the mean of each experiment. P value *<0.05, ***<0.001, using ANOVA one-way test. **(D)** Quantification of the percentage of distant Pep12 puncta in Fig. S4 D. The number of distant Pep12 puncta (not at the vacuole) was divided by the total number of Pep12 puncta per cell. Cells ($n \geq 150$) from three independent experiments were quantified in Fiji. Bar graphs represent the averages from three experiments and puncta represent the mean of each experiment. P value *<0.05, **<0.01, using ANOVA one-way test. **(E)** Quantification of the number of Tco89 puncta per cell in Fig. S4 E. Cells ($n \geq 150$) from three independent experiments were quantified in Fiji. Bar graphs represent the averages from three experiments and puncta represent the mean of each experiment. P value ns, using two-sided Student's *t* test.

overexpression mutants suggests that the identity and possible fusion of signaling endosomes with the vacuole is tightly regulated. This may occur by phosphorylation events like the one of the Fab1 complex (Chen et al., 2021). Other possible targets are the Mon1–Ccz1 complex and Gyp7, whose activities clearly change signaling and late endosome biogenesis (Borchers et al., 2023) (this study). Likewise, HOPS complex activity might also be regulated. We also believe that signaling endosomes form after ESCRTs finish the formation of intraluminal vesicles. This could explain why several *VPS* mutants, including those belonging to Class E, have a TORC1 signaling defect (Gao et al., 2022; Kingsbury et al., 2014). Finally, it is possible that Ypt7 effectors like retromer, Ivy1, and the HOPS complex, compete for the available Ypt7-pool. Further analysis of Gyp7 as a key

regulator will be required to clarify how Ypt7 functions, thus signaling at the late endosome is controlled.

## Materials and methods

### Strains and plasmids
Strains used in this study are listed in Table S1. A PCR- and homologous recombination-based approach with corresponding primers and templates was used to delete or endogenously tag genes (Janke et al., 2004). Plasmids used in this study are listed in Table S2.

### Endogenous mutagenesis by CRISPR/Cas9
CRISPR/Cas9 was used to generate genomic point mutations in yeast strains (Generoso et al., 2016). Therefore, a Cas9-

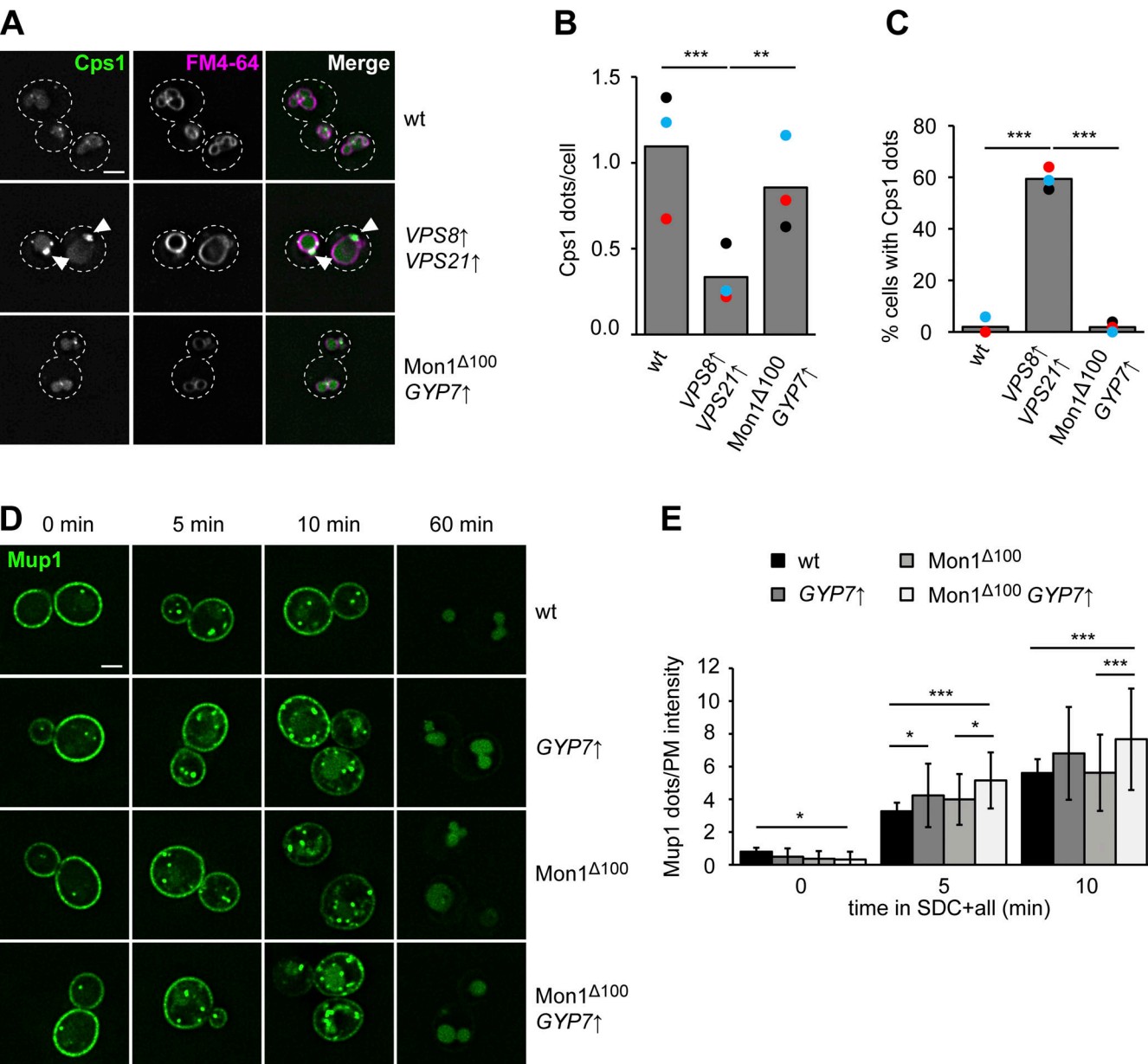

Figure 8. **Enhanced Ypt7 cycling affects endocytic trafficking. (A)** Localization of Cps1 in wild-type (wt), *TEF1*pr-*VPS8 ADH*pr-*VPS21*, and Mon1$^{\Delta 100}$-Ccz1 *TEF1*pr-*GYP7* cells. Vacuolar membranes were stained with FM4-64. Cells were imaged by fluorescence microscopy. Individual slices are shown. Arrowheads depict Cps1 accumulations next to the vacuole. Dashed lines indicate yeast cell boundaries. Scale bar, 2 μm. **(B)** Quantification of the number of Cps1 puncta per cell in A. Cells ($n \geq 140$) from three independent experiments were quantified in Fiji. Bar graphs represent the averages from three experiments and puncta represent the mean of each experiment. P value **<0.01, ***<0.001, using ANOVA one-way test. **(C)** Quantification of the percentage of cells with Cps1 accumulations in A. The number of cells with Cps1 accumulations at the vacuole was divided by the total number of cells. Cells ($n \geq 140$) from three independent experiments were quantified in Fiji. Bar graphs represent the averages from three experiments and puncta represent the mean of each experiment. P value ***<0.001, using ANOVA one-way test. **(D)** Endocytosis of Mup1 in cells with altered expression or activity of Gyp7 and Mon1–Ccz1. Cells were grown to logarithmic phase in SDC-MET media, analyzed by fluorescence microscopy, and then shifted to SDC+all media. Cells were imaged at indicated time points by fluorescence microscopy. Individual slices are shown. Scale bar, 2 μm. This is the same assay shown in Fig. 3 C with different mutants and time points analyzed. **(E)** Quantification of the number of puncta to plasma membrane fluorescence intensity of Mup1 ratio in D. For each cell, the number of Mup1 puncta was divided by the maximum fluorescence intensity of Mup1-GFP signal at the plasma membrane (PM). For each time point, cells ($n \geq 100$) from three independent experiments were quantified in Fiji. Bar graphs represent the averages and error bars the SD from three experiments. P value *<0.05, ***<0.001, using ANOVA one-way test.

containing plasmid was built with a specific gRNA through the Gibson assembly strategy. The plasmid was transformed together with the corresponding homology-directed repair fragment (Table

S2). Cells were recovered in yeast extract peptone dextrose (YPD) at 30°C for 2 h and then plated on the corresponding selection plate. Positive clones were selected by sequencing.

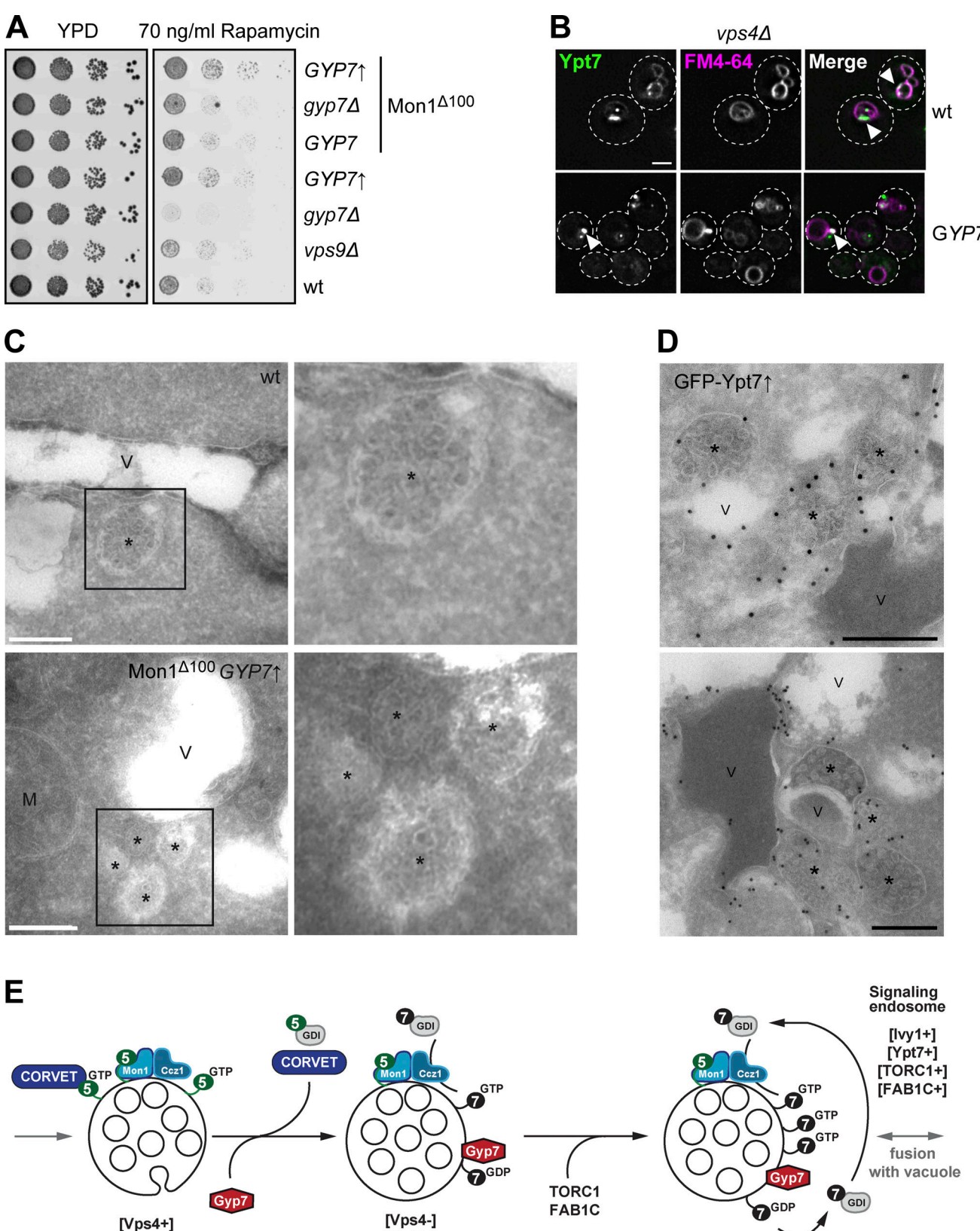

**Figure 9.** **Ypt7 functions on mature endosomes. (A)** Growth assay on rapamycin-containing plates. Indicated yeast strains were grown to the same OD$_{600}$ in YPD media and serial dilutions were spotted onto agar plates containing YPD or YPD supplemented with 70 ng/ml rapamycin. Plates were incubated at 37°C for several days before imaging. Images are representative of three independent experiments. **(B)** Ypt7 accumulates in the Class E compartment. Endogenous mNeon-Ypt7 was expressed from an integrative plasmid in *ypt7Δ vps4Δ* cells. Where indicated, Gyp7 was expressed from the *TEF1* promoter. Vacuolar membranes were stained with FM4-64. Cells were imaged by fluorescence microscopy. Individual slices are shown. Arrowheads depict Ypt7 accumulations in

the Class E compartment. Dashed lines indicate yeast cell boundaries. Scale bar, 2 µm. **(C)** Electron microscopy analysis of cells expressing mNeon-Ypt7 in wild-type (wt) and Mon1$^{\Delta100}$-Ccz1 *TEF1pr-GYP7* cells (see Materials and methods). M, mitochondria; V, vacuole; asterisk, multivesicular body. Scale bars, 200 nm. **(D)** IEM analysis of cells expressing *TEF1pr-GFP-YPT7*. Ypt7 was detected by using anti-GFP antibodies and protein A–conjugated gold (see Materials and methods). Asterisk, multivesicular body; V, vacuole. Scale bars, 200 nm. **(E)** Working model of Gyp7 function on MVBs. MVBs form with the help of ESCRTs on Vps21/Rab5-positive endosomes (left), which carry yet inactive Mon1–Ccz1. Maturation of endosomes includes recruitment of Gyp7 and loss of Rab5 and its effector CORVET. Some of these late endosomes also acquire TORC1 and the Fab1 complex, thus turn into signaling endosomes. This may affect Gyp7 and Mon1–Ccz1 activity and thus control the available Ypt7 pool.

### Expression and purification of proteins from *Escherichia coli*

GST-TEV-Ypt7, Ypt7-His$_6$, His$_6$-TEV-Gyp7, His$_6$-Sumo-Gyp7 TBC, Gyp1-46-His$_6$ and the prenylation machinery, Mrs6-His6, GST-PreSc-GDI, and pCDF-DUET-Bet4 His$_6$-TEV-Bet2 were expressed in *Escherichia coli* BL21 DE3 (Rosetta) cells. Cells were grown in the presence of the corresponding antibiotics at 37°C in Luria Broth medium until an OD$_{600}$ = 0.6, before protein expression was induced by the addition of 0.25 mM (or 0.5 mM for His$_6$-TEV-Gyp7, His$_6$-Sumo-Gyp7 TBC, and Gyp1-46-His$_6$) isopropyl-β-d-thiogalactoside. After 16–18 h of protein expression at 16°C, cells were harvested by centrifugation at 4,000 *g*, 4°C for 10 min. Cells were resuspended in buffer containing 50 mM HEPES, pH 7.5, 150 mM NaCl, 1 mM MgCl$_2$, 1 mM DTT (GST-TEV-Ypt7, Ypt7-His$_6$, Gyp1-46-His$_6$) or buffer containing 20 mM Na$_2$HPO$_4$/NaH$_2$PO$_4$, pH 7.4, 500 mM NaCl (His$_6$-TEV-Gyp7, His$_6$-Sumo-Gyp7 TBC). Cells expressing GST-PreSc-GDI were resuspended in PBS containing 5 mM β-mercaptoethanol (β-MeOH), while cells expressing the other components of the prenylation machinery were resuspended in buffer containing 50 mM Tris-HCl, pH 8.0, 300 mM NaCl, and 2 mM β-MeOH. During lysis, buffers were supplemented with 1 mM phenylmethylsulfonyl fluoride and 0.1× protease inhibitor cocktail (a 20× stock solution contained 2 µg/ml Leupeptin, 10 mM 1,10-Phenanthroline, 10 µg/ml Pepstatin A, and 2 mM Pefablock). Cell lysis was performed in a Microfluidizer (Microfluidics, Inc.) and the cell lysate was cleared during centrifugation at 40,000 *g*, 4°C for 30 min. The cleared lysate was incubated with nickel-nitriloacetic acid agarose (Qiagen) for purification of His-fusion proteins (Ypt7-His$_6$, His$_6$-TEV-Gyp7, His$_6$-Sumo-Gyp7 TBC, Mrs6-His$_6$, and Bet4 His$_6$-TEV-Bet2) or with glutathione sepharose fast-flow beads (GE Healthcare) for GST-fusion proteins (GST-TEV-Ypt7, GST-PreSc-GDI). After incubation for 2 h, 4°C on a turning wheel, and extensive washing of the beads, His-fusion proteins were eluted from the beads with the respective buffer containing 300 mM imidazole. GST-fusion proteins were cleaved from the beads during incubation with TEV protease (GST-TEV-Ypt7) or PreScission protease (GST-PreSc-GDI) for 2 h at 16°C on a turning wheel. His$_6$-TEV-Ypt7, His$_6$-Mrs6, and Bet4 His$_6$-TEV-Bet2 were dialyzed into buffer containing 50 mM HEPES-NaOH, pH 7.4, 150 mM NaCl, 1.5 mM MgCl$_2$, and 1 mM DTT overnight with one buffer exchange. The buffer of purified GDI, His$_6$-TEV-Gyp7, His$_6$-Sumo-Gyp7 TBC, and Gyp1-46-His$_6$ was exchanged using a PD-10 desalting column (GE Healthcare). Proteins were snap-frozen and stored in aliquots at –80°C.

### In vitro prenylation of Rab GTPases

Prenylated Rab-GDI complexes were generated as previously described (Thomas and Fromme, 2016; Langemeyer et al., 2020). First, 40 µM Rab GTPase was preloaded with 200 µM GDP (Sigma-Aldrich) in the presence of 20 mM EDTA, pH 8.0, for 30 min at 30°C. The reaction was filled up with prenylation buffer containing 50 mM HEPES-NaOH, pH 7.4, 150 mM NaCl, 1.5 mM MgCl$_2$, and 1 mM DTT and afterward stopped by addition of 25 mM MgCl$_2$. Excess EDTA and MgCl$_2$ were removed by buffer exchange into prenylation buffer in a Zeba spin column with 10 kDa molecular weight cutoff (Thermo Fisher Scientific). The prenylation reaction was done at 37°C for 1.5 h. For this, the nucleotide-loaded Rab-GTPase, GDI, His$_6$-Mrs6, and Bet4 His$_6$-TEV-Bet2 were incubated at a 10:9:1:1 M ratio with a sixfold molar excess of geranylgeranyl pyrophosphate (Sigma-Aldrich) in prenylation buffer. Afterward, 10 mM imidazole was added to the reaction and a 0.1-fold volume of Ni$^{2+}$ resin was added to remove His$_6$-Mrs6 and Bet4 His$_6$-TEV-Bet2 from the reaction. After 1 h at 4°C, the resin was removed by centrifugation for 3 min at 500 *g* at 4°C. Then the supernatant was loaded to a Superdex200 increase column (GE Healthcare). Fractions containing prenylated Rab-GDI complexes at a stoichiometric ratio were pooled and used as a substrate in GDI extraction assays.

### Preparation of liposomes

Lipids were purchased from Avanti Polar Lipids, Inc., except for ergosterol (Sigma-Aldrich) and 1,1′-dioctadecyl-3,3,3′,3′-tetramethylindodicarbocyanine (DiD; Life Technologies). Liposomes composed of the vacuolar mimicking lipid mix (Zick and Wickner, 2014) or containing 81.5 mol % DLPC (18:2 18:2), 18 mol % DLPE (18:2 18:2), and 0.5 mol % DiD were prepared. The vacuolar mimicking lipid mix contained 47.1 mol % DLPC (18:2 18:2), 18 mol % DLPE (18:2 18:2), 18 mol % soy PI, 1 mol % dipalmitoyl phosphatidylinositol-3-phosphate (PI(3)P diC16), 4.4 mol % dilinoleoyl phosphatidylserine (18:2 18:2), 2 mol % dilinoleoyl phosphatidic acid (18:2 18:2), 8 mol % ergosterol, 1 mol % diacylglycerol (16:0 16:0), and 0.5 mol % DiD. Where indicated, liposomes contained 3 mol % DOGS-NTA (18:1 18:1) and 3 mol % less DLPC. Lipid films were evaporated and either dissolved in a buffer containing 50 mM HEPES-NaOH, pH 7.4, 150 mM NaCl, and 1.5 mM MgCl$_2$ (membrane association assay) or 50 mM HEPES-NaOH, pH 7.4, 150 mM NaCl (HPLC-based GTPase activity assay) or HEPES-NaOH, pH 7.4, 150 mM KOAc, and 2 mM MgCl$_2$ (GDI extraction assay). After five cycles of thawing and freezing in liquid nitrogen, liposomes were extruded to 100 nm using a hand extruder and polycarbonate filters (Avanti Polar Lipids, Inc.).

### Membrane association assay

Membrane association of GTPase activating proteins was analyzed by incubation of 715 µM liposomes with 715 nM protein for 10 or, where indicated, 0 min at 27°C, followed by centrifugation for 45 min, 100,000 *g* at 4°C. Reactions were filled up with a

buffer containing 50 mM HEPES-NaOH, pH 7.4, 150 mM NaCl, and 1.5 mM $MgCl_2$ to a volume of 80 µl. Prior to incubation, proteins were centrifuged for 1 h, 100,000 $g$ at 4°C. Pelleted liposomes were separated from the supernatant. Proteins in the supernatant were precipitated by addition of 13% trichloroacetic acid (TCA). Upon washing with 100% ice-cold acetone, supernatant and pellet fractions were analyzed by SDS-PAGE. Band intensity was measured by Fiji (NIH). To determine the percentage of GAPs bound to membranes, the intensity signal of GAP in the pellet was normalized to the intensity signal in the corresponding supernatant.

## GDI extraction assay

The GTPase activities of GAPs on membranes were analyzed in a GDI extraction assay according to Thomas et al. (2021) with modifications. For activation of prenylated Ypt7 on membranes, 0.6 µM Ypt7-GDI complex was incubated with 250 µM liposomes in the presence of 125 µM GTP (Sigma Aldrich) and 3.75 mM EDTA, pH 8.0, for 30 min at 30°C. Nucleotide loading was stopped by addition of 7.5 mM $MgCl_2$. 3.75 nM Gyp7 was added to the reaction, which was filled up to a volume of 80 µl with buffer containing 20 mM HEPES-NaOH, pH 7.4, 150 mM NaCl, 1.5 mM $MgCl_2$. Where indicated, titration of the respective GAP (Gyp7, Gyp7-TBC, and Gyp1-46) was performed, or reaction buffer was added instead. Furthermore, 6 µM Gdi1 was added to the reactions, where indicated. Reactions were incubated for 10 min at 27°C or for the indicated time points. Liposomes with bound proteins were separated from unbound proteins using discontinuous density gradient centrifugation. For this, 100 µl of 2.5 M sucrose dissolved in HKM buffer (20 mM HEPES-NaOH, pH 7.4, 150 mM KOAc, and 2 mM $MgCl_2$) was added to the reactions ("input"). 150 µl of the reactions were transferred to polycarbonate centrifuge tubes (cat#343778; Beckman Coulter), overlayed with 200 µl of 0.75 M sucrose dissolved in HKM buffer, followed by 50 µl HKM buffer. Centrifugation was done at 285,000 $g$, 20°C for 25 min. Liposomes were collected from the top fraction of the sucrose gradient, and proteins were then precipitated by addition of 13% TCA, followed by washing with 100% ice-cold acetone. Samples were analyzed by SDS-PAGE and western blotting using an antibody against Ypt7 (custom-made). Band intensities of the float and input fractions were measured with Fiji (NIH). To quantify the percentage of Ypt7 bound to liposomes, the intensity signal of floated Ypt7 was compared with the intensity signal of the respective input and then normalized to the average value of the reaction containing no GAP.

## HPLC-based GTPase activity assay

An HPLC-based GTPase activity assay was used to compare the GTPase activities of GAPs toward soluble Ypt7 in the presence and absence of membranes (Eberth and Ahmadian, 2009; Araki et al., 2021). 5 µM Ypt7 was incubated with 5 µM GAP and 50 µM GTP in the presence of 1 mM DTT, 20 mM EDTA, pH 8.0, and 5 mM $MgCl_2$ in reaction buffer (50 mM HEPES-NaOH, pH 7.4, 150 mM NaCl). Where indicated, reactions contained 1 mM liposomes of the VML composition or PC/PE liposomes. Control reactions contained either no Ypt7, no GAP, or neither Ypt7 nor GAP. All reactions had a volume of 160 µl and were incubated at

25°C. 30-µl samples of each reaction were snap-frozen after 0 and 300 min reaction time and, where indicated, after 10, 60, and 180 min. All samples were boiled at 95°C for 5 min and then 10% perchloric acid was added. Samples were spun for 30 min, 20,500 $g$ at 4°C. Supernatants were transferred and 20 µl were analyzed with an Agilent1260 Infinity HPLC system equipped with an autoloader and a diode array detector (190–640 nm). Samples were separated on a Nucleodur C18 Pyramid column (5 µm, 125 × 4 mm, Macherey-Nagel) by applying ion pair conditions using a gradient from buffer X (33.72 mM $K_2HPO_4$, 66.28 mM $KH_2PO_4$, pH 6.5; 10 mM tetrabutylammonium bromide) to buffer Y (1:1 buffer X:acetonitrile). The absorbance at 254 nm was monitored, GDP and GTP were eluted after 7.3 and 10.9 min, respectively, and the peak areas were measured with OpenChrom. For each time point, the percentage of GDP and GTP in each sample was determined. The percentage of GTP left at each time point was normalized to the respective percentage of GTP at $t$ = 0 min.

## Fluorescence microscopy and image analysis

Yeast cells were grown in synthetic complete media (SDC+all) overnight at 30°C. In the morning, cells were diluted to an $OD_{600}$ = 0.15 and grown to logarithmic phase at 30°C. 1 $OD_{600}$ equivalent of cells was pelleted. Vacuoles were stained with 7-amino-4-chloromethylcoumarin (CMAC) or FM4-64 (Thermo Fisher Scientific). For CMAC staining of the vacuolar lumen, cells were incubated with 0.1 mM CMAC for 15 min at 30°C, followed by washing with media twice. For staining of the vacuolar membrane with the lipophilic dye FM4-64, pelleted cells were incubated with 30 µM FM4-64 for 20 min at 30°C. Cells were washed with media twice, incubated for 30 min at 30°C, and washed with media once. When mitochondrial DNA was stained, cells were incubated with 1 mg/ml DAPI (Thermo Fisher Scientific) for 15 min, followed by washing with media twice.

To monitor the uptake of the methionine transporter Mup1-GFP, cells were grown overnight in SDC media lacking methionine (SDC-MET) and diluted in SDC-MET media the next morning. Cells of the logarithmic growth phase were either directly imaged or washed in SDC+all media twice, prior to incubation in SDC+all media for indicated time points. For induction of starvation, cells grown in SDC+all media until logarithmic phase were first washed with synthetic minimal medium lacking nitrogen (SD-N) and then incubated in SD-N for 1 or 2 h.

All cells were imaged in synthetic minimal medium at room temperature at a DeltaVision Elite System, an Olympus IX-71 inverted microscope equipped with a 100× NA 1.49 objective, a sCMOS camera (PCO), and an InsightSSI illumination system, 4′,6-diamidino-2-phenylindole, GFP, mCherry, and Cy5 filters. Cells were imaged in z-stacks with 0.4 µM spacing. Deconvolution of images was performed using SOftWoRx software (Applied Precision). All images were processed in Fiji (NIH) and one representative z-slice is depicted for each image. Quantification details are described in the corresponding figure legends.

## Growth test

Yeast cells were grown overnight in YPD media at 30°C. In the morning, cells were diluted to $OD_{600}$ = 0.1 and grown to

logarithmic phase at 30°C. Cells were diluted to $OD_{600} = 0.25$ in YPD, spotted onto plates in serial dilutions (1:10), and incubated at indicated temperatures. Control and selection plates were used. Growth was monitored for several days.

## ET/VT assay to measure TORC1 activities

The assays were carried out as previously described (Gao et al., 2022). Mutant strains and the respective wild-type were transformed with plasmids harboring either the ET reporter (FYVE-GFP-Sch9[C-term], p3027) or the VT reporter (Sch9[C-term]-GFP-Pho8[N-term], p2976). Cells (10 ml) were grown at 30°C in SDC+all until mid-log phase and treated with TCA at a final concentration of 6%. Cells were isolated by centrifugation and the pellet was washed with cold acetone and dried in a speed-vac. The pellet was resuspended in lysis buffer (50 mM Tris-HCl, pH 7.5, 5 mM EDTA, 6 M urea, 1% SDS), the amount being proportional to the $OD_{600}$ of the original cell culture. To extract proteins, cells were lysed by agitation in a Precellys machine after addition of glass beads. After the addition of 2× Laemmli buffer (350 mM Tris-HCl, pH 6.8, 30% glycerol, 600 mM DTT, 10% SDS, bromophenol blue), the mix was boiled at 98°C for 5 min. The analysis was carried out by SDS-PAGE using phosphospecific rabbit anti-Sch9-pThr737 (custom-made) and mouse anti-GFP (cat#11814460001; Roche) antibodies. Band intensities were quantified using Fiji (NIH).

## IEM

SEY6210 mNeon-Ypt7, SEY6210 mNeon-Ypt7 Mon1[Δ100] *TEF1*pr-*GYP7*, and SEY6210 *TEF1*pr-*GFP-YPT7* strains were grown in YPD to exponential phase and then processed for cryo-sectioning as previously described (Griffith et al., 2008). In brief, cultures were rapidly mixed with an equal volume of double-strength fixative (4% [wt/vol] paraformaldehyde, 0.4% [vol/vol] glutaraldehyde in 0.1 M PHEM buffer [20 mM PIPES, 50 mM HEPES, pH 6.9, 20 mM EGTA, 4 mM $MgCl_2$]) and incubated for 15–20 min on a roller at room temperature. The fixative was then replaced by fresh standard strength fixative and fixation proceeded for 3 h at room temperature. Cells were then resuspended in 1 ml of 0.1 M PHEM buffer and transferred in a 1.5 ml microfuge tube, where they were washed three times with the same buffer. Pellets were resuspended in 1 ml of freshly prepared 1% periodic acid in 0.1 M PHEM buffer and incubated at room temperature for 1 h on a roller. Cells were washed again three times with 1 ml of 0.1 M PHEM buffer, before adding 12% gelatine dissolved in 0.1 M PHEM buffer at 37°C. This resuspension was then kept at 37°C for 10 min to properly infiltrate the yeast. After solidification at 4°C, blocks of about 1 mm³ were trimmed under a dissection microscope at 4°C. These gelatine-embedded blocks were immersed overnight in 2.3 M sucrose in rotating vials at 4°C. They were then mounted on ultramicrotome specimen holders and frozen by plunging into liquid nitrogen. After trimming to a suitable block shape, 70 nm ultrathin cryo-sections were cut at –120°C on dry diamond knives (Diatome AG) using either a UC6 or a UCT cryo-ultramicrotome (Leica). Flat ribbons of sections were shifted from the knife-edge with an eyelash and picked up in a wire loop filled with a drop of 1% (wt/vol) methylcellulose, 1.15 M sucrose

in PBS buffer. Sections were thawed on the pickup droplet and transferred, sections downwards, to Formvar carbon-coated copper grids.

Cryo-sections from the SEY6210 *ypt7*Δ pRS406-Ypt7pr-mNeon-4x(GGSG)-Ypt7-Ypt7term and SEY6210 *ypt7*Δ pRS406-Ypt7pr-mNeon-4x(GGSG)-Ypt7-Ypt7term Mon1[Δ100] *TEF1*pr-*GYP7* strains were directly stained with 2% uranyloxalacetate, pH 7, for 5 min, and methyl-cellulose/uranyl acetate, pH 4, for additional 5 min. Cryosections from the SEY6210 TEF1pr-GFP-Ypt7 were first immunogold-labeled with a polyclonal anti-GFP antibody (cat#ab290; Abcam) as previously described (Griffith et al., 2008). Briefly, grids were placed on a Petri dish containing PBS buffer (pH 7.4), prewarmed at 37°C for 30 min to let the pickup solution diffuse away together with the gelatine. This step was repeated twice. Next, the grids were passed over a series of droplets of washing, blocking, antibody, and protein A-gold solutions for routine labeling procedures. After a final wash in distilled water, the sections were left for 5 min on 2% uranyl oxalate (pH 7) and transferred, via a few seconds on a puddle of distilled water, to a mixture of 1.8% methyl cellulose (pH 4) and 0.6% uranyl acetate. After 5 min, the grids were looped out, the excess viscous solution was drained away, and the sections were allowed to dry.

Finally, cell sections were imaged using either a Jeol-1200 or a Jeol-1400 transmission electron microscope (Jeol).

## Statistical analysis

All statistical tests were performed with Origin Pro, Version 9.0 (OriginLab Corporation). Data distribution was assumed to be normal but this was not formally tested. To test the difference between two groups, a two-sided Student's *t* test was used. To compare multiple groups, a one-way ANOVA followed by Tukey's post-hoc test was used. Statistical significance is noted as follows: $*P \leq 0.05$, $**P \leq 0.01$, and $***P \leq 0.001$. All statistical tests and associated P values are named in the figure legends.

## Online supplemental material

Fig. S1 shows additional data for Fig. 1. Fig. S2 shows additional data for Fig. 3. Fig. S3 shows additional data for Figs. 4 and 5. Fig. S4 shows additional data for Figs. 6 and 7. Fig. S5 shows additional data for Fig. 9. Table S1 includes strains, Table S2 plasmids, and Table S3 primers used in this study.

## Data availability

All data used in this study are available upon request.

## Acknowledgments

We thank Angela Perz for expert technical assistance and Clara Taetz and Kevin Tanzusch for experimental support.

This work was supported by the grants of the Deutsche Forschungsgemeinschaft to C. Ungermann (SFB 944, P11; SFB 1557, P14) and to D. Kümmel (SFB 944, P17; SFB 1557, P10), and the Swiss National Science Foundation (310030_184671) to C. De Virgilio. F. Reggiori is supported by Open Competition ENW-KLEIN (OCENW.KLEIN.118), Swiss National Science Foundation Sinergia (CRSII5_189952), and Novo Nordisk Foundation

(0066384) grants. Open Access funding provided by Universitätsbibliothek Osnabrück.

Author contributions: C. Ungermann and L. Langemeyer conceived the project together with N. Füllbrunn. N. Füllbrunn performed all biochemistry and cell biology experiments with the support of A.-C. Borchers and K. Auffarth. R. Nicastro and C. De Virgilio conducted and interpreted the TORC1 activity assays. M. Mari, J. Griffith, and F. Reggiori conducted and interpreted the IEM analysis. E. Herrmann, R. Rasche, and D. Kümmel analyzed the in vitro GAP assays together with N. Füllbrunn. N. Füllbrunn, C. Ungermann, and L. Langemeyer wrote the manuscript with contributions of all authors.

Disclosures: The authors declare no competing interests exist.

Submitted: 12 May 2023

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

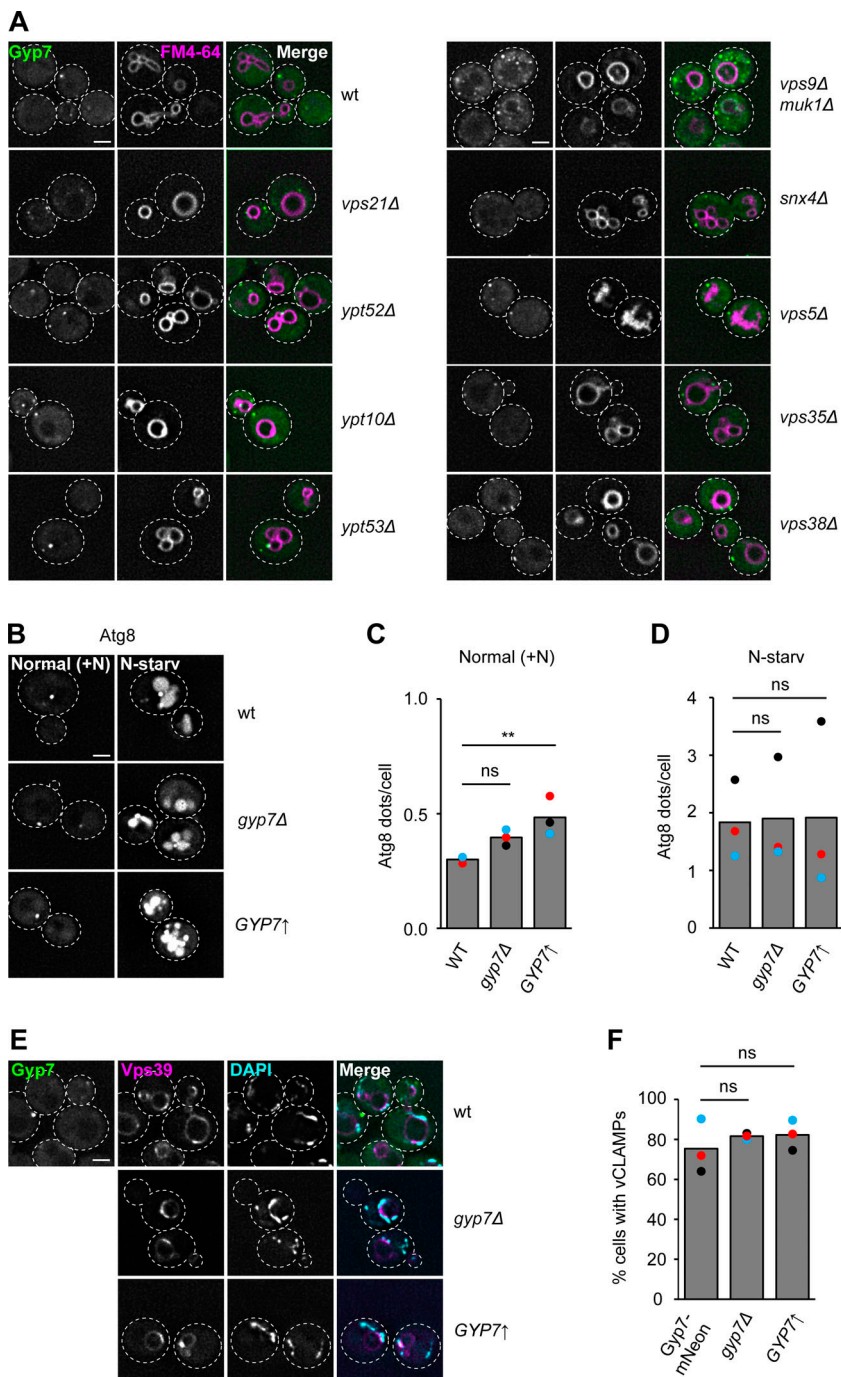

Figure S1. **Gyp7 does not affect Ypt7 function in autophagy or vCLAMP formation. (A)** Localization of Gyp7 in selected deletion mutants. Gyp7 was tagged with mNeonGreen in wild-type (wt), *vps21Δ*, *ypt52Δ*, *ypt10Δ*, *ypt53Δ*, *vps9Δ muk1Δ*, *snx4Δ*, *vps5Δ*, *vps35Δ*, and *vps38Δ* cells. Vacuolar membranes were stained with FM4-64. Cells were imaged by fluorescence microscopy and individual slices are shown. Dashed lines indicate yeast cell boundaries. Scale bar, 2 μm. **(B)** Localization of Atg8 upon deletion or overexpression of Gyp7. Gyp7 was either deleted or expressed from the *TEF1* promoter in cells encoding the autophagy-specific marker protein mCherry-Atg8. Cells were grown in SDC+all and then shifted to SD-N for 1 h (see Materials and methods). Cells were imaged by fluorescence microscopy and individual slices are shown. Dashed lines indicate yeast cell boundaries. Scale bar, 2 μm. **(C)** Quantification of Atg8 dots per cell in B during growth. Cells (*n* ≥ 150) from three independent experiments were quantified in Fiji. Bar graphs represent the averages from three experiments and dots represent the mean of each experiment. P value ns, **<0.01, using ANOVA one-way test. **(D)** Quantification of Atg8 dots per cell in B during N-starvation. Quantification was performed as in C. P value ns, using ANOVA one-way test. **(E)** Formation of vCLAMPs upon deletion or overexpression of Gyp7. The vCLAMP-forming protein mCherry-Vps39 was expressed from the *TEF1* promoter in cells encoding Gyp7-mNeonGreen, *gyp7Δ*, and *TEF1*pr-*GYP7* cells. Mitochondria were stained with DAPI (see Materials and methods). Cells were imaged by fluorescence microscopy and individual slices are shown. Dashed lines indicate yeast cell boundaries. Scale bar, 2 μm. **(F)** Quantification of E. Colocalization of mCherry-Vps39 enrichments and DAPI-stained mitochondria was defined as vCLAMP. Cells with ≥1 vCLAMP were counted as vCLAMP-positive cells. Cells (*n* ≥ 150) from three independent experiments were quantified in Fiji. Bar graphs represent the averages from three experiments and dots represent the mean of each experiment. P value ns, using ANOVA one-way test.

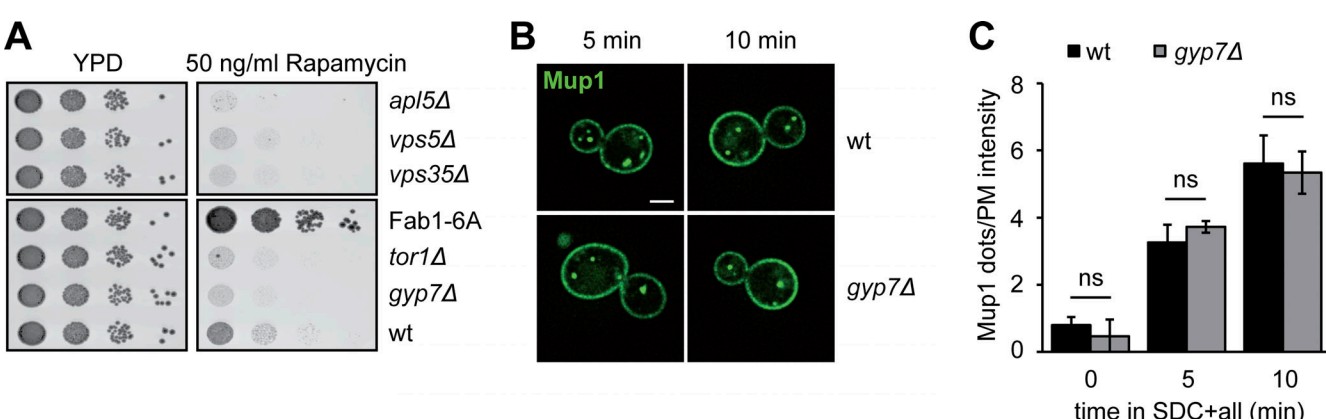

Figure S2.    **Gyp7 function is required for normal TORC1 activity. (A)** Growth assay on rapamycin-containing plates. Indicated yeast strains were grown to the same OD$_{600}$ in YPD media and serial dilutions were spotted onto agar plates containing YPD or YPD supplemented with 50 ng/ml rapamycin. Plates were incubated at 30°C for several days before imaging. Images are representative of three independent experiments. **(B)** Endocytosis of Mup1 in wild-type (wt) and *gyp7Δ* cells as in Fig. 3 C. Cells were imaged at indicated time points by fluorescence microscopy. Individual slices are shown. Scale bar, 2 µm. **(C)** Quantification of the number of dots to plasma membrane (PM) intensity of Mup1 ratio in B and Fig. 3 C. For each cell, the number of Mup1 dots was divided by the maximum fluorescence intensity of Mup1-GFP signal at the plasma membrane. For each time point, cells (*n* ≥ 100) from three independent experiments were quantified in Fiji. Bar graphs represent the averages and error bars the SD from three experiments. P value ns, using two-sided Student's *t* test.

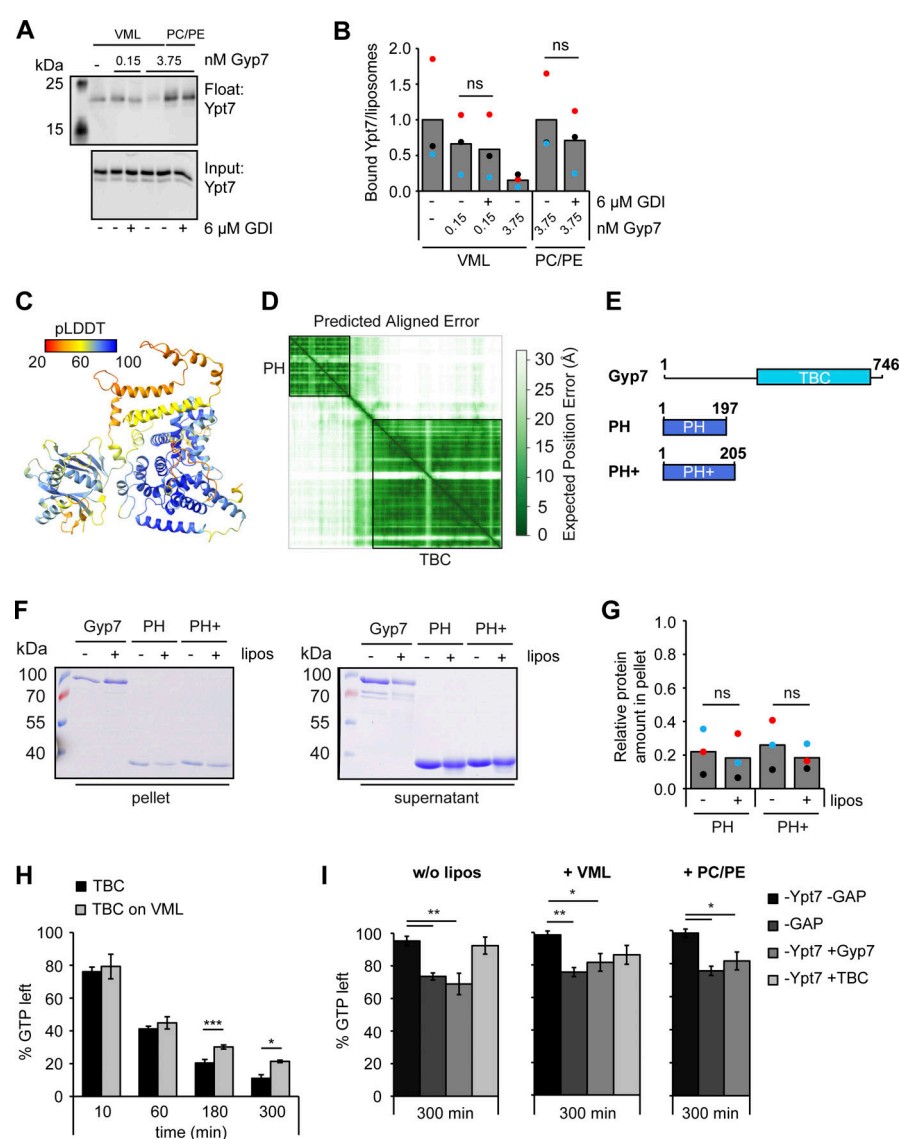

Figure S3. **The N-terminal PH domain of Gyp7 does not bind membranes. (A)** 250 µM PC/PE liposomes or liposomes of VML composition were preloaded with 0.6 µM Ypt7-GDI complex in the presence of 3.75 mM EDTA and 125 µM GTP. Nucleotide binding was stabilized by addition of 7.5 mM MgCl₂. Reactions were incubated with different amounts of Gyp7 and, where indicated, with 6 µM GDI for 10 min. Liposomes were floated in a sucrose gradient. Control reaction contained respective liposomes of VML composition and no Gyp7. 40% of the float was analyzed together with 3% input by western blotting using an anti-Ypt7 antibody. **(B)** Quantification of bound Ypt7 to liposomes in A. Band intensity of Ypt7 signal in float was measured in Fiji and compared with input. Reactions containing Gyp7 were normalized to the average value of the control reaction. Bar graphs represent the averages from three independent experiments and dots represent the mean of each experiment. P value ns, using ANOVA one-way test. **(C)** AlphaFold2 structure prediction of Gyp7 color-coded according to the pLDDT values. **(D)** Plot of the Predicted Aligned Error of C with the PH and TBC domain of Gyp7 labeled. **(E)** Comparison of full-length Gyp7 and its PH domain. Two PH domain constructs (PH = aa 1–197; PH+ = aa 1–205) contain the N-terminal region of Gyp7 (aa 1–746). **(F)** Membrane association of the PH domain compared to full-length Gyp7. 715 nM protein was incubated with 715 µM liposomes of VML composition for 10 min. Membranes were separated from supernatant by centrifugation at 100,000 $g$ and both fractions were analyzed by SDS-PAGE and Coomassie staining. Control reactions contained no liposomes. **(G)** Quantification of the relative protein amount in the pellet in F. Band intensity of protein signal in the pellet was measured in Fiji and compared with the protein signal in the supernatant. Bar graphs represent the averages from three independent experiments and dots represent the mean of each experiment. P value ns, using ANOVA one-way test. **(H)** Comparison of the Gyp7 TBC domain activity toward soluble Ypt7-GTP in solution and on membranes. 5 µM Ypt7 was incubated with 5 µM GAP and 50 µM GTP in the presence of 1 mM DTT, 20 mM EDTA, and 5 mM MgCl₂. Where indicated, reactions contained 1 mM liposomes with VML composition. Control reactions contained no Ypt7, no GAP, or neither Ypt7 nor GAP (see Fig. S3 I). Reactions were stopped after 0, 10, 60, 180, and 300 min by snap-freezing and boiling at 95°C. Samples were applied to a HPLC system and the absorbance of GDP and GTP was monitored at 254 nm. Peaks were analyzed with OpenChrom and for each time point the percentage of GDP and GTP in the samples was determined. The percentage of GTP left at each time point was normalized to the respective percentage of GTP at $t$ = 0 min. Normalized % GTP left plotted against the time in min. Bar graphs represent the averages and error bars the SD from three independent experiments. P value *<0.05, ***<0.001, using ANOVA one-way test. **(I)** No major GTP hydrolysis in control reactions of Fig. 5 J and H after 300 min. Control reactions contained no Ypt7, no GAP, or neither Ypt7 nor GAP. For each time point the percentage of GDP and GTP in the reactions was determined. The percentage of GTP left of each sample at $t$ = 300 min was normalized to the respective percentage of GTP at $t$ = 0 min. Normalized % GTP left plotted against the time in min. Bar graphs represent the averages and error bars the SD from three independent experiments. P value *<0.05, **<0.01, using ANOVA one-way test. Source data are available for this figure: SourceData FS3.

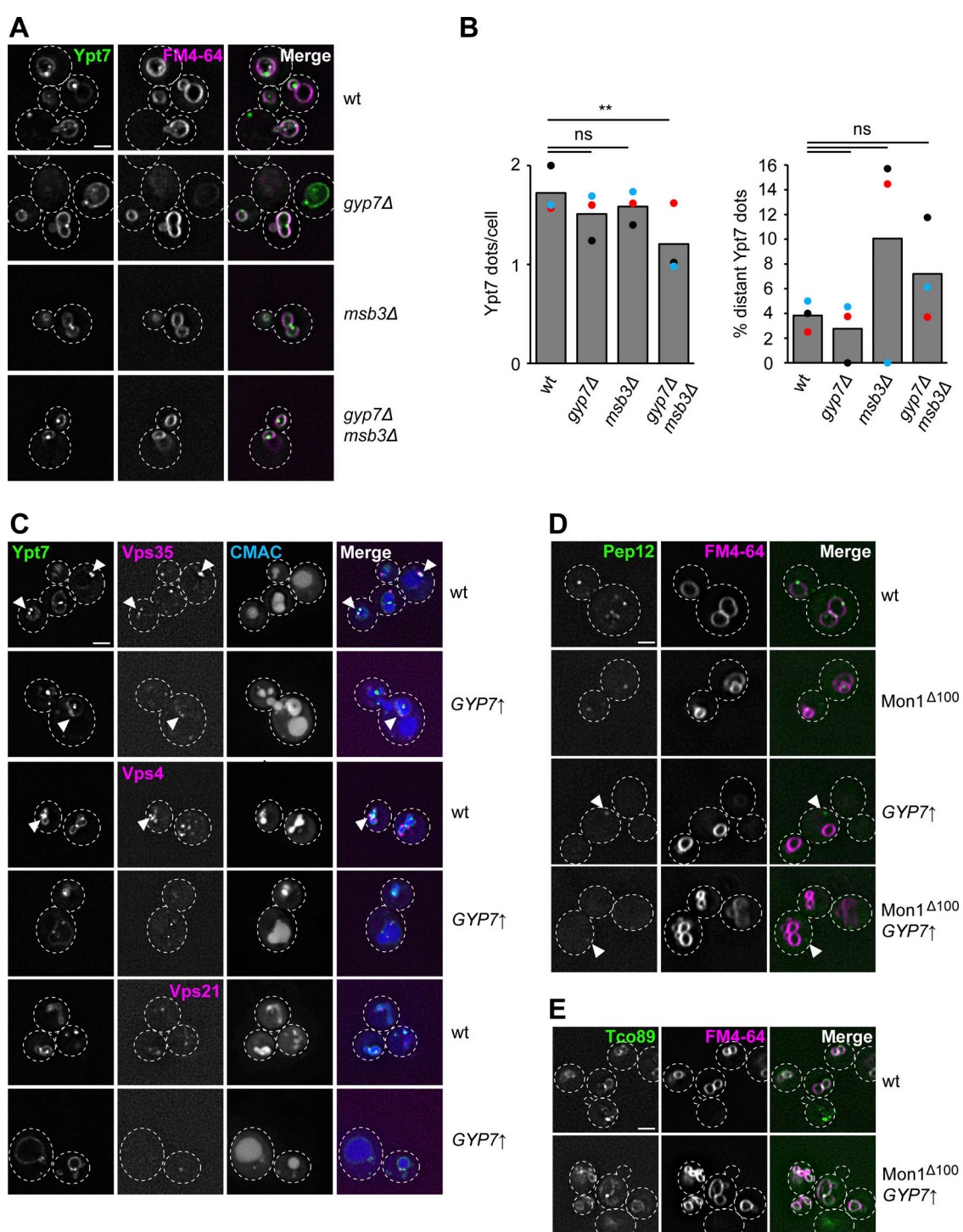

Figure S4. **Confined Ypt7 dots correspond to signaling endosomes. (A)** Localization of mNeon-Ypt7 in wild-type (wt) cells and cells with Gyp7 and/or Msb3 deleted. Vacuolar membranes were stained with FM4-64. Cells were imaged by fluorescence microscopy. Individual slices are shown. Dashed lines indicate yeast cell boundaries. Scale bar, 2 µm. **(B)** Quantification of the total number of Ypt7 dots per cell and the percentage of distant Ypt7 dots in A. The number of distant Ypt7 dots (not at the vacuole) was divided by the total number of Ypt7 dots per cell. Cells (*n* ≥ 150) from three independent experiments were quantified in Fiji. Bar graphs represent the averages from three experiments and dots represent the mean of each experiment. P value ns, **<0.01, using ANOVA one-way test. **(C)** Localization of mNeon-Ypt7 dots relative to endosomal marker proteins. Endosomal markers Vps35-mKate, Vps4-3xHA-mCherry, and mCherry-Vps21 were co-expressed in *TEF1*pr-*GYP7* or wild-type cells encoding endogenous mNeon-Ypt7. Vacuoles were stained with CMAC. Cells were imaged by fluorescence microscopy. Individual slices are shown. Arrowheads depict representative colocalization. Dashed lines indicate yeast cell boundaries. Scale bar, 2 µm. **(D)** Localization of GFP-Pep12 in wild-type cells and cells expressing Gyp7 from the *TEF1* promoter and/or Mon1$^{\Delta100}$-Ccz1. Vacuolar membranes were stained with FM4-64. Cells were imaged by fluorescence microscopy. Individual slices are shown. Arrowheads depict distant Pep12 dots. Dashed lines indicate yeast cell boundaries. Scale bar, 2 µm. **(E)** Localization of Tco89-mNeon in wild-type and *TEF1*pr-*GYP7* cells. Vacuolar membranes were stained with FM4-64. Cells were imaged by fluorescence microscopy. Individual slices are shown. Dashed lines indicate yeast cell boundaries. Scale bar, 2 µm.

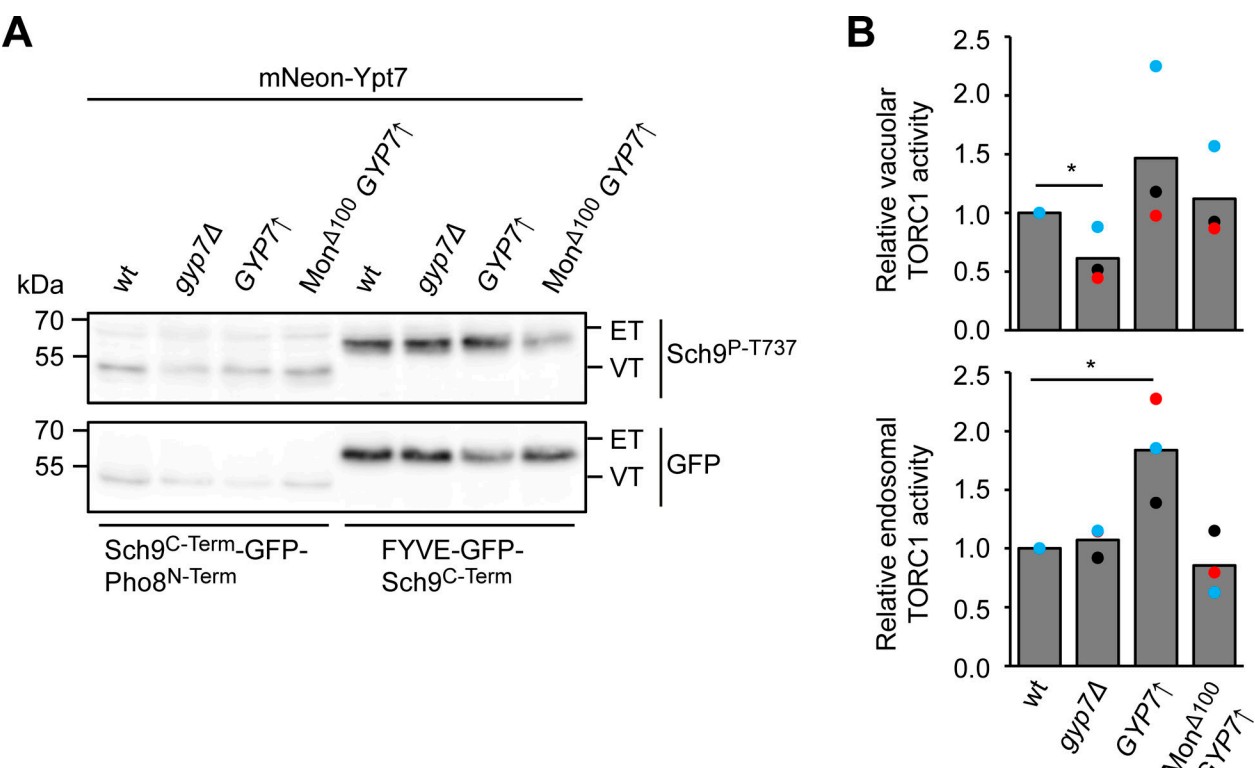

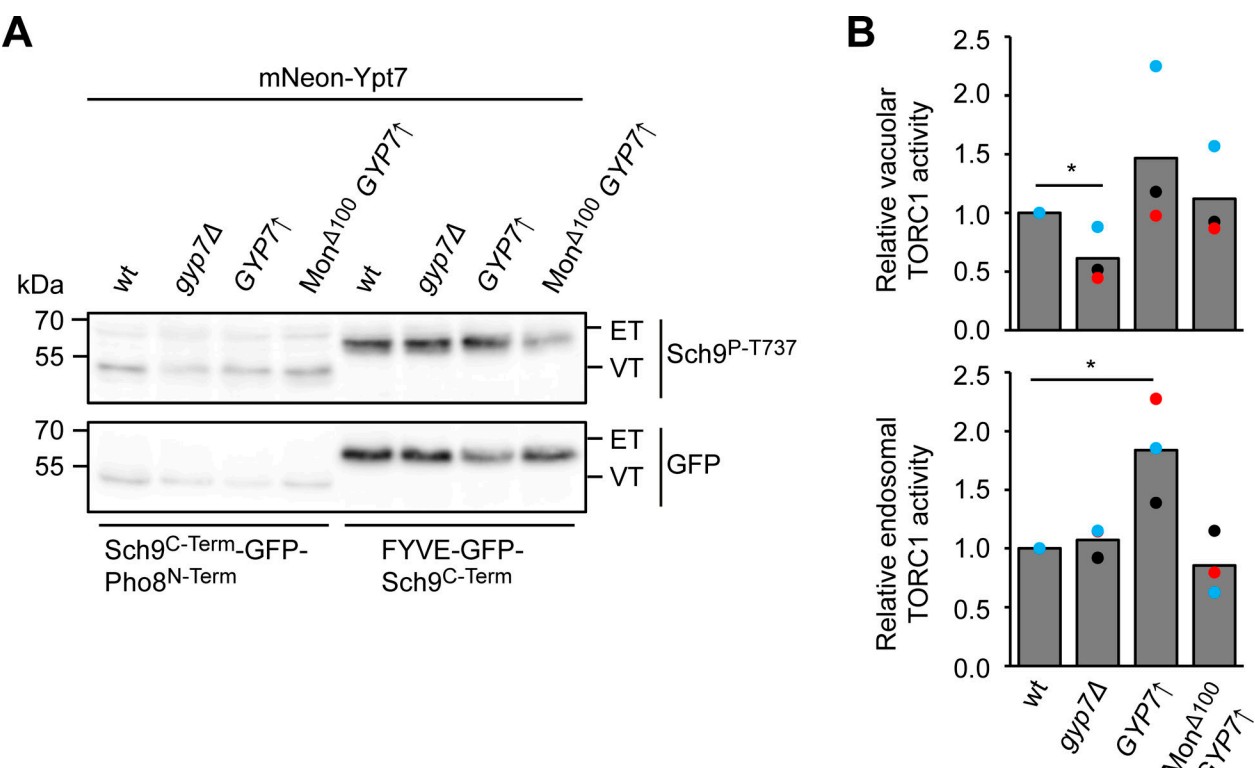

Figure S5. **Gyp7 affects endosomal and vacuolar TORC1 activity. (A)** mNeon-Ypt7 background cells with either wild-type (wt) expression levels of Gyp7, Gyp7 deleted, Gyp7 overexpressed from the *TEF1* promoter or both overexpressed Gyp7 combined with truncation of Mon1 (Mon1$^{\Delta100}$) were transformed with either vacuolar (VT, Sch9$^{C-term}$-GFP-Pho8$^{N-term}$) or endosomal (ET, FYVE-GFP-Sch9$^{C-term}$) TORC1 activity reporters and grown exponentially in SDC+all. After SDS-PAGE analysis of the corresponding extracted proteins, the expression of the ET/VT reporters was detected by immunoblotting using anti-GFP antibodies, and their phosphorylation levels by using phospho-specific anti-Sch9-pThr$^{737}$ antibodies were analyzed likewise. **(B)** Quantifications of the ET/VT assay in A, expressed as the ratios of Sch9-pThr$^{737}$/GFP signals and normalized with wild-type cells. Bar graphs represent the averages from three independent experiments, and dots represent the mean of each experiment. P value *<0.05, using a two-sided Student's *t* test. Source data are available for this figure: SourceData FS5.

**Provided online are three tables. Table S1 shows the strains used in this study. Table S2 lists the plasmids used in this study. Table S3 lists primers used in this study.**

