## [Peer Review File · The Journal of Cell Biology]

The GTPase activating protein Gyp7 regulates Rab7/Ypt7 activity on late endosomes

Nadia Füllbrunn, Raffaele Nicastro, Muriel Mari, Janice Griffith, Eric Herrmann, Rene Rasche, Ann-Christin Borchers, Kathrin Auffarth, Daniel Kümmel, Fulvio Reggiori, Claudio De Virgilio, Lars Langemeyer, and Christian Ungermann

Corresponding Author(s): Christian Ungermann, Osnabrück University

Review Timeline:

Submission Date:	2023-05-12
Editorial Decision:	2023-07-13
Revision Received:	2024-01-22
Editorial Decision:	2024-02-27
Revision Received:	2024-03-06

Monitoring Editor: Harald Stenmark

Scientific Editor: Tim Fessenden

Transaction Report:

DOI: <https://doi.org/10.1083/jcb.202305038>

July 13, 2023

Re: JCB manuscript #202305038

Prof. Christian Ungermann
Osnabrück University
Biology/Chemistry
Barbarastrasse 13
Osnabrück 49076
Germany

Dear Prof. Ungermann,

Thank you for submitting your manuscript entitled "The GTPase activating protein Gyp7 regulates the activity of the Rab7-like Ypt7 and signaling at late endosomes". The manuscript has been evaluated by expert reviewers, whose reports are appended below. Unfortunately, after an assessment of the reviewer feedback, our editorial decision is against publication in JCB.

You will see that reviewers commended the intriguing new observations on the regulation of Ypt7 by the GAP Gyp7 based in part on its membrane localization. However, reviewers raised significant concerns over data interpretation and controls, which reduced their confidence in the main conclusions set forth in this study. In particular, Reviewer 2 noted that multiple important conclusions relied on overexpression constructs without confirmation of key results using endogenous gene expression levels. This reviewer also sought evidence of Ypt7 GTPase activity and vacuole lipid composition (point 4). Multiple reviewers also requested measurements of Ypt7 localization at endosomes vs at vacuoles. Last, Reviewer 1 requested improvements to the text towards greater clarity.

We feel that the requests made by the reviewers are more substantial than can be addressed in a typical revision period. If you wish to expedite publication of the current data, it may be best to pursue publication at another journal. However, given interest in the topic and the JCB's interest in publishing this work, we would be open to resubmission to JCB of a significantly revised manuscript that fully addresses the reviewers' concerns noted above and is subject to further peer-review. Should you wish to pursue publication with a revised manuscript, please provide a plan for revision in an appeal request. Please note that we may discuss the revision plan with at least one reviewer. If and when you would like to resubmit this work to JCB, please contact the journal office to discuss an appeal of this decision or you may submit an appeal directly through our manuscript submission system.

Regardless of how you choose to proceed, we hope that the comments below will prove constructive as your work progresses. We would be happy to discuss the reviewer comments further once you've had a chance to consider the points raised in this letter. You can contact the journal office with any questions, cellbio@rockefeller.edu or call (212) 327-8588.

Thank you for thinking of JCB as an appropriate place to publish your work.

Sincerely,

Harald Stenmark
Monitoring Editor
Journal of Cell Biology

Tim Fessenden
Scientific Editor
Journal of Cell Biology

Reviewer #1 (Comments to the Authors (Required)):

This is an interesting study investigating the function of the Rab-GAP Gyp7 in budding yeast. The authors use a combination of approaches to characterize the role of Gyp7 in regulation of Ypt7, the yeast Rab7 homolog.

The authors show that Gyp7 localizes to endosomes and that forcing Gyp7 to localize to the vacuole (yeast lysosome) by fusing

it to vacuolar proteins alters vacuolar morphology. They also find that Gyp7 is required for normal cellular resistance to ZnCl₂ and rapamycin and efficient endocytosis of Mup1, indicating loss of Gyp7 sensitizes cells to endocytic stress and TORC1 inhibition.

The authors find that Gyp7 localization does not require several endosomal proteins for its localization. In order to gain more information regarding how Gyp7 localizes to endosomes, the authors perform in vitro studies in which they examine the requirements for Gyp7 membrane-binding and GAP activity. They find that Gyp7 binds well to and has Ypt7 GAP activity upon liposomes comprised of vacuolar lipids but not simple PC/PE lipids. Interestingly, even when Gyp7 is forced to bind to PC/PE liposomes, using a His-tag and nickel-chelated-lipids, Gyp7 but is still not very active. This suggests a specific membrane environment is important for both binding and activity of Gyp7.

They find that while loss of Gyp7 has no obvious effect on Ypt7 localization, overexpression of Gyp7 essentially removes Ypt7 from the lysosome (vacuole) membrane and therefore results in enrichment of Ypt7 on endosomes. They find that hyperactivation of Ypt7 at endosomes by overexpression of Gyp7 slows the kinetics of Mup1 endocytosis. Interestingly, this means that both loss of Gyp7 and overexpression of Gyp7 have similar effects on endocytosis. They also find that hyperactivation of Ypt7 on endosomes results in slight resistance to rapamycin. Finally, they observe that overexpression of Gyp7 results in accumulation of the endocytic tracer FM-464 in Ypt7-positive endosomes in the absence of ESCRT function. Taken together the authors interpret these results to mean that Ypt7 functions on "signaling" endosomes.

Overall this is an interesting study but at times I found the explanation or interpretation of results to be a bit unclear. Below are my suggestions for improvement:

1. I found the presentation of the Gyp7 localization results to be a bit unclear regarding which compartment the authors consider it to localize to. Is it possible that the differential localization of Gyp7 and other endosomal proteins reflects different timing/kinetics rather than distinct compartments? For example, different Golgi proteins appear to have different localizations but when observed over time they are seen to localize to the same compartment just with different kinetics. This possibility is mentioned in the discussion but it would be good to clarify and mention this possibility when the results are presented. These are the phrases that made me a bit confused: "We further show that Gyp7 overproduction can retain Ypt7 on late endosomes, which enhances endosomal TORC1 signaling. These Ypt7-positive endosomes lack ESCRTs, yet require ESCRTs for their formation. We thus speculate that these late endosomes correspond to signaling endosomes." and "We thus conclude that Ypt7 functions on mature MVBs, which in part correspond to signaling endosomes." Are signaling endosomes a subset of late endosomes? How are they defined?
2. Similarly, can the authors include at an earlier point in their manuscript an explicit description of how they are distinguishing "signaling endosomes" from "late endosomes", and also how each of these relates to what has been called the "pre-vacuolar endosome (PVE)"? They have some description of signaling endosomes in the discussion, but I found it confusing to see this term mentioned multiple times in the results sections without understanding how they are distinguishing a signaling endosome from a late endosome or PVE.
3. In Figure 1, how can the authors distinguish the difference between disrupted endosomal morphology versus disrupted Gyp7 recruitment to endosomes? Also, what is special about Mvp1 versus other ESCRT components?
4. The following two statements seem to conflict with each other, and I think the second statement is more accurate than the first statement:
"Our data suggest that a functional Rab5 system is required for correct Gyp7 localization to endosomes." (line 165)
"This suggests that Gyp7 recruitment to endosomes occurs independent of the analyzed endosomal proteins." (line 176)
5. It would be very helpful to include a more straightforward analysis of the relationship between Gyp7 and Ypt7 localizations. The experiments involving how overexpression of Gyp7 induce more Ypt7 localization at endosomes, which is apparently the same compartment where Gyp7 itself localizes, are a bit puzzling. In principle one would expect a GAP to antagonize the localization of its Rab. One possibility is that overexpression of Gyp7 causes a shift in localization of Gyp7 to the vacuole. It would be straightforward for the authors to test if this is the case by repeating the Gyp7 overexpression experiments using a fluorescent-tagged version of Gyp7. This could potentially provide a simple explanation for the observed effects on Ypt7 localization. For example, in Figure 6A, the localization of Ypt7 is shown with and without Gyp7 and when Gyp7 is overexpressed, but Gyp7 localization itself is not observed at the same time. Do Gyp7 and Ypt7 normally co-localize? Do they colocalize when Gyp7 is overexpressed?
6. I think the sentence: "Thus, Gyp7 function is required for normal TORC1 activity within the endolysosomal system" (lines 220-221) is a bit of an overstatement at this point in the manuscript because the authors have only shown sensitivity to Rapamycin and have not shown any direct measure of TORC1 activity (i.e. changes in substrate phosphorylation).
7. The loss of Gyp7 function does not affect Ypt7 localization. One might expect Ypt7 to have a more broad or intense localization in the absence of its GEF. Can the authors comment on whether this might be because another GYP gene also acts

as a GAP for Ypt7?

8. There appears to be some redundancy in these two sentences: (line 406) "Surprisingly, Gyp7 overproduction does not liberate Ypt7 from endosomes, but rather confines it to a subpopulation proximal to the vacuole. This effect is even stronger when Gyp7 is overexpressed, and ..."

Reviewer #2 (Comments to the Authors (Required)):

In this clearly written manuscript, Füllrunn and coworkers report studies of the budding yeast Rab GAP Gyp7. They present genetic and cell biological studies which confirm and extend prior work from three other labs showing that Gyp7 is the major GAP that inactivates Rab7 (Ypt7), and present data which they interpret to indicate that an endosomal compartment or compartments is the major *in vivo* site of Gyp7 action. Biochemical experiments show that Gyp7 has a membrane binding activity that exhibits selectivity for lipid composition. Several of the reported experiments are interesting but as discussed below key conclusions are based on non-physiological genetic perturbations (overexpression) and several experiments do not include controls necessary for interpretation of the results, tempering my enthusiasm for the manuscript. It is possible that some of the needed data are already in hand but not shown. With some additions and a more tempered interpretation of the results, I'd be happy to take another look at this study.

Major points.

1. "Gyp7 localizes to endosomes." [line 142] The authors show that overexpressed Gyp7 localizes to punctate structures that appear to label with the endocytic tracer FM4-64. However, no co-localization with known protein markers of endosomes is shown, except to a limited extent in a *vps4Δ* background, where dozens of markers accumulate at class E compartments. This is an odd omission. Moreover, the authors see *more* localization of Gyp7 to punctate structures when Rab5 or Rab5 effector function is impaired, not less - and these punctae do not seem to be marked by FM4-64. It is hard to see this as support for the hypothesis that Gyp7 localizes to endosomes. Could these be, for example, Atg8 accumulations rather than endosomes?

2. "Relocalization of Gyp7 to vacuoles impairs vacuole morphology." [line 178] This is a reasonable conclusion on the basis of overexpression as previously reported and experiments shown here (Fig. 2A,B). However, the re-targeting experiments (Fig. 2C,E) show much larger effects for the affinity-tagged Vac8-CB used as an anchor to relocalize Gyp7 than for the relocalization itself. Or perhaps I'm misreading the experiment? I asked two other experienced people in my lab to read this section of the paper, and they read it the same way. I don't see how this experiment can be interpreted using a background with what seems to be a reasonably strong *vac8* hypomorph.

Additionally, it's hard to see how expression of a presumptively spontaneous nucleotide-exchanging variant of Ypt7 is a better control here than a catalytic-dead Gyp7 (R458K), as used in previous studies (Eitzen, EMBO J 2000; Brett, JCB 2008). Use of this well-characterized mutant could have strengthened several experiments in the present study. It's perplexing that R458K was not employed in this study.

3. (Gyp7 is required for homeostasis of the endosomal system.) [line 204] The authors show data suggesting that perturbation of Gyp7 function alters TORC1 signaling, consistent with the known role of endolysosomal traffic in the TORC1 pathway. It is interesting that an *msb3* (Rab5 GAP) mutant phenocopies the *gyp7* deletion for this readout.

Data are also shown suggesting that traffic kinetics through the endosomal MVB pathway to the vacuole are (very) subtly regulated by Gyp7 activity. The experiments do not clearly delineate whether the target of this regulation is Ypt7 residing on the endosome, on the vacuole, or both.

4. "Gyp7 activity depends on the membrane environment." [line 232]. It is persuasively shown that Gyp7 binds membranes, that it prefers to bind membranes with a vacuole-like membrane mixture (an endosomal vs. vacuole lipid mixture was not tested, as might have been expected given the overall argument of the paper), and that this activity depends on a PH-like domain near the protein's N-terminus. The PH-like domain alone does not bind membranes in the experimental configurations employed.

The authors use mainly GDI extraction as a proxy for Gyp7 activity against Ypt7/Rab7. There's nothing wrong with this approach, as such. But curiously direct assay of Ypt7 GTPase activity is reported solely in Fig. 5J. The authors claim that this shows allosteric regulation of Gyp7 activity against soluble (non-lipidated) Ypt7 by membranes. The result shows a very small but apparently reproducible difference in activity. But given the advantages of a chemically defined system, why was GTPase activity not assayed directly throughout? This is not hard to do using well-described colorimetric, fluorescence, or [32]P orthophosphate release assays, or presumably the HPLC assay in Fig. 5J.

Given the absence of direct readouts of GTP hydrolysis, it is important to test whether the lipid mix used (VML vs. PC/PE) influences the ability of GDI to extract Ypt7-GDP. This control is important if extraction is used as the main proxy for the Rab's

nucleotide state. Also, it was not clear to this reader whether GDI is present in excess to Ypt7, or what the final GDI concentration was in the extraction experiments.

Overall, the experiments support the idea that direct membrane association increases Gyp7 activity against Ypt7. They do not strongly support the idea that membrane association has a major allosteric effect on Gyp7 catalytic activity.

5. "Gyp7 activity confines Ypt7 to late endosomes and signaling endosomes." Taken literally, this is obviously wrong, since Ypt7 on vacuoles is needed for vacuole fusion, as exhaustively demonstrated by many labs including the authors', and the data show (as entirely expected) lots of Ypt7 on the vacuole in wild type cells. Fig. 6A also shows that overproduction of Gyp7 removes Ypt7 from the vacuole, and if anything, increases its localization to (presumptively) endosomal punctae. This would seem to argue that Gyp7 preferentially targets Ypt7 on the vacuole, not on the endosome as the authors suggest earlier in the manuscript.

Other experiments here are based on a truncation of the GEF subunit Mon1 that results in elevated Ypt7 activity, as nicely shown in recent work from the same group. But Gyp7 is not shown to colocalize with Mon1 or Ypt7 under these circumstances. An interesting observation is that endosomes marked by Pep12 increase in number in a *MON1Δ100* mutant that also overproduces Gyp7. However, it's not tested whether this phenotype is due to one of these genetic manipulations, or both (Fig. 6E).

6. "Endolysosomal transport is delayed upon Ypt7 confinement to late endosomes." [line 338]. The delays are again subtle but apparently statistically significant, and consistent with the ability of Gyp7 to deplete Ypt7 from the vacuole as shown in Fig. 6A.

7. "Ypt7-positive structures correspond to signaling endosomes." Immunogold EM shows that overproduced Ypt7 can be detected on endosomal structures, and Ypt7 accumulates on Class E compartments in a *vps4Δ* mutant (along with dozens of other endolysosomal proteins). In Fig. S6A,B a reporter system is used to assay endosomal vs. vacuolar phosphorylation of Sch9 by Tor1. In a *gyp7Δ* mutant vs. wild type, a significant decrease in TORC1 activity is seen at the vacuole and *not* at the endosome. Overproduction does increase signaling at the endosome, but given the lack of a deletion phenotype, this is not a strong argument for a normal physiological function of Gyp7 at the endosome per se. I wonder if stronger phenotypes would emerge in nitrogen limited conditions.

Minor issues.

8. The paper by Eitzen (EMBO J 2000) is not cited, and should be.
9. Line 216: In yeast, Apl5 is not an endosomal trafficking protein.
10. Line 224: Fig. 3C is not mentioned in the Results, so far as I can tell.
11. Fig. S2B: genotypes should be labeled.

- Alexey Merz

Reviewer #3 (Comments to the Authors (Required)):

In the present study, Füllbrunn et al. dissect the endocytic localization and function of the Ypt7 (RAB7) specific GAP protein Gyp7 in yeast. While Gyp7 is already known to be a GAP for RAB7, the precise localization and membrane dependency of Ypt7 inactivation through Gyp7 remained to be elucidated.

The authors demonstrate that Gyp7 localizes primarily to endosomes but not to the vacuole and that this localization partially depends on an intact Vps21 (RAB5) system. Additional localization experiments indicate that Gyp7 functions on endosomes but likely not on the vacuolar membrane. Deletion of Gyp7 delayed endosomal transport towards the vacuole and altered endosomal mTORC1 signaling, suggesting that Gyp7 is required for the homeostasis and signaling function of endosomes. In an additional line of experimentation, the authors demonstrate that Gyp7 requires endosomal membranes for its GAP activity as membrane free Gyp7 was hardly active towards Ypt7. Finally, the authors demonstrate that Gyp7 activity confines Ypt7 to late endosomes which are also signaling endosomes.

Overall, the data is of high quality and the authors' conclusions appear reasonable to this reviewer. The authors thoroughly dissect the localization of Gyp7, its effect on Ypt7 and its role within the endocytic network. With this being said, I think that the manuscript is somewhat uninspiring as Gyp7 was already known to be the dominant Ypt7 GAP protein in yeast. It is still a solid and thorough cell biological analysis of a previously known RAB7 GAP in yeast but it doesn't add a lot of groundbreaking insight into the function of this endocytic protein. While I am generally supportive of publication I am not sure whether JCB is an appropriate venue for this manuscript.

Minor points:

Figure 4A: "floatation" seems odd

Below is a list of experiments that we performed to address the reviewers' comments:

- Determine the precise localization of Gyp7 in the endosomal system:
 - Localization of Gyp7 relative to endosomal and other organellar marker proteins (Vps8, Vps21, Ccz1, Vps35, Vps5, Vps4, Ivy1, Mnn9, Sec7, Ypt7) (Reviewer #1 and #2, not incorporated in Figures, attached to this document)
 - Localization of fluorescently-labeled Gyp7 relative to endogenously expressed Ypt7 and Mon1-Ccz1 in wild-type vs. Gyp7-overproducing vs. Mon1 Δ 100-Ccz1 expressing cells (Reviewer #1 and #2; Figure 6C-E)
- Determine if another GAP can replace Gyp7:
 - Localization of Ypt7 in *gyp7 Δ msb3 Δ* cells (Reviewer #1, Figure S5A-B)
- Show that Gyp7 activity is responsible for the vacuole phenotype due to relocalization:
 - Relocalization of wild-type Gyp7 and the catalytic-deficient Gyp7 mutant (R458K) to vacuoles with a Chromobody attached to the vacuolar protein Zrc1 (Reviewer #2; Figure 2C-G)
- Establish whether GDI extracts Ypt7 on all membranes:
 - Gyp7 activity towards Ypt7 on VMLs vs. PC/PE liposomes in the presence of excess GDI (Reviewer #2, Figure S3A-B)
 - GDI extraction assay: Gyp1-46 activity towards Ypt7 on VMLs vs. PC/PE liposomes (Reviewer #2, Figure 4L-M)
- Determine which of the two factors (hyperactive GEF, Mon1 Δ 100, or Gyp7 overproduction) has the predominant effect on the endosomal system:
 - Localization of Pep12 in Mon1 Δ 100-Ccz1 expressing vs. Gyp7-overproducing cells (Reviewer #2, Figure 7C-D, S6B)
- Examine whether the *gyp7 Δ* mutant has a stronger phenotype if challenged by nitrogen starvation, we analyzed the vacuole morphology under these conditions (Reviewer #2, Figure 3G-H)
- Furthermore, we implemented electron microscopy analysis of cells expressing mNeon-Ypt7 in wild-type and Mon1 Δ 100-Ccz1 *TEF1pr-GYP7* cells to analyze a potential effect on MVBs (morphology and number per cell) (Figure 9C).

Reviewer #1 (Comments to the Authors (Required)):

This is an interesting study investigating the function of the Rab-GAP Gyp7 in budding yeast. The authors use a combination of approaches to characterize the role of Gyp7 in regulation of Ypt7, the yeast Rab7 homolog.

The authors show that Gyp7 localizes to endosomes and that forcing Gyp7 to localize to the

vacuole (yeast lysosome) by fusing it to vacuolar proteins alters vacuolar morphology. They also find that Gyp7 is required for normal cellular resistance to ZnCl₂ and rapamycin and efficient endocytosis of Mup1, indicating loss of Gyp7 sensitizes cells to endocytic stress and TORC1 inhibition.

The authors find that Gyp7 localization does not require several endosomal proteins for its localization. In order to gain more information regarding how Gyp7 localizes to endosomes, the authors perform in vitro studies in which they examine the requirements for Gyp7 membrane-binding and GAP activity. They find that Gyp7 binds well to and has Ypt7 GAP activity upon liposomes comprised of vacuolar lipids but not simple PC/PE lipids. Interestingly, even when Gyp7 is forced to bind to PC/PE liposomes, using a His-tag and nickel-chelated-lipids, Gyp7 but is still not very active. This suggests a specific membrane environment is important for both binding and activity of Gyp7.

They find that while loss of Gyp7 has no obvious effect on Ypt7 localization, overexpression of Gyp7 essentially removes Ypt7 from the lysosome (vacuole) membrane and therefore results in enrichment of Ypt7 on endosomes. They find that hyperactivation of Ypt7 at endosomes by overexpression of Gyp7 slows the kinetics of Mup1 endocytosis. Interestingly, this means that both loss of Gyp7 and overexpression of Gyp7 have similar effects on endocytosis. They also find that hyperactivation of Ypt7 on endosomes results in slight resistance to rapamycin. Finally, they observe that overexpression of Gyp7 results in accumulation of the endocytic tracer FM-464 in Ypt7-positive endosomes in the absence of ESCRT function. Taken together the authors interpret these results to mean that Ypt7 functions on "signaling" endosomes.

Just for clarification - the reviewer might have misunderstood our data in part. If Gyp7 is overproduced, we observe faster endocytosis, whereas the deletion of Gyp7 results in slower endocytosis of Mup1 (Fig. 8D,E).

Overall this is an interesting study but at times I found the explanation or interpretation of results to be a bit unclear. Below are my suggestions for improvement:

1. I found the presentation of the Gyp7 localization results to be a bit unclear regarding which compartment the authors consider it to localize to. Is it possible that the differential localization of Gyp7 and other endosomal proteins reflects different timing/kinetics rather than distinct compartments? For example, different Golgi proteins appear to have different localizations but when observed over time they are seen to localize to the same compartment just with different kinetics. This possibility is mentioned in the discussion but it would be good to clarify and mention this possibility when the results are presented. These are the phrases that made me a bit confused: "We further show that Gyp7 overproduction can retain Ypt7 on late endosomes, which enhances endosomal TORC1 signaling. These Ypt7-positive endosomes lack ESCRTs, yet require ESCRTs for their formation. We thus speculate that these late endosomes correspond to signaling endosomes." and "We thus conclude that Ypt7 functions on mature MVBs, which in part correspond to signaling endosomes." Are signaling endosomes a subset of late endosomes? How are they defined?

We agree with the reviewer that Gyp7 and other endosomal proteins could localize to the same compartment but have different timing/kinetics. Our strongest argument of the endosomal localization is the observation that Gyp7 accumulates in the class E compartment of *vps4Δ* cells. However, we rephrased our statement as we did not really observe a strong colocalization of Gyp7 with any distinct endosomal marker, suggesting a very dynamic association. We did not include this analysis in the data set as not informative, but present it below for the reviewers' information (see also point 1 of Reviewer #2). As we do not know the binding partner of Gyp7, a more specific analysis has to wait the identification of this binding partner.

Furthermore, we interpret our results that signaling endosomes are a subset of late endosomes, where Ypt7 resides. In agreement with this, we find that the Ypt7 confinement by Gyp7 overproduction results in the increased resistance of cells to rapamycin, whereas the deletion of *GYP7* causes a hypersensitivity to rapamycin.

2. Similarly, can the authors include at an earlier point in their manuscript an explicit description of how they are distinguishing "signaling endosomes" from "late endosomes", and also how each of these relates to what has been called the "pre-vacuolar endosome (PVE)"? They have some description of signaling endosomes in the discussion, but I found it confusing to see this term mentioned multiple times in the results sections without understanding how they are distinguishing a signaling endosome from a late endosome or PVE.

We agree with the reviewer that the term signaling endosome has to be introduced by taking the previous nomenclature into account. The PVE is probably a mixture of the Vps21-positive endosomes and the Ypt7-positive late endosomes. Within the latter ones, the signaling endosomes will be a subpopulation. We adjusted the introduction accordingly.

3. In Figure 1, how can the authors distinguish the difference between disrupted endosomal morphology versus disrupted Gyp7 recruitment to endosomes? Also, what is special about Mvp1 versus other ESCRT components?

We agree with the reviewer that altered Gyp7 localization in strains lacking endosomal proteins, in particular the Class D mutants (*vps21Δ ypt52Δ*, *vps9Δ muk1Δ*, *vps3Δ*, *vps45Δ*), could be caused by disrupted endosomal morphology or disrupted recruitment of Gyp7 onto endosomes. Therefore, we now propose two possible scenarios in the text.

Mvp1 is one of the proteins involved in retrograde transport, but is not an ESCRT protein. It is part of a family of proteins with BAR domains (Chi et al., JCS 2014). Interestingly, the

number of Gyp7 puncta per cell is decreased in the *mvp1Δ* deletion mutant but not in other deletion mutants impaired in retrograde transport such as *vps35Δ*, *vps5Δ*, and *snx4Δ* cells (Figure 1E,F, S1A). This suggests that Gyp7 localization is somehow linked to one of the retrograde pathways from the endosome to the Golgi (Suzuki et al., *elife* 2021).

4. The following two statements seem to conflict with each other, and I think the second statement is more accurate than the first statement:

"Our data suggest that a functional Rab5 system is required for correct Gyp7 localization to endosomes." (line 165)

"This suggests that Gyp7 recruitment to endosomes occurs independent of the analyzed endosomal proteins. (line 176)

We agree with the reviewer that the two statements conflict each other. Gyp7 recruitment does not depend on the presence of single endosomal proteins as their absence does not lead to loss of membrane localization of Gyp7. The differential localization of Gyp7 in all Class D mutants is presumably caused by a disrupted endolysosomal system per se. We adjusted the text accordingly.

5. It would be very helpful to include a more straightforward analysis of the relationship between Gyp7 and Ypt7 localizations. The experiments involving how overexpression of Gyp7 induce more Ypt7 localization at endosomes, which is apparently the same compartment where Gyp7 itself localizes, are a bit puzzling. In principle one would expect a GAP to antagonize the localization of its Rab. One possibility is that overexpression of Gyp7 causes a shift in localization of Gyp7 to the vacuole. It would be straightforward for the authors to test if this is the case by repeating the Gyp7 overexpression experiments using a fluorescently-tagged version of Gyp7. This could potentially provide a simple explanation for the observed effects on Ypt7 localization. For example, in Figure 6A, the localization of Ypt7 is shown with and without Gyp7 and when Gyp7 is overexpressed, but Gyp7 localization itself is not observed at the same time.

We appreciate the reviewer's comment and analyzed the localization of fluorescently-tagged Gyp7 relative to Ypt7 upon endogenous or overexpression Gyp7 as well as upon expression of the hyperactive *Mon1^{Δ100}-Ccz1* (Fig. 6C,E). Gyp7 does not colocalize with Ypt7 puncta in both condition, whereas the colocalization of Gyp7 and the GEF subunit Ccz1 strongly increases upon overexpression of Gyp7 (Fig. 6C,D). This suggests that the Ypt7's GEF and GAP can indeed localize to the same endosomal compartment, while Ypt7 shifts from a vacuolar to an endosomal population. Importantly, overexpressed Gyp7 does not localize to and inactivate Ypt7 on the vacuole.

6. I think the sentence: "Thus, Gyp7 function is required for normal TORC1 activity within the endolysosomal system" (lines 220-221) is a bit of an overstatement at this point in the manuscript because the authors have only shown sensitivity to Rapamycin and have not shown any direct measure of TORC1 activity (i.e. changes in substrate phosphorylation).

The reviewer is right. We can only interpret the endosomal or vacuolar TORC1 activity from Fig. S7 on. We discuss the effect of Gyp7 function on TORC1 activity in more detail below.

7. The loss of Gyp7 function does not affect Ypt7 localization. One might expect Ypt7 to

have a more broad or intense localization in the absence of its GEF. Can the authors comment on whether this might be because another GYP gene also acts as a GAP for Ypt7?

We agree with the reviewer that one might expect altered Ypt7 localization in the absence of its GAP Gyp7, which might be overwritten by the function of another GAP. Indeed, our previous study indicated that the GAP of Vps21 and Sec4, Msb3, can inactivate Ypt7 as well, since it inhibits *in vitro* vacuole fusion (Lachmann et al., 2012). Therefore, we analyzed Ypt7 localization in the *msb3Δ* mutant as well as in the *gyp7Δ msb3Δ* strain (Fig. S5 A-B). Interestingly, we noticed a slight, though significant decrease in the number of Ypt7 puncta per cell in the double deletion strain, indicating that indeed multiple GAPs could affect Ypt7 localization and activity. However, we believe that Gyp7 is the major Ypt7 GAP as also shown in previous studies and other GAPs probably function only upon loss of Gyp7 function or under certain conditions. This could explain why loss of Gyp7 function alone does not affect Ypt7 localization. We incorporated this possibility in the text accordingly.

8. There appears to be some redundancy in these two sentences: (line 406) "Surprisingly, Gyp7 overproduction does not liberate Ypt7 from endosomes, but rather confines it to a subpopulation proximal to the vacuole. This effect is even stronger when Gyp7 is overexpressed, and ..."

We agree with the reviewer and modified the text accordingly.

Reviewer #2 (Comments to the Authors (Required)):

In this clearly written manuscript, Füllbrunn and coworkers report studies of the budding yeast Rab GAP Gyp7. They present genetic and cell biological studies which confirm and extend prior work from three other labs showing that Gyp7 is the major GAP that inactivates Rab7 (Ypt7), and present data which they interpret to indicate that an endosomal compartment or compartments is the major *in vivo* site of Gyp7 action. Biochemical experiments show that Gyp7 has a membrane binding activity that exhibits selectivity for lipid composition. Several of the reported experiments are interesting but as discussed below key conclusions are based on non-physiological genetic perturbations (overexpression) and several experiments do not include controls necessary for interpretation of the results, tempering my enthusiasm for the manuscript. It is possible that some of the needed data are already in hand but not shown. With some additions and a more tempered interpretation of the results, I'd be happy to take another look at this study.

Major points.

1. "Gyp7 localizes to endosomes." [line 142] The authors show that overexpressed Gyp7 localizes to punctate structures that appear to label with the endocytic tracer FM4-64. However, no co-localization with known protein markers of endosomes is shown, except to a limited extent in a *vps4Δ* background, where dozens of markers accumulate at class E compartments. This is an odd omission. Moreover, the authors see *more* localization of Gyp7 to punctate structures when Rab5 or Rab5 effector function is impaired, not less - and these punctae do not seem to be marked by FM4-64. It is hard to see this as support for the hypothesis that Gyp7 localizes to endosomes. Could these be, for example, Atg8 accumulations rather than endosomes?

Reviewer 1 had similar points, and we have not identified the identity of Gyp7 puncta yet. We tested colocalization with many endosomal markers and Atg8, but did not find any significant colocalization. Even by time-lapse imaging, we were unable to find colocalization. However, Gyp7 accumulates in Class E endosomes if *vps4* is deleted. This observation is quite similar to the behavior of the GEF Vps9, which is mainly cytosolic, and only found in endosomes under these conditions. In addition, overexpressed Gyp7 colocalizes strongly with Mon1-Ccz1, next to Ypt7 puncta, suggesting an endosomal origin also of this structure. We speculate that the Gyp7-positive puncta in wild-type cells might correspond to Rab5-deficient endosomal structures.

2. "Relocalization of Gyp7 to vacuoles impairs vacuole morphology." [line 178] This is a reasonable conclusion on the basis of overexpression as previously reported and experiments shown here (Fig. 2A,B). However, the re-targeting experiments (Fig. 2C,E) show much larger effects for the affinity-tagged Vac8-CB used as an anchor to relocalize Gyp7 than for the relocalization itself. Or perhaps I'm misreading the experiment? I asked two other experienced people in my lab to read this section of the paper, and they read it the same way. I don't see how this experiment can be interpreted using a background with what seems to be a reasonably strong *vac8* hypomorph.

We understand the reviewer's concern regarding this experiment since chromobody-fused Vac8 seems to have partially impaired function. Therefore, we repeated the experiment with the chromobody fused to Zrc1, a vacuolar membrane zinc transporter, as an additional readout for vacuolar recruitment of Gyp7. Here, the number of vacuoles does not increase upon simple tagging of Zrc1 with the chromobody, while recruitment of Gyp7 to the vacuole via Zrc1-CB causes a strong vacuolar morphology defect. Thus, we replaced the microscopy data of chromobody-fused Vac8 with chromobody-fused Zrc1 (Fig. 2C-E).

Additionally, it's hard to see how expression of a presumptively spontaneous nucleotide-exchanging variant of Ypt7 is a better control here than a catalytic-dead Gyp7 (R458K), as used in previous studies (Eitzen, EMBO J 2000; Brett, JCB 2008). Use of this well-characterized mutant could have strengthened several experiments in the present study. It's perplexing that R458K was not employed in this study.

We appreciate the reviewer's suggestion. In addition to the expression of the Ypt7^{K127E} in the Gyp7-GFP Zrc1-CB background (Fig. 2D,E), we included expression of the catalytic-dead Gyp7 in our relocalization experiments. Importantly, we are able to show that recruitment of Gyp7^{R458K}-GFP to neither the vacuole (Zrc1-CB) or endosomes (Vps8-CB) affects vacuole morphology (Fig. 2F,G). Therefore, our data provides evidence that relocalization of functional Gyp7 to the vacuole and thus GAP-mediated Ypt7 inactivation impairs vacuole morphology.

3. "Gyp7 is required for homeostasis of the endosomal system." [line 204] The authors show data suggesting that perturbation of Gyp7 function alters TORC1 signaling, consistent with the known role of endolysosomal traffic in the TORC1 pathway. It is interesting that an *msb3* (Rab5 GAP) mutant phenocopies the *gyp7* deletion for this readout.

Data are also shown suggesting that traffic kinetics through the endosomal MVB pathway to

the vacuole are (very) subtly regulated by Gyp7 activity. The experiments do not clearly delineate whether the target of this regulation is Ypt7 residing on the endosome, on the vacuole, or both.

We agree with the reviewer that our experiments do not clearly distinguish which pool of Ypt7 is primarily targeted by Gyp7. However, Gyp7 is only found in puncta and not on the vacuolar membrane, even upon overexpression of the protein, which shifts a Ypt7 pool from the vacuole to endosomes (Fig. 6A,B). Therefore, it is likely that Gyp7 acts on the endosomal Ypt7 pool.

4. "Gyp7 activity depends on the membrane environment." [line 232]. It is persuasively shown that Gyp7 binds membranes, that it prefers to bind membranes with a vacuole-like membrane mixture (an endosomal vs. vacuole lipid mixture was not tested, as might have been expected given the overall argument of the paper), and that this activity depends on a PH-like domain near the protein's N-terminus. The PH-like domain alone does not bind membranes in the experimental configurations employed.

The authors use mainly GDI extraction as a proxy for Gyp7 activity against Ypt7/Rab7. There's nothing wrong with this approach, as such. But curiously direct assay of Ypt7 GTPase activity is reported solely in Fig. 5J. The authors claim that this shows allosteric regulation of Gyp7 activity against soluble (non-lipidated) Ypt7 by membranes. The result shows a very small but apparently reproducible difference in activity. But given the advantages of a chemically defined system, why was GTPase activity not assayed directly throughout? This is not hard to do using well-described colorimetric, fluorescence, or [32]P orthophosphate release assays, or presumably the HPLC assay in Fig. 5J.

Given the absence of direct readouts of GTP hydrolysis, it is important to test whether the lipid mix used (VML vs. PC/PE) influences the ability of GDI to extract Ypt7-GDP. This control is important if extraction is used as the main proxy for the Rab's nucleotide state. Also, it was not clear to this reader whether GDI is present in excess to Ypt7, or what the final GDI concentration was in the extraction experiments.

We agree with the reviewer that it is an important control to show whether GDI is able to extract Ypt7-GDP from PC/PE liposomes. In our normal experimental setup, the molar ratio between Ypt7 and GDI is 1:1 (600 nM each). Now we provide data, which show that a 10x excess of GDI (6 μ M) does not lead to further extraction of Ypt7 either bound to VMLs or to PC/PE liposomes (Fig. S3). Furthermore, we analyzed the extraction of Ypt7 from liposomes after incubation with the catalytically active TBC domain of Gyp1 (Gyp1-46), which does not rely on membranes for its activity (Fig. 4L,M). Here, we observed no difference in GDI-mediated extraction of Ypt7 from VMLs vs. PC/PE liposomes, indicating that GDI in principle is able to extract Ypt7-GDP from both VMLs as well as from PC/PE liposomes. Together, the data show that the function of Gyp7 but not of GDI depends on the membrane composition.

Overall, the experiments support the idea that direct membrane association increases Gyp7 activity against Ypt7. They do not strongly support the idea that membrane association has a major allosteric effect on Gyp7 catalytic activity.

We agree with the reviewer's comment and adjusted the text accordingly. We currently do not know how the membrane composition influences Gyp7 activity, and we can only speculate here.

5. "Gyp7 activity confines Ypt7 to late endosomes and signaling endosomes." Taken literally, this is obviously wrong, since Ypt7 on vacuoles is needed for vacuole fusion, as exhaustively demonstrated by many labs including the authors', and the data show (as entirely expected) lots of Ypt7 on the vacuole in wild type cells. Fig. 6A also shows that overproduction of Gyp7 removes Ypt7 from the vacuole, and if anything, increases its localization to (presumptively) endosomal punctae. This would seem to argue that Gyp7 preferentially targets Ypt7 on the vacuole, not on the endosome as the authors suggest earlier in the manuscript.

The reviewer is right; the statement is not quite correct and misleading. We meant to say that the pool of Ypt7 is shifted from a primary vacuole localization to a strongly confined endosomal pool. As Gyp7 only found in puncta and not at the vacuolar rim, we interpret this in favor of an inactivation of Ypt7 here in endosomal compartments rather than on the vacuole. Of course, we cannot exclude an additional role of Gyp7 on the vacuole, which may escape our detection. We therefore discussed this issue in more detail in the manuscript.

Other experiments here are based on a truncation of the GEF subunit Mon1 that results in elevated Ypt7 activity, as nicely shown in recent work from the same group. But Gyp7 is not shown to colocalize with Mon1 or Ypt7 under these circumstances. An interesting observation is that endosomes marked by Pep12 increase in number in a $MON1\Delta100$ mutant that also overproduces Gyp7. However, it's not tested whether this phenotype is due to one of these genetic manipulations, or both (Fig. 6E).

Importantly, the overall number of Pep12 puncta per cell does not increase but decrease, while the number of Pep12 puncta, which do not colocalize with the vacuole, significantly increases (see Fig. 7C,D, Fig. S6B). However, we agree with the reviewer and further dissected whether one or both genetic manipulations cause this phenotype. Interestingly, we find that overproduction of Gyp7 leads to the overall decrease of Pep12 puncta and their localization distant from the vacuole. Expression of the truncated and hyperactive GEF causes a slight, though significant decrease in the number of the same structures, while it does not affect the subcellular distribution of Pep12 puncta. Thus, the data suggest that Gyp7 does not only affect Ypt7 localization and TORC1 activity but is rather important for the overall endosomal system organization/functioning.

Furthermore, we addressed the colocalization of Gyp7 with Mon1-Ccz1 and Ypt7 in wild-type cells as well as upon genetic manipulation of the Ypt7 GEF and GAP. We find that Gyp7 does not colocalize with Ypt7 puncta in both condition, whereas the colocalization of Gyp7 and the GEF subunit Ccz1 strongly increases upon overexpression of Gyp7 (Fig. 6C,D). This result suggests that the Ypt7 GEF and GAP indeed localize to the same endosomal compartment, while Ypt7 shifts from a vacuolar to an endosomal population. Importantly, overexpressed Gyp7 does not localize and inactivate Ypt7 on the vacuole.

6. "Endolysosomal transport is delayed upon Ypt7 confinement to late endosomes." [line 338]. The delays are again subtle but apparently statistically significant, and consistent with

the ability of Gyp7 to deplete Ypt7 from the vacuole as shown in Fig. 6A.

We rephrased this part to make clear that this is a subtle defect. This is probably also expected for a regulator of Ypt7 activity such as a GAP.

7. "Ypt7-positive structures correspond to signaling endosomes." Immunogold EM shows that overproduced Ypt7 can be detected on endosomal structures, and Ypt7 accumulates on Class E compartments in a *vps4Δ* mutant (along with dozens of other endolysosomal proteins). In Fig. S6A,B a reporter system is used to assay endosomal vs. vacuolar phosphorylation of Sch9 by Tor1. In a *gyp7Δ* mutant vs. wild type, a significant decrease in TORC1 activity is seen at the vacuole and *not* at the endosome. Overproduction does increase signaling at the endosome, but given the lack of a deletion phenotype, this is not a strong argument for a normal physiological function of Gyp7 at the endosome per se. I wonder if stronger phenotypes would emerge in nitrogen limited conditions.

We appreciate the reviewer's suggestion, yet also disagree in part. The *gyp7Δ* mutant does not impair vacuole morphology, yet has a clear defect in Mup1 uptake and in TORC1 signaling. The phenotype is certainly not as drastic as a fusion mutant, which is also not expected, given that Gyp7 is a regulator of Ypt7. Nevertheless, we agree with the reviewer that the *gyp7Δ* mutant might show a stronger phenotypic response if cells are additionally challenged by nitrogen starvation. Therefore, we compared vacuole morphology of wild-type vs. *gyp7Δ* cells upon 2 h of nitrogen starvation (Fig. 3F-H). Again, *gyp7Δ* cells do not behave differently than wild-type cells in both growth conditions and upon nitrogen starvation. As suggested previously (Reviewer #1, comment 7), it is a reasonable possibility that upon loss of Gyp7 function another Ypt7 GAP might take over its function.

Minor issues.

8. The paper by Eitzen (EMBO J 2000) is not cited, and should be.
9. Line 216: In yeast, Apl5 is not an endosomal trafficking protein.
10. Line 224: Fig. 3C is not mentioned in the Results, so far as I can tell.
11. Fig. S2B: genotypes should be labeled.

We agree with the reviewer and addressed these issues.

- Alexey Merz

Reviewer #3 (Comments to the Authors (Required)):

In the present study, Füllbrunn et al. dissect the endocytic localization and function of the Ypt7 (RAB7) specific GAP protein Gyp7 in yeast. While Gyp7 is already known to be a GAP for RAB7, the precise localization and membrane dependency of Ypt7 inactivation through Gyp7 remained to be elucidated.

The authors demonstrate that Gyp7 localizes primarily to endosomes but not to the vacuole and that this localization partially depends on an intact Vps21 (RAB5) system. Additional localization experiments indicate that Gyp7 functions on endosomes but likely not on the

vacuolar membrane. Deletion of Gyp7 delayed endosomal transport towards the vacuole and altered endosomal mTORC1 signaling, suggesting that Gyp7 is required for the homeostasis and signaling function of endosomes. In an additional line of experimentation, the authors demonstrate that Gyp7 requires endosomal membranes for its GAP activity as membrane free Gyp7 was hardly active towards Ypt7. Finally, the authors demonstrate that Gyp7 activity confines Ypt7 to late endosomes which are also signaling endosomes.

Overall, the data is of high quality and the authors' conclusions appear reasonable to this reviewer. The authors thoroughly dissect the localization of Gyp7, its effect on Ypt7 and its role within the endocytic network. With this being said, I think that the manuscript is somewhat uninspiring as Gyp7 was already known to be the dominant Ypt7 GAP protein in yeast. It is still a solid and thorough cell biological analysis of a previously known RAB7 GAP in yeast but it doesn't add a lot of groundbreaking insight into the function of this endocytic protein. While I am generally supportive of publication I am not sure whether JCB is an appropriate venue for this manuscript.

We respectfully disagree with the reviewer's opinion on the suitability of our manuscript for JCB. The manuscript addresses here the role of a GAP in controlling the Rab7 function by taking both functional assays (GAP assays, GAP relocalization, TORC1 activity measurements) and *in vivo* analyses into account. The results of this analysis show that Gyp7 controls Ypt7 function and consequently a pool of late endosomes, for which we have coined the name signaling endosomes. What is most surprising is the strong effect of Gyp7 overproduction on expanding the Ypt7 pool proximal to the vacuole (Ypt7 puncta), and subsequently altering TORC1 signaling. This suggests that Gyp7 functions at an endosomal compartment and controls Ypt7 function here. This analysis is, as the other two reviewers also agree with, novel and unexpected and thus within the general scope of JCB.

Minor points:

Figure 4A: "floatation" seems odd

We corrected this.

**The GTPase activating protein Gyp7 regulates the activity**
**of the Rab7-like Ypt7 on late endosomes**

Nadia Füllbrunn^{1,2}, Raffaele Nicastro³, Muriel Mari⁴, Janice Griffith⁵, Eric Herrmann⁶, René
Rasche⁶, Ann-Christin Borchers¹, Kathrin Auffarth¹, Daniel Kümmel⁶, Fulvio Reggiori^{4,5},
Claudio De Virgilio³, Lars Langemeyer^{1,2*}, Christian Ungermann^{1,2*}

¹ Osnabrück University
Department of Biology/Chemistry
Biochemistry section
49076 Osnabrück, Germany

² Center of Cellular Nanoanalytics (CellNanOS)
Osnabrück University
49076 Osnabrück, Germany

³ University of Fribourg
Department of Biology
CH-1700 Fribourg, Switzerland

⁴ Aarhus University
Department of Biomedicine
8000 Aarhus, Denmark

⁵ University Medical Center Utrecht
Department of Cell Biology
3584 CX Utrecht, The Netherlands

⁶ University of Münster
Institute of Biochemistry
48149 Münster, Germany

* Corresponding authors:
Email: lars.langemeyer@uos.de (L.L.), cu@uos.de (C.U.)
<http://www.biochemie.uni-osnabrueck.de/>
Phone: +49-541-969-2752

**Running title:** GAP control in Rab7 localization and function

**Abstract**

Organelles of the endomembrane system contain Rab GTPases as identity markers.
Localization of Rab GTPases is determined by specific activating guanine nucleotide exchange
factors (GEFs) and GTPase activating proteins (GAPs). It remains largely unclear, however,
how these regulators are specifically targeted to organelles and how their activity is regulated.
Here, we focus on the GAP Gyp7, which acts on the Rab7-like Ypt7 protein in yeast, and
surprisingly observe the protein exclusively in puncta proximal to the vacuole. Mistargeting of
Gyp7 to the vacuole strongly affects vacuole morphology, suggesting that endosomal
localization is needed for function. In agreement, efficient endolysosomal transport requires
Gyp7. *In vitro* assays reveal that Gyp7 requires a distinct lipid environment for membrane
binding and GAP activity. Overexpression of Gyp7 concentrates Ypt7 in late endosomes, and
results in resistance to rapamycin, an inhibitor of the target of rapamycin complex 1 (TORC1),
suggesting that these late endosomes are signaling endosomes. We postulate that Gyp7 is
part of a regulatory machinery involved in late endosome function.

**Keywords:** Gyp7, Ypt7, GAP, Rab GTPase, endosome, lysosome

[revised manuscript text omitted]

as previously described in Griffith et al. (2008). 70 nm ultrathin cryo-sections were stained with
with 2 % uranyl-oxalacetate, pH 7, for 5 min, and methyl-cellulose/uranyl acetate, pH 4, for
additional 5 min. Cell sections were imaged using a Jeol-1400 transmission electron
microscope equipped with a digital camera.

The strain expressing GFP-Ypt7 from the *TEF1* promoter was grown to an exponential phase
before being processed for immunogold labeling of cryosections as previously described
(Griffith et al., 2008). Cryo-sections were labelled with a polyclonal anti-GFP antibody (Abcam,
cat# ab290-50) and viewed in a Jeol 1200 transmission electron microscope (Jeol, Tokyo,
Japan), and images were recorded on Kodak 4489 sheet films (Eastman Kodak, Rochester,
NY).

**Acknowledgements**

We thank Angela Perz for expert technical assistance and Clara Taetz and Kevin Tanzusch
for experimental support. This work was supported by the grants of the Deutsche
Forschungsgemeinschaft (DFG) to C.U. (SFB 944, P11; SFB 1557, P14), and to D.K. (SFB
944, P17; SFB 1557, P10), and the Swiss National Science Foundation (310030_184671) to
C.D.V. F.R. is supported by Open Competition ENW-KLEIN (OCENW.KLEIN.118), SNSF
Sinergia (CRSII5_189952) and Novo Nordisk Foundation (0066384) grants.

**Author contributions**

CU and LL conceived the project together with NF. NF performed all biochemistry and cell
biology experiments with support of ACB. RN and CdV conducted and interpreted the TORC1
activity assays. MM, JG and FR conducted and interpreted the IEM analysis. EH, RR and DK

[revised manuscript text omitted]

February 27, 2024

RE: JCB Manuscript #202305038R-A

Prof. Christian Ungermann
Osnabrück University
Biology/Chemistry
Barbarastrasse 13
Osnabrück 49076
Germany

Dear Prof. Ungermann:

Thank you for submitting your revised manuscript entitled "The GTPase activating protein Gyp7 regulates the activity of the Rab7-like Ypt7 on late endosomes". We would be happy to publish your paper in JCB pending resolution of remaining minor concerns by reviewers, and final revisions necessary to meet our formatting guidelines (see details below).

A. MANUSCRIPT ORGANIZATION AND FORMATTING:

Full guidelines are available on our Instructions for Authors page, <http://jcb.rupress.org/submission-guidelines#revised>. Submission of a paper that does not conform to JCB guidelines will delay the acceptance of your manuscript.

1) Text limits: Character count for Articles is < 40,000, not including spaces. Count includes abstract, introduction, results, discussion, and acknowledgments. Count does not include title page, figure legends, materials and methods, references, tables, or supplemental legends.

2) Figures limits: Articles may have up to 10 main figures and 5 supplemental figures/tables.

** Please combine supplemental figure panels into corresponding main figure panels, or eliminate supplemental data to reduce total supplemental figures to 5. If appropriate, an additional main figure may also be generated.

3) Figure formatting: Scale bars must be present on all microscopy images, including inset magnifications. Molecular weight or nucleic acid size markers must be included on all gel electrophoresis. Please avoid pairing red and green for images and graphs to ensure legibility for color-blind readers. If red and green are paired for images, please ensure that the particular red and green hues used in micrographs are distinctive with any of the colorblind types. If not, please modify colors accordingly or provide separate images of the individual channels.

** Please include scale bars on Figure 9B and S1A.

** Please add molecular weight markers to Fig S7A.

4) Statistical analysis: Error bars on graphic representations of numerical data must be clearly described in the figure legend. The number of independent data points (n) represented in a graph must be indicated in the legend. Statistical methods should be explained in full in the materials and methods. For figures presenting pooled data the statistical measure should be defined in the figure legends. Please also be sure to indicate the statistical tests used in each of your experiments (either in the figure legend itself or in a separate methods section) as well as the parameters of the test (for example, if you ran a t-test, please indicate if it was one- or two-sided, etc.). Also, if you used parametric tests, please indicate if the data distribution was tested for normality (and if so, how). If not, you must state something to the effect that "Data distribution was assumed to be normal but this was not formally tested."

5) Abstract and title: The abstract should be no longer than 160 words and should communicate the significance of the paper for a general audience. The title should be less than 100 characters including spaces. Make the title concise but accessible to a general readership.

** We recommend changing the title to something slightly shorter: "The GTPase activating protein Gyp7 regulates Rab7/Ypt7 activity on late endosomes"

6) Materials and methods: Should be comprehensive and not simply reference a previous publication for details on how an experiment was performed. Please provide full descriptions in the text for readers who may not have access to referenced manuscripts. We also provide a report from SciScore and an associate score, which we encourage you to use as a means of evaluating and improving the methods section.

** Please provide full details for in vitro prenylation of Rab GTPases and immune-electron microscopy.

7) Please be sure to provide the sequences for all of your primers/oligos and RNAi constructs in the materials and methods. You must also indicate in the methods the source, species, and catalog numbers (where appropriate) for all of your antibodies. Please also indicate the acquisition and quantification methods for immunoblotting/western blots.

8) Microscope image acquisition: The following information must be provided about the acquisition and processing of images:

- a. Make and model of microscope
- b. Type, magnification, and numerical aperture of the objective lenses
- c. Temperature
- d. Imaging medium
- e. Fluorochromes
- f. Camera make and model
- g. Acquisition software
- h. Any software used for image processing subsequent to data acquisition. Please include details and types of operations involved (e.g., type of deconvolution, 3D reconstitutions, surface or volume rendering, gamma adjustments, etc.).

10) Supplemental materials: There are strict limits on the allowable amount of supplemental data. Articles may have up to 5 supplemental figures. Please also note that tables, like figures, should be provided as individual, editable files. A summary of all supplemental material should appear at the end of the Materials and methods section.

13) ORCID IDs: ORCID IDs are unique identifiers allowing researchers to create a record of their various scholarly contributions in a single place. At resubmission of your final files, please provide an ORCID ID for all authors.

15) A data availability statement is required for all research article submissions. The statement should address all data underlying the research presented in the manuscript. Please visit the JCB instructions for authors for guidelines and examples of statements at (<https://rupress.org/jcb/pages/editorial-policies#data-availability-statement>).

Please note that JCB requires authors to submit Source Data used to generate figures containing gels and Western blots with all revised manuscripts. This Source Data consists of fully uncropped and unprocessed images for each gel/blot displayed in the main and supplemental figures. Since your paper includes cropped gel and/or blot images, please be sure to provide one Source Data file for each figure that contains gels and/or blots along with your revised manuscript files. File names for Source Data figures should be alphanumeric without any spaces or special characters (i.e., SourceDataF#, where F# refers to the associated main figure number or SourceDataFS# for those associated with Supplementary figures). The lanes of the gels/blots should be labeled as they are in the associated figure, the place where cropping was applied should be marked (with a box), and molecular weight/size standards should be labeled wherever possible. Source Data files will be directly linked to specific figures in the published article.

B. FINAL FILES:

-- High-resolution figure and MP4 video files: See our detailed guidelines for preparing your production-ready images,

<https://jcb.rupress.org/fig-vid-guidelines>.

Thank you for this interesting contribution, we look forward to publishing your paper in Journal of Cell Biology.

Sincerely,

Harald Stenmark
Monitoring Editor
Journal of Cell Biology

Tim Fessenden
Scientific Editor
Journal of Cell Biology

Reviewer #1 (Comments to the Authors (Required)):

In this revision, all of my concerns have been addressed except for one very minor issue. I do not agree with the statement "Overall, we suggest that Gyp7 activity shifts Ypt7 from a primary vacuolar localization to a subset of endosomes" (lines 351-352). Under wild-type conditions, Ypt7 localization is primarily at the vacuole and Gyp7 localization is on endosomes. Only under perturbation conditions (Gyp7 overexpression) does Ypt7 relocalize to endosomes. The overexpression condition does not tell us what the function of normal Gyp7 activity is, as overexpression may cause gain of function. I suggest removing this sentence.

Reviewer #2 (Comments to the Authors (Required)):

I think the authors have addressed all of the most significant technical concerns with the experiments.

I am still on the fence with respect to the "signaling endosome" concept (at least in budding yeast). For example, it remains unclear (to me, at least) whether these entities represent one of a diversified set of distinct endosomal maturation pathways, or instead a transient intermediate on a common maturation pathway. This question does not seem to be resolved.

Hence, I could argue with the conceptual framework. However, I think that there are many useful experiments in this study, and that the conceptual framing is at least plausible enough to serve as a useful working model. Consequently I think that both the results and ideas here will move the field forward, and that the paper should be published without additional delay.

Minor points:

Fig 2 C-E, It might be worth mentioning that the estimated copy number of Zrc1 is significantly higher than that of Gyp7 [<http://dx.doi.org/10.1016/j.cels.2017.12.004>]. Excess of receptor over ligand-fused target protein is an important precondition for a knock-sideways experiment, and here it seems to be satisfied.

The paper by Generoso et al. (cited in Methods) is not in the Bibliography.

Reviewer #3 (Comments to the Authors (Required)):

My only real concern was a lack of novelty. This concern was apparently not shared by the other two reviewers and the editors, rendering my comment obsolete. I thought that the data were of high quality and the conclusions appeared sound so there are no further objections from my side.

Universität Osnabrück · FB 5 · 49076 Osnabrück

To
Andrea Marat
Senior Editor at *JCB*

Fachbereich
Biologie/Chemie

Abt. Biochemie

PROF. DR. CHRISTIAN UNGERMANN
(HANS-MÜHLENHOFF-
STIFTUNGSPROFESSUR)

Barbarastraße 13
49076 Osnabrück
Telefon: +49 541 969 2752
Telefax: +49 541 969 2884

www.uni-osnabrueck.de

Dear Andrea, dear Harald,

In response to the reviewers' requests and the editorial instructions, we have adjusted the manuscript. Our corrections are indicated below.

Best,

A. MANUSCRIPT ORGANIZATION AND FORMATTING:

Full guidelines are available on our Instructions for Authors page, <http://jcb.rupress.org/submission-guidelines#revised>. Submission of a paper that does not conform to JCB guidelines will delay the acceptance of your manuscript.

1) Text limits: Character count for Articles is < 40,000, not including spaces. Count includes abstract, introduction, results, discussion, and acknowledgments. Count does not include title page, figure legends, materials and methods, references, tables, or supplemental legends.

Done

2) Figures limits: Articles may have up to 10 main figures and 5 supplemental figures/tables.

** Please combine supplemental figure panels into corresponding main figure panels, or eliminate supplemental data to reduce total supplemental figures to 5. If appropriate, an additional main figure may also be generated.

Supplemental Figures have been adjusted to 5 in total.

3) Figure formatting: Scale bars must be present on all microscopy images, including

inset magnifications. Molecular weight or nucleic acid size markers must be included on all gel electrophoresis. Please avoid pairing red and green for images and graphs to ensure legibility for color-blind readers. If red and green are paired for images, please ensure that the particular red and green hues used in micrographs are distinctive with any of the colorblind types. If not, please modify colors accordingly or provide separate images of the individual channels.

** Please include scale bars on Figure 9B and S1A.

** Please add molecular weight markers to Fig S7A.

Done

4) **Statistical analysis:** Error bars on graphic representations of numerical data must be clearly described in the figure legend. The number of independent data points (n) represented in a graph must be indicated in the legend. Statistical methods should be explained in full in the materials and methods. For figures presenting pooled data the statistical measure should be defined in the figure legends. Please also be sure to indicate the statistical tests used in each of your experiments (either in the figure legend itself or in a separate methods section) as well as the parameters of the test (for example, if you ran a t-test, please indicate if it was one- or two-sided, etc.). Also, if you used parametric tests, please indicate if the data distribution was tested for normality (and if so, how). If not, you must state something to the effect that "Data distribution was assumed to be normal but this was not formally tested."

A section has been added.

5) **Abstract and title:** The abstract should be no longer than 160 words and should communicate the significance of the paper for a general audience. The title should be less than 100 characters including spaces. Make the title concise but accessible to a general readership.

** We recommend changing the title to something slightly shorter: "The GTPase activating protein Gyp7 regulates Rab7/Ypt7 activity on late endosomes"

We agree and adjusted the title accordingly.

6) **Materials and methods:** Should be comprehensive and not simply reference a previous publication for details on how an experiment was performed. Please provide full descriptions in the text for readers who may not have access to referenced manuscripts. We also provide a report from SciScore and an associate score, which we encourage you to use as a means of evaluating and improving the methods section.

** Please provide full details for in vitro prenylation of Rab GTPases and immune-electron microscopy.

Now included.

7) Please be sure to provide the sequences for all of your primers/oligos and RNAi constructs in the materials and methods. You must also indicate in the methods the

source, species, and catalog numbers (where appropriate) for all of your antibodies. Please also indicate the acquisition and quantification methods for immunoblotting/western blots.

A primer Table is included.

8) Microscope image acquisition: The following information must be provided about the acquisition and processing of images:

- a. Make and model of microscope
- b. Type, magnification, and numerical aperture of the objective lenses
- c. Temperature
- d. Imaging medium
- e. Fluorochromes
- f. Camera make and model
- g. Acquisition software
- h. Any software used for image processing subsequent to data acquisition. Please include details and types of operations involved (e.g., type of deconvolution, 3D reconstitutions, surface or volume rendering, gamma adjustments, etc.).

All details have been included now.

This has been done.

10) Supplemental materials: There are strict limits on the allowable amount of supplemental data. Articles may have up to 5 supplemental figures. Please also note that tables, like figures, should be provided as individual, editable files. A summary of all supplemental material should appear at the end of the Materials and methods section.

Done.

Done.

Done.

13) ORCID IDs: ORCID IDs are unique identifiers allowing researchers to create a record of their various scholarly contributions in a single place. At resubmission of your final files, please provide an ORCID ID for all authors.

Done.

Done.

15) A data availability statement is required for all research article submissions. The statement should address all data underlying the research presented in the manuscript. Please visit the JCB instructions for authors for guidelines and examples of statements at (<https://rupress.org/jcb/pages/editorial-policies#data-availability-statement>).

Please note that JCB requires authors to submit Source Data used to generate figures containing gels and Western blots with all revised manuscripts. This Source Data consists of fully uncropped and unprocessed images for each gel/blot displayed in the main and supplemental figures. Since your paper includes cropped gel and/or blot images, please be sure to provide one Source Data file for each figure that contains gels and/or blots along with your revised manuscript files. File names for Source Data figures should be alphanumeric without any spaces or special characters (i.e., SourceDataF#, where F# refers to the associated main figure number or SourceDataFS# for those associated with Supplementary figures). The lanes of the gels/blots should be labeled as they are in the associated figure, the place where cropping was applied should be marked (with a box), and molecular weight/size standards should be labeled wherever possible. Source Data files will be directly linked to specific figures in the published article.

A statement has been added accordingly.

B. FINAL FILES:

Thank you for this interesting contribution, we look forward to publishing your paper in Journal of Cell Biology.

Sincerely,

Harald Stenmark
Monitoring Editor
Journal of Cell Biology

Tim Fessenden
Scientific Editor
Journal of Cell Biology

Reviewer #1 (Comments to the Authors (Required)):

In this revision, all of my concerns have been addressed except for one very minor issue. I do not agree with the statement "Overall, we suggest that Gyp7 activity shifts Ypt7 from a primary vacuolar localization to a subset of endosomes" (lines 351-352). Under wild-type conditions, Ypt7 localization is primarily at the vacuole and Gyp7 localization is on endosomes. Only under perturbation conditions (Gyp7 overexpression) does Ypt7 relocate to endosomes. The overexpression condition does not tell us what the function of normal Gyp7 activity is, as overexpression may cause gain of function. I suggest removing this sentence.

Thank you. We removed the sentence as requested.

Reviewer #2 (Comments to the Authors (Required)):

I think the authors have addressed all of the most significant technical concerns with the experiments.

I am still on the fence with respect to the "signaling endosome" concept (at least in budding yeast). For example, it remains unclear (to me, at least) whether these entities represent one of a diversified set of distinct endosomal maturation pathways, or instead a transient intermediate on a common maturation pathway. This question does not seem to be resolved.

Hence, I could argue with the conceptual framework. However, I think that there are many useful experiments in this study, and that the conceptual framing is at least plausible enough to serve as a useful working model. Consequently I think that both the results and ideas here will move the field forward, and that the paper should be published without additional delay.

Thank you for the kind feedback.

Minor points:

Fig 2 C-E, It might be worth mentioning that the estimated copy number of Zrc1 is significantly higher than that of Gyp7 [<http://dx.doi.org/10.1016/j.cels.2017.12.004>]. Excess of receptor over ligand-fused target protein is an important precondition for a knock-sideways experiment, and here it seems to be satisfied.

Thank you, the reference was included and a statement was added to the text.

The paper by Generoso et al. (cited in Methods) is not in the Bibliography.

Now added.

Reviewer #3 (Comments to the Authors (Required)):

My only real concern was a lack of novelty. This concern was apparently not shared by the other two reviewers and the editors, rendering my comment obsolete. I thought that the data were of high quality and the conclusions appeared sound so there are no further objections from my side.

We appreciate that the reviewer acknowledges our efforts and agrees with the publication.